# Optimal Rates for Bandit Nonstochastic Control

**Y. Jennifer Sun** *
Princeton University
Google DeepMind
ys7849@princeton.edu

**Stephen Newman** *
Princeton University
sn9581@princeton.edu

**Elad Hazan**
Princeton University & Google DeepMind
ehazan@princeton.edu

## Abstract

Linear Quadratic Regulator (LQR) and Linear Quadratic Gaussian (LQG) control are foundational and extensively researched problems in optimal control. We investigate LQR and LQG problems with semi-adversarial perturbations and time-varying adversarial bandit loss functions. The best-known sublinear regret algorithm of Gradu et al. [2020] has a $T^{\frac{3}{4}}$ time horizon dependence, and the authors posed an open question about whether a tight rate of $\sqrt{T}$ could be achieved. We answer in the affirmative, giving an algorithm for bandit LQR and LQG which attains optimal regret (up to logarithmic factors) for both known and unknown systems. A central component of our method is a new scheme for bandit convex optimization with memory, which is of independent interest.

## 1   Introduction

Linear-Quadratic Regulator (LQR) and the more general Linear-Gaussian (LQG) control problems have been extensively studied in the field of control theory due to their wide range of applications and admittance of analytical solutions by the seminal works of Bellman [1954] and Kalman [1960]. LQR and LQG control problems study the design of a feedback control policy for a linear dynamical system with the goal of minimizing cumulative, possibly time-varying quadratic costs. The discrete version of the problem studies the control of the following linear dynamical system governed by dynamics $(A, B, C)$ [1]:

$$\mathbf{x}_{t+1} = A\mathbf{x}_t + B\mathbf{u}_t + \mathbf{w}_t \ , \ \mathbf{y}_t = C\mathbf{x}_t + \mathbf{e}_t \ ,$$

where at time $t$, $\mathbf{x}_t$ represents the system's state, $\mathbf{u}_t$ represents the control exerted on the system, and $\{\mathbf{w}_t\}_{t=1}^T$ represents a sequence of i.i.d. centered Gaussian perturbations injected to the system. For a simple example of the model, consider the problem of controlling an elevator to reach the desired floors while minimizing the discomfort (due to acceleration or deceleration) and the energy consumption. In this case, the state $\mathbf{x}_t$ would be a vector consisting of the elevator's position and velocity at time $t$. The control input $\mathbf{u}_t$ is the force applied by the elevator motor at time $t$. $\mathbf{w}_t$ represents process noise at time $t$, caused possibly by unexpected mechanical disturbances. In the generality of LQG, the system's states are not accessible. In the elevator example, one may not have the access to the measurement of the elevator's velocity or the precise position of the elevator. Instead, the algorithm has access to an observation $\mathbf{y}_t$, which is a linear transformation of state perturbed by a sequence $\{\mathbf{e}_t\}_{t=1}^T$ of i.i.d. centered Gaussian noises, i.e. the elevator's observed position. The cost is

---

[1]The LQR/LQG dynamics can be generalized to time-varying linear dynamical systems. Here we restrict ourselves to linear time-invariant systems for simplicity.

37th Conference on Neural Information Processing Systems (NeurIPS 2023).

a quadratic function of both the observation and the control exerted. The goal in LQR/LQG problems is to find a control policy $\pi$ in some policy class $\Pi$ that minimizes the cumulative cost over a finite time horizon $T$. With $\mathbf{y}_t^\pi, \mathbf{u}_t^\pi$ denoting the observation and control at time $t$ resulted from executing policy $\pi$, the objective is formally given by

$$\underset{\pi \in \Pi}{\text{minimize}} \quad J_T(\pi) \overset{\text{def}}{=} \sum_{t=1}^{T} c_t(\mathbf{y}_t^\pi, \mathbf{u}_t^\pi) = \sum_{t=1}^{T} \mathbf{y}_t^{\pi\top} Q_t \mathbf{y}_t^\pi + \mathbf{u}_t^{\pi\top} R_t \mathbf{u}_t^\pi,$$

where $Q_t, R_t$ are some possibly time-varying state-weighting and control-weighting matrices, respectively. In the elevator example, if our goal is to control the elevator to reach the desired floors while minimizing the discomfort and the energy consumption, the cost function at every time step $t$ may take the form $c_t(\mathbf{y}_t, \mathbf{u}_t) = c_1 \|\mathbf{y}_t - z_{\text{goal}}\|_2^2 + c_2 \|\mathbf{u}_t\|_2^2$, where $z_{\text{goal}}$ is the target position of the elevator, $c_1, c_2 > 0$.

Variations of the LQR/LQG problem have garnered considerable interest in the machine learning community. In the recent literature of **online nonstochastic control**, several setting-based generalizations to the aforementioned linear-control framework have been explored, including:

- **Adversarial cost functions** (Cohen et al. [2018], Agarwal et al. [2019a]): In this setting, the cost functions are adversarially chosen in an unpredictable way to the learner. This generalized formulation allows for applications such as building climate control in the presence of time-varying energy costs, as the energy costs may depend on unexpected demand fluctuation (Cohen et al. [2018]). More recently, the model of adversarially chosen cost functions, along with adversarial perturbations in the subsequent paragraph, give rise to building connection between control theory and meta optimization (Chen and Hazan [2023]), where the nonconvex problem of hyperparameter tuning (e.g. learning rate tuning) can be reduced to online nonstochastic control, which often admits a convex formulation.

- **Adversarial perturbations** (Agarwal et al. [2019a], etc.): In this setting, differing from the usual Gaussian assumption on the perturbation distribution ($\mathbf{w}_t \overset{i.i.d.}{\sim} N(0, \Sigma)$), nonstochastic control often considers arbitrary disturbances $\mathbf{w}_t$ in the dynamics. The relaxed assumption on the perturbations allows for the control of nonlinear systems under model inaccuracies by approximating them as linear systems (Ghai et al. [2022]).

- **Bandit feedback**: Together with the aforementioned two generalizations, control under bandit feedback further generalizes the control problem and suits many real-world applications. In the bandit setting, the cost functions is unknown to the learner. The learner has access *only* to the cost incurred by their chosen control. In addition, the learner has access to the system's state ($\mathbf{x}_t$ in the fully observed LQR setting) or a noisy, possibly and usually low-dimensional observation ($\mathbf{y}_t$ in the partially observed LQG setting). The bandit LQG problem can be illustrated with a motivating example. In the problem of controlling medical device such as a ventilator, the state usually represents a number of health status indicators of the patient, such as heart rate, blood pressure, and other related metrics. However, this state may not be fully observable. After the treatment is provided, the learner can observe the patient's response, e.g. changes in some vital indicators, with some observation noise and a cost function, e.g. deviation from the targeted breathing pattern. In this limited information setting, can a learner hope to learn meaning controls under minimal assumptions? Previously, the work of Gradu et al. [2020] and Cassel and Koren [2020] both studied the bandit nonstochastic control problem of fully observed systems and provided provable guarantees.

Taken together, these settings give rise to a general setting in differentiable reinforcement learning that strictly contains a variety of classical problems in optimal and robust control. Naturally, when adversarial costs and perturbations are considered, an optimal solution is *not* defined a priori. Instead, the primary performance metric is *regret*: the difference between the total cost of a control algorithm and that of the best controller from a specific policy class in hindsight.

As mentioned above, this general setting of bandit nonstochastic control was considered in the recent work of Gradu et al. [2020], whose proposed Bandit Perturbation Controller (BPC) algorithm has a provable regret guarantee of $\tilde{O}(T^{\frac{3}{4}})$ when compared to a rich policy class called disturbance action controllers (DAC) for fully observed systems. Similar setting has also been studied by Cassel and Koren [2020], who established an optimal regret up to logarithmic factor of $\tilde{O}(\sqrt{T})$ for fully

observable systems under stochastic perturbations and adversarially chosen cost functions. Recently Ghai et al. [2023] improves the dimension dependence of regret bounds using exploration in action-space rather than parameter-space. However, bandit control for partially observable systems (e.g. LQG) is less understood. Thus, these developments in the search for efficient, low-regret bandit online control algorithms leave a central open question (also stated by by Gradu et al. [2020]):

Can we achieve optimal regret $O(\sqrt{T})$ with **bandit LQG** and **nonstochastic noise**?

Our work answers this question up to logarithmic factors. Our novel Ellipsoidal Bandit Perturbation Controller (EBPC) achieves a $\tilde{O}(\sqrt{T})$ regret guarantee in the presence of semi-adversarial perturbations in bandit LQG problems with strongly convex cost functions, with the additional generality of possibly unknown system dynamics. By Shamir [2013], this is asymptotically optimal up to logarithmic factors, as bandit optimization over quadratics reduces to bandit control with quadratic losses under $A = 0, B = I$. Our work therefore resolves the upper-bound/lower-bound gap for this generalization of LQR/LQG. The following table gives a comprehensive comparison between the regret guarantee of EBPC and existing results in literature.

Table 1: Comparison of previous results to our contributions.

| Algorithm | Noise | Observation | Feedback | System | Regret |
|---|---|---|---|---|---|
| Agarwal et al. [2019a] | Adversarial | full | full | known | $\tilde{O}(\sqrt{T})$ |
| Agarwal et al. [2019b] | Stochastic | full | full | known | $\tilde{O}(1)$ |
| Foster and Simchowitz [2020] | Adversarial | full | full | known | $\tilde{O}(1)$ |
| Simchowitz et al. [2020] | Semi-Adv. | partial | full | known | $\tilde{O}(1)$ |
| Simchowitz et al. [2020] | Semi-Adv. | partial | full | unknown | $\tilde{O}(\sqrt{T})$ |
| Gradu et al. [2020] | Adversarial | full | bandit | unknown | $\tilde{O}(T^{\frac{3}{4}})$ |
| Cassel and Koren [2020] | Stochastic | full | bandit | known | $\tilde{O}(\sqrt{T})$ |
| Cassel and Koren [2020] | Adversarial | full | bandit | known | $\tilde{O}(T^{\frac{2}{3}})$ |
| **Theorem 4.1** | **Semi-Adv.** | **partial** | **bandit** | **known** | $\tilde{O}(\sqrt{T})$ |
| **Theorem 4.2** | **Semi-Adv.** | **partial** | **bandit** | **unknown** | $\tilde{O}(\sqrt{T})$ |

## 1.1 Related work

**Online Nonstochastic Control and Online LQR.** In the last decade, much research has been devoted to the intersection of learning and control. Abbasi-Yadkori and Szepesvári [2011] and Ibrahimi et al. [2012] considered the problem of learning a controller in LQR for known quadratic cost functions and stochastic/martingale difference perturbation sequence when the dynamics of the system is unknown, and achieved $\tilde{O}(\sqrt{T})$-regret in this case. Dean et al. [2018] provided the first provable low-regret, efficient algorithm to solve LQR problems with known cost functions and stochastic perturbations. Cohen et al. [2018] extended this result to changing quadratic costs with stochastic perturbations and provided a regret guarantee of $O(\sqrt{T})$. Lale et al. [2021] consider the LQG problem with stochastic noise and unknown systems.

More recently, interest has turned to **nonstochastic control**, in which the cost functions and the perturbations can be adversarially chosen Agarwal et al. [2019a]. A broad spectrum of control problems were reconsidered from the nonstochastic perspective, and several different generalizations were derived. See Hazan and Singh [2022] for a comprehensive text detailing these results.

**Online Bandit Convex Optimization with Memory.** A classical approach to nonstochastic control of stable/stabilizable linear dynamical systems is to reduce control problems to online convex optimization with bounded memory. This reduction stems from the observation that the effects of past controls decay exponentially in stable/stabilizable systems. In the bandit online convex optimization with bounded memory setting, the learner iteratively plays a decision $x_t$ in a convex set $\mathcal{K} \subseteq \mathbb{R}^d$ and suffers an adversarially chosen loss $F_t(x_{t-H+1:t})$, where $x_{t-H+1:t}$ is the sequence of points $x_{t-H+1}, ..., x_t$. In particular, the loss depends on the last $H$ points played by the algorithm, and the only information revealed to the learner is the scalar loss that they incurred. The goal is to minimize

*regret*, the difference between the loss actually suffered and the loss suffered under the best single play in hindsight:

$$\text{Regret}_T \overset{\text{def}}{=} \sum_{t=H}^{T} F_t(x_{t-H+1:t}) - \min_{x \in \mathcal{K}} \sum_{t=H}^{T} F_t(x, \ldots, x).$$

Since the loss function is unknown to the learner in the bandit setting, online bandit convex optimization algorithms make use of low-bias estimators of the true gradient or Hessian. Therefore, it is standard to measure the algorithm's performance by regret over the expectation of losses, where the expectation is taken over the randomness injected when creating such estimators.

Online convex optimization with memory in the full information setting, where the loss function is known to the learner, was proposed by Anava et al. [2015]. The work of Agarwal et al. [2019a], was the first to connect this to control, and to give a regret bound for online control with adversarial perturbations.

The $\tilde{O}(T^{\frac{3}{4}})$ regret bound of Gradu et al. [2020] leveraged the bandit convex optimization method of Flaxman et al. [2005]. We focus on the bandit LQR/LQG setting with strongly convex and smooth loss functions. It is thus natural to use the techniques of Hazan and Levy [2014], who obtained a $\tilde{O}(\sqrt{T})$ regret guarantee for bandit convex optimization without memory.

**Online Learning with Delay.**    One technical difficulty in extending OCO with memory to the bandit setting arises from the requirement of independence between every play and the noises injected from the recent $H$ steps. We resolve this issue by adapting online learning with delay to the subroutine algorithm used in our BCO algorithm. Online learning with delay was introduced by Quanrud and Khashabi [2015]. In particular, Flaspohler et al. [2021] relates online learning with delay to online learning with optimism and established a sublinear regret guarantee for mirror descent algorithms. A similar delay scheme was seen in [Gradu et al., 2020].

### 1.2    Notations and organization

**Notation.**    For convenience and succinctness, we denote $\bar{H} \overset{\text{def}}{=} H - 1$. We use lowercase bold letters (e.g. $\mathbf{x}, \mathbf{y}, \mathbf{u}, \mathbf{w}, \mathbf{e}$) to denote the states, observations, controls, and noises of the dynamical system [2], and $d_{\mathbf{x}}, d_{\mathbf{u}}, d_{\mathbf{y}}$ to denote their corresponding dimensions. We use $S^{n-1}$ to denote the unit sphere in $\mathbb{R}^n$, as $S^{n-1} \cong \mathbb{R}^{n-1}$. $\mathbb{R}_+, \mathbb{R}_{++}$ denote all non-negative, positive real numbers, respectively. For a differentiable function $F : (\mathbb{R}^n)^H \to \mathbb{R}$, we denote the gradient of $F$ with respect to its $i$th argument vector by $\nabla_i F(\cdot)$. $\rho(\cdot)$ acting on a square matrix measures the spectral radius of the matrix. For a sequence $M = (M^{[i]})_{i \in I}$, we use $\|M\|_{\ell_1, \text{op}}$ to denote the sum of the operator norm: $\|M\|_{\ell_1, \text{op}} \overset{\text{def}}{=} \sum_{i \in I} \|M^{[i]}\|_{\text{op}}$. $\sigma(A)$ denotes the $\sigma$-algebra generated by a random variable $A$. We use $O(\cdot)$ to hide all universal constants, $\tilde{O}(\cdot)$ to hide $\text{poly}(\log T)$ terms, and $\mathcal{O}(\cdot)$ to hide all natural parameters.

**Organization.**    Our method has two main components: a novel algorithm for BCO with memory (`EBCO-M`), and its application to building a novel bandit perturbation controller (`EBPC`). Section 2 describes our problem setting. Section 3 gives `EBCO-M` and its near-optimal regret guarantee. Section 4 introduces `EBPC` and its regret guarantees to both known and unknown systems.

## 2    The Bandit LQG Problem

In this section we provide necessary background and describe and formalize the main problem of interest. We consider control of linear time-invariant dynamical systems of the form

$$\mathbf{x}_{t+1} = A\mathbf{x}_t + B\mathbf{u}_t + \mathbf{w}_t , \quad \mathbf{y}_t = C\mathbf{x}_t + \mathbf{e}_t . \tag{2.1}$$

with dynamics matrices $A \in \mathbb{R}^{d_{\mathbf{x}} \times d_{\mathbf{x}}}, B \in \mathbb{R}^{d_{\mathbf{x}} \times d_{\mathbf{u}}}, C \in \mathbb{R}^{d_{\mathbf{y}} \times d_{\mathbf{x}}}$. Here, consistent with previous notations, $\mathbf{x}_t \in \mathbb{R}^{d_{\mathbf{x}}}$ is the state of the system at time $t$, $\mathbf{u}_t \in \mathbb{R}^{d_{\mathbf{u}}}$ is the control applied at time $t$, and

---

[2] It shall be noted that unbolded $x, y, u$ are used in the `EBCO-M` algorithm (Algorithm 1) and is unrelated to the linear dynamical system.

$\mathbf{w}_t \in \mathbb{R}^{d_\mathbf{x}}, \mathbf{e}_t \in \mathbb{R}^{d_\mathbf{y}}$ are the system and measurement perturbations. At each timestep, the learner may observe $\mathbf{y}_t \in \mathbb{R}^{d_\mathbf{y}}$, which usually represents a possibly noisy projection of the state $\mathbf{x}_t$ onto some (possibly and usually) low-dimensional space.

In the online bandit setting, the learner is asked to perform a control $\mathbf{u}_t$ at time $t$. After the control is performed, the adversary chooses a quadratic cost function $c_t(\mathbf{y}_t, \mathbf{u}_t)$. The learner observes the scalar $c_t(\mathbf{y}_t, \mathbf{u}_t) \in \mathbb{R}_+$ and the signal $\mathbf{y}_t$, but no additional information about $c_t(\cdot, \cdot)$, $\mathbf{x}_t$, $\mathbf{w}_t$, or $\mathbf{e}_t$ is given. The goal of the learner is to minimize *expected regret*, where *regret* against the controller class $\Pi$ is defined as

$$\text{Regret}_T \overset{\text{def}}{=} \sum_{t=1}^{T} c_t(\mathbf{y}_t, \mathbf{u}_t) - \min_{\pi \in \Pi} \sum_{t=1}^{T} c_t(\mathbf{y}_t^\pi, \mathbf{u}_t^\pi) , \tag{2.2}$$

where $\mathbf{u}_t^\pi$ is the control exerted at time $t$ by policy $\pi$, and $\mathbf{y}_t^\pi$ is the time-$t$ observation that would have occurred against the same costs/noises if the control policy $\pi$ were carried out from the beginning.

In controlling linear dynamical systems with partial observations, we often make use of the system's counterfactual signal had no controls been performed since the beginning of the instance:

**Definition 2.1** (Nature's $\mathbf{y}$). *Nature's $\mathbf{y}$ at time $t$, denoted by $\mathbf{y}_t^{\text{nat}}$, is the signal that the system would have generated at time $t$ under $\mathbf{u}_{1:t} = 0$. We may compute this as*

$$\mathbf{x}_{t+1}^{\text{nat}} = A\mathbf{x}_t^{\text{nat}} + \mathbf{w}_t , \quad \mathbf{y}_t^{\text{nat}} = C\mathbf{x}_t^{\text{nat}},$$

*or, equivalently, $\mathbf{y}_t^{\text{nat}} = \mathbf{e}_t + \sum_{i=1}^{t-1} CA^{t-i-1}\mathbf{w}_i$.*

Critically, this may be calculated via the Markov operator:

**Definition 2.2** (Markov operator). *The Markov operator $G = [G^{[i]}]_{i \in \mathbb{N}}$ corresponding to a linear system parametrized by $(A, B, C)$ as in Eq.(2.1) is a sequence of matrices in $\mathbb{R}^{d_\mathbf{y} \times d_\mathbf{u}}$ such that $G^{[i]} \overset{\text{def}}{=} CA^{i-1}B$, $G^{[0]} \overset{\text{def}}{=} \mathbf{0}_{d_\mathbf{y} \times d_\mathbf{u}}$.*

It follows immediately that $\mathbf{y}_t^{\text{nat}}$ may be computed from observations as $\mathbf{y}_t^{\text{nat}} = \mathbf{y}_t - \sum_{i=1}^{t} G^{[i]}\mathbf{u}_{t-i}$.

## 2.1 Assumptions

We impose four core assumptions on the problem:

**Assumption 2.3** (Stable system). *We assume the system is stable: the spectral radius $\rho(A) < 1$ [3].*

This assumption has the following important consequence:

**Remark 2.4** (Decay of stable systems). *That the system is stable implies that $\exists P \succ \mathbf{0}_{d_\mathbf{x} \times d_\mathbf{x}}$, $P \in \text{Sym}(d_\mathbf{x})$ such that $rP \succeq A^\top PA$ for some $0 \le r < 1$, and therefore $\exists \kappa$ depending on $\|B\|_{\text{op}}, \|C\|_{\text{op}}, \sigma_{\min}(P)$ such that $\|G^{[i]}\|_{\text{op}} \le \kappa r^{i-1}$. Then with $H = O(\log T)$, we can assume that $\|G\|_{\ell_1,\text{op}} = \sum_{i=0}^{\infty} \|G^{[i]}\|_{\text{op}} \le R_G$ and $\psi_G(H) \overset{\text{def}}{=} \sum_{i=H}^{\infty} \|G^{[i]}\|_{\text{op}} \le \frac{R_G}{T}$.*

Together, Assumption 2.3 and its consequence Remark 2.4 allows modeling the nonstochastic control problem as an online convex optimization problem against adversary with bounded memory, as the effect of past controls decays exponentially. Assumption 2.5-2.7 introduce the assumptions on the adversarial cost functions and perturbations.

**Assumption 2.5** (Noise model). *The perturbations $\{\mathbf{w}_t, \mathbf{e}_t\}_{t=1}^{T}$ are assumed to be semi-adversarial: $\mathbf{w}_t, \mathbf{e}_t$ decompose as sums of adversarial and stochastic components $\mathbf{w}_t = \mathbf{w}_t^{\text{adv}} + \mathbf{w}_t^{\text{stoch}}$ and $\mathbf{e}_t = \mathbf{e}_t^{\text{adv}} + \mathbf{e}_t^{\text{stoch}}$. The stochastic components of the perturbations are assumed to come from distributions satisfying $\mathbb{E}[\mathbf{w}_t^{\text{stoch}}] = \mathbb{E}[\mathbf{e}_t^{\text{stoch}}] = 0$, $\mathbb{E}\left[\mathbf{w}_t^{\text{stoch}}\mathbf{w}_t^{\text{stoch}\top}\right] \succeq \sigma_\mathbf{w}^2 I$, $\mathbb{E}\left[\mathbf{e}_t^{\text{stoch}}\mathbf{e}_t^{\text{stoch}\top}\right] \succeq \sigma_\mathbf{e}^2 I$, $\sigma_\mathbf{e} > 0$. $\{\mathbf{w}_t, \mathbf{e}_t\}_{t=1}^{T}$ are bounded such that $\|\mathbf{y}_t^{\text{nat}}\|_2 \le R_{\text{nat}}$, $\forall t$, for some parameter $R_{\text{nat}}$.*

---

[3]Although out of scope of this work, we refer the readers to Appendix G of Simchowitz et al. [2020] for a possible and expected extension of out results to the relaxed assumption of stability: the learner is given access to a stabilizing controller of the system.

The bound on $\mathbf{y^{nat}}$ is implied by bounded noise, which is a standard assumption in literature, and the stability of the system. The semi-adversarial assumption is also seen in prior work [Simchowitz et al., 2020], and is a necessary condition for our analysis: we depend on the regret guarantee of a bandit online convex optimization with memory algorithm which requires the strong convexity of the expected loss functions conditioned on all but the $\Theta(\text{poly}(\log T))$ most recent steps of history. This assumption is essentially equivalent to the adversarial assumption in applications: in almost all systems, noise is either endemic or may be injected. We also emphasize that this assumption is much weaker than that of previous optimal-rate work: even in the known-state, known-dynamic case, the previous optimal guarantee in the bandit setting depended on *no* adversarial perturbation (see Table 1).

**Assumption 2.6** (Cost model). *The cost functions $c_t(\cdot,\cdot)$ are assumed to be quadratic, $\sigma_c$-strongly convex, $\beta_c$-smooth, i.e. $c_t(\mathbf{y},\mathbf{u}) = \mathbf{y}^\top Q_t \mathbf{y} + \mathbf{u}^\top R_t \mathbf{u}$ with $\beta_c I \succeq Q_t \succeq \sigma_c I, \beta_c I \succeq R_t \succeq \sigma_c I \; \forall t$. They are also assumed to obey the following Lipschitz condition: $\forall (\mathbf{y},\mathbf{u}), (\mathbf{y}',\mathbf{u}') \in \mathbb{R}^{d_\mathbf{y}+d_\mathbf{u}}$,*

$$|c_t(\mathbf{y},\mathbf{u}) - c_t(\mathbf{y}',\mathbf{u}')| \le L_c(\|(\mathbf{y},\mathbf{u})\|_2 \vee \|(\mathbf{y}',\mathbf{u}')\|_2)\|(\mathbf{y}-\mathbf{y}',\mathbf{u}-\mathbf{u}')\|_2. \tag{2.3}$$

These conditions are relatively standard for bandit convex optimization algorithms, and are needed for the novel BCO-with-memory algorithm which underpins our control algorithm.

**Assumption 2.7** (Adversary). $\{c_t(\cdot,\cdot), \mathbf{w}_t^{\text{adv}}, \mathbf{e}_t^{\text{adv}}\}_{t=1}^T$ *is chosen by the adversary ahead of time.*

## 2.2 Disturbance Response Controllers

Regret compares the excess cost from executing our proposed control algorithm with respect to the cost of the best algorithm in hindsight from a given policy class. In particular, low regret against a rich policy class is a very strong near-optimality guarantee. We take the comparator policy class $\Pi$ to be the set of disturbance response controllers (DRC), formally given by the following definition.

**Definition 2.8** (Disturbance Response Controllers). *The disturbance response controller (DRC) and the DRC policy class are defined as:*

- *A **disturbance response controller (DRC)** is a policy $\pi_M$ parametrized by $M = (M^{[j]})_{j=0}^{\bar{H}}$, where $H \in \mathbb{Z}_{++}$, a sequence of $H$ matrices in $\mathbb{R}^{d_\mathbf{u} \times d_\mathbf{y}}$ s.t. the control at time $t$ given by $\pi_M$ is $\mathbf{u}_t^{\pi_M} = \sum_{j=0}^{\bar{H}} M^{[j]} \mathbf{y}_{t-j}^{\mathbf{nat}}$. We shorthand $\mathbf{u}_t^M \stackrel{\text{def}}{=} \mathbf{u}_t^{\pi_M}$.*

- *The **DRC policy class** parametrized by $H \in \mathbb{Z}_{++}$, $R \in \mathbb{R}_+$ is the set of all disturbance response controller with bounded length $H$ and norm $R$: $\mathcal{M}(H,R) = \{M = (M^{[j]})_{j=0}^{\bar{H}} \mid \|M\|_{\ell_1,\text{op}} = \sum_{j=0}^{\bar{H}} \|M^{[j]}\|_{\text{op}} \le R\}$.*

Previous works have demonstrated the richness of the DRC policy class. In particular, Theorem 1 from Simchowitz et al. [2020] has established that the DRC policy class generalizes the state-of-art benchmark class of stabilizing linear dynamic controllers (LDC) with error $e^{-\Theta(H)}$.

## 2.3 Approach and Technical Challenges

The classical approach in online nonstochastic control of stable/stabilizable systems is to reduce to a problem of online convex optimization with memory. This insight relies on the exponentially decaying effect of past states and controls on the present, which allows approximating the cost functions as functions of the most recent controls.

A core technical challenge lies in the bandit convex optimization problem obtained from the bandit control problem. In the bandit setting, no gradient information is given to the learner, and thus the learner needs to construct a low-bias gradient estimator. Previous work uses the classical spherical gradient estimator proposed by Flaxman et al. [2004], but the regret guarantee is suboptimal. We would like to leverage the ellipsoidal gradient estimator proposed by Hazan and Levy [2014]. However, when extending to loss functions with memory, there is no clear mechanism for obtaining a low-bias bound for general convex functions. We exploit the quadratic structure of the LQR/LQG cost functions to build `EBCO-M` (Algorithm 1), which uses ellipsoidal gradient estimators. We note that even outside of the control applications, `EBCO-M` may be of independent interests in bandit online learning theory.

# 3 BCO with Memory: Quadratic and Strongly Convex Functions

As with previous works, our control algorithm will depend crucially on a generic algorithm for bandit convex optimization with memory (BCO-M). We present a new online bandit convex optimization with memory algorithm that explores the structure of quadratic costs to achieve near-optimal regret.

## 3.1 Setting and working assumptions

In the BCO-M setting with memory length $H$, we consider an algorithm playing against an adversary. At time $t$, the algorithm is asked to play its choice of $y_t$ in the convex constraint set $\mathcal{K}$. The adversary chooses a loss function $F_t : \mathcal{K}^H \to \mathbb{R}_+$ which takes as input the algorithm's current play as well as its previous $\bar{H}$ plays. The algorithm then observes a cost $F_t(y_{t-\bar{H}}, \ldots, y_t)$ (and no other information about $F_t(\cdot)$) before it chooses and plays the next action $y_{t+1}$. The goal is to minimize regret with respect to the expected loss, which is the excessive loss incurred by the algorithm compared to the best fixed decision in $\mathcal{K}$:

$$\text{Regret}_T \overset{\text{def}}{=} \sum_{t=H}^{T} \mathbb{E}[F_t(y_{t-\bar{H}}, \ldots, y_t)] - \min_{x \in \mathcal{K}} \sum_{t=H}^{T} \mathbb{E}[F_t(x, \ldots, x)].$$

For notation convenience, we will at times shorthand $y_{t-\bar{H}:t} \overset{\text{def}}{=} (y_{t-\bar{H}}, \ldots, y_t) \in \mathcal{K}^H$.

### 3.1.1 BCO-M assumptions

We make the following assumptions on the loss functions $\{F_t\}_{t=H}^{T}$ and the constraint set $\mathcal{K}$.

**Assumption 3.1** (Constraint set). *$\mathcal{K}$ is convex, closed, and bounded with non-empty interior. $diam(\mathcal{K}) = \sup\limits_{z, z' \in \mathcal{K}} \|z - z'\|_2 \le D$.*

**Assumption 3.2** (Loss functions). *The loss functions chosen by the adversary obeys the following regularity and curvature assumptions:*

- *$F_t : \mathcal{K}^H \to R_+$ is quadratic and $\beta$-smooth:*

  - *Quadratic: $\exists W_t \in \mathbb{R}^{nH \times nH}, b_t \in \mathbb{R}^{nH}, c_t \in \mathbb{R}$ such that $F_t(w) = w^\top W_t w + b_t^\top w + c_t, \forall w \in \mathcal{K}^H$.*
  - *Smooth: $W_t \preceq \beta I_{nH \times nH}$.*

- *$F_t : \mathcal{K}^H \to \mathbb{R}_+$ is $\sigma$-strongly convex in its induced unary form: $f_t : \mathcal{K} \to \mathbb{R}_+$ with $f_t(z) = F_t(z, \ldots, z)$ is $\sigma$-strongly convex, i.e. $f_t(z) \ge f_t(z') + \nabla f_t(z')^\top (z - z') + \frac{\sigma}{2}\|z - z'\|_2^2$, $\forall z, z' \in \mathcal{K}$.*

- *$F_t$ satisfies the following diameter and gradient bound on $\mathcal{K}$: $\exists B, L > 0$ such that*

$$B = \sup_{w, w' \in \mathcal{K}^H} |F_t(w) - F_t(w')|, \quad L = \sup_{w \in \mathcal{K}^H} \|\nabla F_t(w)\|_2.$$

In the online control problems, when formulating the cost function $c_t$ as a function $F_t$ of the most recent $H$ controls played, the function $F_t$ itself may depend on the entire history of the algorithm through step $t - H$. Therefore, it is essential to analyze the regret guarantee of our BCO-M algorithm when playing against an adversary that can be $(t - H)$-adaptive, giving rise to the following assumption.

**Assumption 3.3** (Adversarial adaptivity). *The adversary chooses $F_t$ independently of the noise $u_{t-\bar{H}:t}$ which is drawn by the algorithm in the $H$ most recent steps, but possibly not independently of earlier noises.*

Note that Assumption 3.3 is minimal for BCO: if this fails, then in the subcase of a delayed loss, the adversary may fully control the agent's observations, resulting in no possibility of learning.

**Self-concordant barrier.** The algorithm makes use of a *self-concordant barrier* $R(\cdot)$ of $\mathcal{K}$ as the regularization function in the updates.

**Definition 3.4** (Self-concordant barrier). *A three-time continuously differentiable function $R(\cdot)$ over a closed convex set $\mathcal{K} \subset \mathbb{R}^n$ with non-empty interior is a $\nu$-self-concordant barrier of $\mathcal{K}$ if it satisfies the following two properties:*

1. *(Boundary property) For any sequence $\{x_n\}_{n \in \mathbb{N}} \subset \text{int}(\mathcal{K})$ such that $\lim_{n \to \infty} x_n = x \in \partial\mathcal{K}$, $\lim_{n \to \infty} R(x_n) = \infty$.*

2. *(Self-concordant) $\forall x \in \text{int}(\mathcal{K})$, $h \in \mathbb{R}^n$,*

   (a) $|\nabla^3 R(x)[h, h, h]| \leq 2|\nabla^2 R(x)[h, h]|^{3/2}$.
   (b) $|\langle \nabla R(x), h \rangle| \leq \sqrt{\nu}|\nabla^2 R(x)[h, h]|^{1/2}$.

### 3.2 Algorithm specification and regret guarantee

We present `EBCO-M` (Algorithm 1) for online bandit convex optimization with memory. The key novelty is the use of an ellipsoidal gradient estimator. It is difficult to establish a low-bias guarantee for ellipsoidal gradient estimator for general convex loss functions. However, thanks to the quadratic structure of the loss functions in LQR/LQG problems, we can show provable low bias for the ellipsoidal gradient estimator, and therefore achieve optimal regret.

---

**Algorithm 1** Ellipsoidal BCO with memory (`EBCO-M`)

---

1: Input: Convex, closed set $\mathcal{K} \subseteq \mathbb{R}^n$ with non-empty interior, time horizon $T$, memory length $H$, step size $\eta$, $\nu$-self-concordant barrier $R(\cdot)$ over $\mathcal{K}$, convexity strength parameter $\sigma$.
2: Initialize $x_t = \arg\min_{x \in \mathcal{K}} R(x)$, $\forall t = 1, \ldots, H$.
3: Compute $A_t = (\nabla^2 R(x_t) + \eta\sigma t I)^{-1/2}$, $\forall t = 1, \ldots, H$.
4: Sample $u_1, \ldots, u_H \sim S^{n-1}$ i.i.d. uniformly at random.
5: Set $y_t = x_t + A_t u_t$, $\forall t = 1, \ldots, H$.
6: Set $g_t = 0$, $\forall t = 1, \ldots, \bar{H}$.
7: Play $y_1, \ldots, y_{\bar{H}}$.
8: **for** $t = H, \ldots, T$ **do**
9:     Play $y_t$, suffer loss $F_t(y_{t-\bar{H}:t})$.
10:     Store $g_t = nF_t(y_{t-\bar{H}:t}) \sum_{i=0}^{\bar{H}} A_{t-i}^{-1} u_{t-i}$.
11:     Set $x_{t+1} = \arg\min_{x \in \mathcal{K}} \sum_{s=H}^{t} \left( g_{s-\bar{H}}^\top x + \frac{\sigma}{2} \|x - x_{s-\bar{H}}\|^2 \right) + \frac{1}{\eta} R(x)$.
12:     Compute $A_{t+1} = (\nabla^2 R(x_{t+1}) + \eta\sigma(t+1)I)^{-1/2}$.
13:     Sample $u_{t+1} \sim S^{n-1}$ uniformly at random.
14:     Set $y_{t+1} = x_{t+1} + A_{t+1} u_{t+1}$.
15: **end for**

---

Before analyzing the regret, we first make note of two properties of Algorithm 1.

**Remark 3.5** (Delayed dependence). *In Algorithm 1, $x_t$ is independent of $u_{t-\bar{H}:t}$, $\forall t$, and therefore $A_t$ is independent of $u_{t-\bar{H}:t}$ as $A_t$ is determined by $x_t$.*

**Remark 3.6** (Correctness). *$y_t$ played by Algorithm 1 lies in $\mathcal{K}$: $\|y_t - x_t\|^2_{\nabla^2 R(x_t)} = \|A_t u_t\|^2_{\nabla^2 R(x_t)} \leq \|u_t\|_2^2 = 1$, and by Proposition C.1, the Dikin ellipsoid centered at $x_t$ is contained in $\mathcal{K}$.*

We give the following two theorems on regret guarantees of the `EBCO-M` algorithm. In particular, Theorem 3.6.B is a slight generalization of Theorem 3.6.A in its relaxation on the curvature assumption on the sequence of loss functions $F_{H:T}$.

**Theorem 3.6.A** (`EBCO-M` regret with strong convexity). *For any sequence of cost functions $\{F_t\}_{t=H}^T$ satisfying Assumption 3.2, constraint set $\mathcal{K}$ satisfying Assumption 3.1, adversary satisfying Assumption 3.3, and $H = \text{poly}(\log T)$, Algorithm 1 satisfies the regret bound*

$$Regret_T(\textit{EBCO-M}) = \sum_{t=H}^{T} \mathbb{E}[F_t(y_{t-\bar{H}:t})] - \min_{x \in \mathcal{K}} \sum_{t=H}^{T} \mathbb{E}[f_t(x)] \leq \tilde{\mathcal{O}}\left(\frac{\beta n}{\sigma}\sqrt{T}\right),$$

*where the expectation is taken over the randomness of the exploration noises $u_{1:T}$, with $\tilde{\mathcal{O}}$ hiding all natural parameters $(B, D, L)$ and logarithmic dependence on $T$.*

**Theorem 3.6.B** (EBCO-M regret with conditional strong convexity). *Suppose Algorithm 1 is run on $\mathcal{K}$ satisfying Assumption 3.1 against an adversary satisfying Assumption 3.3 with a sequence of random cost functions $\{F_t\}_{t=H}^T$ satisfying that*

1. *$F_t$ is quadratic, convex, $\beta$-smooth, has diameter bound $B$ and gradient bound $L$ on $\mathcal{K}^H$.*

2. *$\exists$ filtration $\mathcal{G}_{1:T}$ such that $u_{1:T}$ is independent of $\mathcal{G}_T$, $F_t \in \sigma(\mathbf{u}_{1:t-H}) \cup \mathcal{G}_t$, and $F_t$ is conditionally $\sigma$-strongly convex in its induced unary form: $\bar{f}_t(z) \overset{def}{=} \mathbb{E}[f_t(z) \mid u_{1:t-H}, \mathcal{G}_{t-H}]$ is $\sigma$-strongly convex.*

*Then, Algorithm 1 satisfies the same regret bound attained in Theorem 3.6.A, i.e.*

$$Regret_T(\text{EBCO-M}) = \sum_{t=H}^T \mathbb{E}[F_t(y_{t-\bar{H}:t})] - \min_{x \in \mathcal{K}} \sum_{t=H}^T \mathbb{E}[f_t(x)] \leq \tilde{\mathcal{O}}\left(\frac{\beta n}{\sigma}\sqrt{T}\right),$$

*where the expectation is taken over the randomness of the exploration noises $u_{1:T}$ and the random functions $F_{H:T}$, with $\tilde{\mathcal{O}}$ hiding all natural parameters $(B, D, L)$ and logarithmic dependence on $T$.*

## 4 Bandit Controller: Known and Unknown Systems

We will now use our BCO-with-memory algorithm to find an optimal controller (as in Gradu et al. [2020]), arguing that regret in choice of controller transfers into the setting discussed in the previous section. We first consider the case where the system is known, and then reduce the unknown system case to the known system case.

### 4.1 Known systems

Applying Algorithm 1 to predict controllers[4] with losses given by control losses, we obtain Algorithm 2.

---

**Algorithm 2** Ellipsoidal Bandit Perturbation Controller (EBPC)

---

1: Input: Time horizon $T$, memory length $H$, Markov operator $G$. BCO-M parameters $\sigma, \eta$. Self-concordant barrier $R(\cdot)$ over $\mathcal{M}(H, R) \subset \mathbb{R}^{H \times d_{\mathbf{u}} \times d_{\mathbf{y}}}$.

2: Initialize $M_1 = \cdots = M_H = \underset{M \in \mathcal{M}(H,R)}{\arg\min} R(M)$.

3: Compute $A_i = (\nabla^2 R(M_i) + \eta\sigma t I)^{-1/2}, \forall i = 1, \ldots, H$.

4: Sample $\varepsilon_1, \ldots, \varepsilon_H \sim S^{H \times d_{\mathbf{u}} \times d_{\mathbf{y}} - 1}$ i.i.d. uniformly at random.

5: Set $\widetilde{M}_i = M_i + \varepsilon_i, \forall i = 1, \ldots, H$. Set $g_i = 0, \forall i = 1, \ldots, \bar{H}$.

6: Play control $\mathbf{u}_i = 0$, incur cost $c_i(\mathbf{y}_i, \mathbf{u}_i), \forall i = 1, \ldots, \bar{H}$.

7: **for** $t = H, \ldots, T$ **do**

8:     Play control $\mathbf{u}_t = \mathbf{u}_t(\widetilde{M}_t) = \sum_{i=0}^{\bar{H}} \widetilde{M}_t^{[i]} \mathbf{y}_{t-i}^{\mathbf{nat}}$, incur cost $c_t(\mathbf{y}_t, \mathbf{u}_t)$.

9:     Observe $\mathbf{y}_{t+1}$ and compute signal $\mathbf{y}_{t+1}^{\mathbf{nat}} = \mathbf{y}_{t+1} - \sum_{i=1}^{t} G^{[i]} \mathbf{u}_{t-i}$.

10:    Store $g_t = d_{\mathbf{u}} d_{\mathbf{y}} H c_t(\mathbf{y}_t, \mathbf{u}_t) \sum_{i=0}^{\bar{H}} A_{t-i}^{-1} \varepsilon_{t-i}$.

11:    Update $M_{t+1} = \underset{M \in \mathcal{M}(H,R)}{\arg\min} \sum_{s=H}^{t} \left(\langle g_{s-\bar{H}}, M\rangle + \frac{\sigma}{2}\|M - M_{s-\bar{H}}\|^2\right) + \frac{1}{\eta}R(M)$.

12:    Compute $A_{t+1} = (\nabla^2 R(M_{t+1}) + \eta\sigma(t+1)I)^{-1/2}$.

13:    Sample $\varepsilon_{t+1} \sim S^{H \times d_{\mathbf{u}} \times d_{\mathbf{y}} - 1}$ uniformly at random. Set $\widetilde{M}_{t+1} = M_{t+1} + A_{t+1}\varepsilon_{t+1}$.

14: **end for**

---

**Theorem 4.1** (Known system control regret). *Consider a linear dynamical system governed by known dynamics $(A, B, C)$ and the interaction model with adversarially chosen cost functions*

---

[4]Notation: while our controller $M$ is typically a tensor, it should be thought of as the output vector of Algorithm 1. As such, the relevant vector and matrix operations in that algorithm will correspond to tensor operations here, and the notation reflects that correspondence. In particular, the inner product on line 11 is an all-dimension tensor dot product and $A$ is a square "matrix" which acts on tensors of shape $(H, d_{\mathbf{u}}, d_{\mathbf{y}})$.

*and perturbations satisfying Assumption 2.3, 2.5, 2.6, 2.7. Then running Algorithm 2 with $H = \Theta(\text{poly}(\log T))$, $\sigma = \sigma_c \left( \sigma_{\mathbf{e}}^2 + \sigma_{\mathbf{w}} \frac{\sigma_{\min}(C)}{1 + \|A\|_{\mathrm{op}}^2} \right)$, and $\eta = \Theta \left( \frac{1}{d_{\mathbf{u}} d_{\mathbf{y}} L_c H^3 \sqrt{T}} \right)$ guarantees*

$$\mathbb{E}[Regret_T(EBPC)] \leq \tilde{\mathcal{O}} \left( \frac{\beta_c d_{\mathbf{u}} d_{\mathbf{y}}}{\sigma_c} \sqrt{T} \right),$$

*where regret is defined as in Eq.(2.2), the expectation is taken over the exploration noises $\varepsilon_{1:T}$ of the algorithm as well as the stochastic components $\{\mathbf{w}_t^{\mathrm{stoch}}, \mathbf{e}_t^{\mathrm{stoch}}\}_{t=1}^T$ of the perturbations $\{\mathbf{w}_t, \mathbf{e}_t\}_{t=1}^T$, and $\tilde{\mathcal{O}}(\cdot)$ hides all universal constants, natural parameters, and logarithmic dependence on $T$.*

### 4.2 Unknown systems: control after estimation

Note that EBPC (Algorithm 2) relies on the access to the system's Markov operator $G$, which is available if and only if the system dynamics $(A, B, C)$ are known. When the system dynamics is unknown, we can identify the system using a system estimation algorithm, obtain an estimated Markov operator $\hat{G}$, and run EBPC with $G \leftarrow \hat{G}$. Algorithm 3 outlines the estimation method of system dynamics via least squares.

---

**Algorithm 3** System estimation via least squares (SysEst-LS)

---

1: Input: estimation sample size $N$, system length $H$.
2: Initialize: $\hat{G}^{[t]} = 0, \forall t \geq H$.
3: Sample and play $\mathbf{u}_t \sim N(0, I_{d_{\mathbf{u}} \times d_{\mathbf{u}}}), \forall t = 1, \dots, N$.
4: Set $\hat{G}^{[0:\bar{H}]} = \arg\min \sum_{t=H}^N \|\mathbf{y}_t - \sum_{i=0}^{\bar{H}} \hat{G}^{[i]} \mathbf{u}_{t-i}\|_2^2$.
5: Return $\hat{G}$.

---

**Theorem 4.2** (Unknown system control regret). *Consider a linear dynamical system governed by unknown dynamics $(A, B, C)$ and the interaction model with adversarially chosen cost functions and perturbations satisfying Assumption 2.3, 2.5, 2.6, 2.7. Suppose we obtain an estimated Markov operator $\hat{G}$ from Algorithm 3 with $N = \lceil \sqrt{T} \rceil$ and $H = \Theta(\text{poly}\log T)$. Then Algorithm 2 with $G \leftarrow \hat{G}$, $H \leftarrow 3H$, $\sigma = \frac{1}{8} \sigma_c \sigma_{\mathbf{e}}^2$, and $\eta = \Theta \left( \frac{1}{d_{\mathbf{u}} d_{\mathbf{y}} L_c H^3 \sqrt{T}} \right)$ guarantees*

$$\mathbb{E}[Regret_T(EBPC)] \leq \tilde{\mathcal{O}} \left( \frac{\beta_c d_{\mathbf{u}} d_{\mathbf{y}}}{\sigma_c} \sqrt{T} \right),$$

*where regret is defined as in Eq.(2.2), the expectation is taken over the exploration noises $\varepsilon_{1:T}$ in Algorithm 2, the sampled Gaussian controls $\mathbf{u}_{1:N}$ in Algorithm 3, and the stochastic components $\{\mathbf{w}_t^{\mathrm{stoch}}, \mathbf{e}_t^{\mathrm{stoch}}\}_{t=1}^T$ of the perturbations $\{\mathbf{w}_t, \mathbf{e}_t\}_{t=1}^T$, and $\tilde{\mathcal{O}}(\cdot)$ hides all universal constants, natural parameters, and logarithmic dependence on $T$.*

## 5 Discussion and conclusion

We solve the open problem put forth by Gradu et al. [2020] on the optimal rate for online bandit control for the case of LQR/LQG control, improving to regret $\tilde{O}(\sqrt{T})$ from $\tilde{O}(T^{\frac{3}{4}})$ in the semi-adversarial noise model and for strongly convex LQR/LQG cost functions. Our method builds upon recent advancements in bandit convex optimization for quadratic functions, providing the first near-optimal regret algorithm for bandit convex optimization with memory in a nonstochastic setting.

It would be interesting to investigate (1) whether the results can be extended to fully adversarial noise, (2) whether a similar stable controller recovery as seen in [Chen and Hazan, 2021] for fully observable systems can be established for partially observable systems, and whether that can be incorporated to extend our result to stabilizable systems even without access to a stabilizing controller.

## 6 Acknowledgement

Elad Hazan acknowledges funding from the Office of Naval Research grant N000142312156, the NSF award 2134040, and Open Philanthropy.

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
