# Contents

# A  Notations and Organization

## A.1  Organization

### A.1.1  Appendix B: Experiments

Appendix B provides brief emprical results in a standard control problem with a few classic perturbation patterns, and compares to classical LQR control and the more advanced control of Gradu et al. [2020].

### A.1.2  Appendix C: Proof of `EBCO-M` regret guarantee

Appendix C proves regret guarantee (Theorem 3.6.B) for `EBCO-M` (Algorithm 1) under Assumption 3.1, a relaxed 3.2, and 3.3:

- Section C.1: properties of self-concordant barriers used in the proof
- Section C.2: conditional bias guarantee for the proposed gradient estimator in Algorithm 1
- Section C.3: regret analysis for Algorithm 1

### A.1.3  Appendix D: Proof of `EBPC` Regret Guarantee for Known Systems

Appendix D proves `EBPC` regret guarantee for known systems as stated in Theorem 4.1:

- Section D.1: construction of with-history loss functions based on cost functions
- Section D.2: establishes the following regularity conditions for with-history loss functions
  - Construction of with-history functions and unary forms: Definition D.2.
  - Norm bound on $\mathbf{y}_t, \mathbf{u}_t$: Lemma D.4
  - Diameter bound $B$ of $c_t$ and $F_t$: Lemma D.5
  - Diameter bound $D$ of $\mathcal{M}(H, R)$: Lemma D.5
  - Lipschitz bound $L_F$ of $F_t$: Lemma D.6
  - Conditional strong convexity parameter $\sigma_f$ of $f_t$: Lemma D.6
  - Smoothness parameter $\beta_F$ of $F_t$: Lemma D.6
- Section D.3: `EBPC` regret analysis for known systems

### A.1.4  Appendix E: Proof of `EBPC` Regret Guarantee for Unknown Systems

Appendix E proves `EBPC` regret guarantee for unknown systems as stated in Theorem 4.2:

- Section E.1: system estimation error guarantee
- Section E.2: construction of with-history loss functions and pseudo loss functions
- Section E.3: regret guarantee for Regularized Follow-the-Leader with Delay (RFTL-D) with erroneous gradients
- Section E.4: regularity conditions for pseudo-loss and with-history loss functions:
  - Construction of with-history functions, pseudo loss functions, and unary forms: Definition E.5, E.6
  - Norm bound on $\hat{\mathbf{y}}_t^{\mathbf{nat}}, \mathbf{y}_t, \mathbf{u}_t$: Lemma E.12
  - Diameter bound $B$ of $c_t$ and $\hat{F}_t, \mathring{F}_t$: Lemma E.13
  - Diameter bound $D$ of $\mathcal{M}(H^+, R^+)$: Lemma E.13
  - Lipschitz bound $L_{\mathring{F}}$ and $L_{\hat{F}}$ for $\mathring{F}_t$ and $\hat{F}_t$: Lemma E.14
  - Smoothness parameter $\beta_{\mathring{F}}$ and $\beta_{\hat{F}}$ for $\mathring{F}_t$ and $\hat{F}_t$: Lemma E.14
  - Conditional strong convexity parameter $\sigma_{\mathring{f}}$ and $\sigma_{\hat{f}}$ for $\mathring{f}_t$ and $\hat{f}_t$: Lemma E.14
- Section E.5: `EBPC` regret analysis for unknown systems

## A.2 Complete List of Notations

- **Asymptotic equivalence.** We use $\lesssim, \gtrsim, \asymp$, or equivalently, $O(\cdot), \Omega(\cdot), \Theta(\cdot)$, to denote asymptotic inequalities and equivalence. In particular, $a \lesssim b$ ($a = O(b)$), $a \gtrsim b$ if $\exists$ universal constant $c$ such that $a \leq cb$, $a \geq cb$, respectively. $a \asymp b$ if $a \lesssim b$ and $a \gtrsim b$.

- **Derivative.** For $f : \mathbb{R}^m \to \mathbb{R}^n$, we use $\mathbf{D}f \in \mathbb{R}^{m \times n}$ to denote its derivative.

- **Spectral radius.** For $A \in \mathbb{R}^{n \times n}$, $\rho(A)$ measures $A$'s spectral radius, or maximum of the absolute values of $A$'s eigenvalues.

- **Norms.**

| Notation | Meaning | Domain* | Definition |
|---|---|---|---|
| $\|\cdot\|_p$ | $\ell_p$-norm | $\mathbb{R}^n$ | $v \mapsto (\sum_{i=1}^n v_i^p)^{\frac{1}{p}}$ |
| $\|\cdot\|_F$ | Frobenius norm | $\mathbb{R}^{m \times n \times r}$ | $M \mapsto (\sum_{i=1}^m \sum_{j=1}^n \sum_{k=1}^r M_{ijk}^2)^{\frac{1}{2}}$ |
| $\|\cdot\|_{\mathrm{op}}$ | operator norm | $\mathbb{R}^{m \times n}$ | $M \mapsto \sup_{v \in \mathbb{R}^n, \|v\|_2=1} \|Mv\|_2$ |
| $\|\cdot\|_M,\ M \in \mathbb{R}^{n \times n}$ | local norm induced by $M$ | $\mathbb{R}^n$ | $v \mapsto (v^\top M v)^{\frac{1}{2}}$ |
| $\|\cdot\|_{\ell_1,\mathrm{op}}$ | $\ell_1$-operator norm | $(\mathbb{R}^{m \times n})^{\mathbb{N}}$ | $(M_i)_{i \in I \subseteq \mathbb{N}} \mapsto \sum_{i \in I} \|M_i\|_{\mathrm{op}}$ |
| $\|\cdot\|^*$ | dual norm of $\|\cdot\|$ | same as $\|\cdot\|$ | $v \mapsto \sup\{\langle u, v \rangle : \|u\| \leq 1\}$ |
| $\|\cdot\|_t, \|\cdot\|_{t,t+1}$ | local norm at time $t$ | $\mathbb{R}^n$ | see Definition C.4 |

  * : $m, n, r$ are arbitrary dimensions that may be specifically defined throughout the paper.

- **System, dynamics, and parameters.**

| | |
|---|---|
| $d_\mathbf{x}, d_\mathbf{u}, d_\mathbf{y}$ | dimension of states, controls, observations |
| $A, B, C$ | system matrices for linear dynamical system |
| $G$ | Markov operator for linear dynamical system |
| $\hat{G}$ | estimated Markov operator |
| $\mathbf{x}_t \in \mathbb{R}^{d_\mathbf{x}}$ | state at time $t$ |
| $\mathbf{u}_t \in \mathbb{R}^{d_\mathbf{u}}$ | control at time $t$ |
| $\mathbf{w}_t \in \mathbb{R}^{d_\mathbf{x}}$ | system perturbation (disturbance) at time $t$ |
| $\mathbf{e}_t \in \mathbb{R}^{d_\mathbf{y}}$ | state-observation projection noise at time $t$ |
| $\mathbf{y}_t \in \mathbb{R}^{d_\mathbf{y}}$ | observation at time $t$ |
| $\mathbf{y}_t^{\mathbf{nat}} \in \mathbb{R}^{d_\mathbf{y}}$ | nature's $\mathbf{y}$, the would-be observation at time $t$ assuming no controls are ever played |
| $\hat{\mathbf{y}}_t^{\mathbf{nat}} \in \mathbb{R}^{d_\mathbf{y}}$ | algorithm calculated nature's $\mathbf{y}$ using the estimated Markov operator $\hat{G}$ |
| $H, \bar{H}, H^+, \overline{H^+}$ | history length of a policy class, $\bar{H} = H - 1$, $H^+ = 3H$, $\overline{H^+} = H^+ - 1$ |
| $R, R^+$ | DRC policy class $\ell_1$-operator norm bound, $R^+ = 2R$ |
| $R_{\mathrm{nat}}$ | nature's $\mathbf{y}$ $\ell_2$-norm bound |
| $R_G$ | $\ell_1$-operator norm bound on $G$ |
| $\mathcal{M}(H, R)$ | DRC policy class with length $H$ and $\ell_1$-operator norm bound $R$ |

- **Cost and loss functions.**

| Notation | Meaning | Domain |
|---|---|---|
| $c_t(\cdot, \cdot)$ | cost function for controlling linear dynamical system | $\mathbb{R}^{d_\mathbf{y}} \times \mathbb{R}^{d_\mathbf{u}}$ |
| $F_t(\cdot), \hat{F}_t(\cdot)$ | with-history loss function with history length $H$ | $\mathcal{K}^H$ for convex Euclidean set $\mathcal{K}$ |
| $f_t(\cdot), \hat{f}_t(\cdot)$ | unary form induced by $f_t(x) = F_t(x, \dots, x)$ | some convex Euclidean set $\mathcal{K}$ |
| $\mathring{F}_t(\cdot), \mathring{f}_t(\cdot)$ | pseudo-loss and induced unary form | $\mathcal{K}^H, \mathcal{K}$ for convex Euclidean set $\mathcal{K}$ |
| $B$ | bound on function diameter | |
| $D$ | bound on constraint set diameter | |
| $L_c, L_F, L_{\mathring{F}}, L_{\hat{F}}$ | Lipschitz bound on function $c_t, F_t, \mathring{F}_t, \hat{F}_t$ | |
| $\beta_c, \beta_F, \beta_{\mathring{F}}, \beta_{\hat{F}}$ | smoothness parameter of $c_t, F_t, \mathring{F}_t, \hat{F}_t$ | |
| $\sigma_c$ | strong convexity parameter of $c_t$ | |
| $\sigma_f, \sigma_{\mathring{f}}, \sigma_{\hat{f}}$ | conditional strong convexity parameter of $f_t, \mathring{f}_t, \hat{f}_t$ | |

# B    Experiments

To compare our controller against previous work, we test our control scheme empirically in the same settings as Gradu et al. [2020]. Our experiments use the package Deluca developed by Gradu et al. [2021]. We test control of a barely-stable LDS – a damped double-integrator system given by

$$A = \begin{bmatrix} .9 & .9 \\ -0.01 & .9 \end{bmatrix}, B = \begin{bmatrix} 0 \\ 1 \end{bmatrix}$$

We attempt control under several different classes of noise. Relevant details are below:

- As the controller of Gradu et al. [2020] does not support partial observation, we test in the full-observation case.
- Both controllers are given access to the optimal LQR controller $K$ (that is, we run Algorithm 2 as opposed to Algorithm 3 for simplicity of comparison).
- State is initialized randomly, and perturbations are stochastic (to facilitate direct comparison with the experiments of Gradu et al., who did the same).
- We test both algorithms with $H = 5$, which was found to produce nearly-optimal results for both algorithms (theoretical performance is increasing in $H$, but converges with exponential falloff to a supremum).
- Noise magnitude is chosen arbitrarily across experiments. However, as the results are linear in magnitude (since both the systems and the control algorithms are linear), direct comparison to the experimental results of Gradu et al. [2020] is possible via scaling.

We also make two important nonstandard modifications to the experimental setup. Following the example of Gradu et al. [2020], we searched to find optimal multipliers for learning rate. This was found in their work to substantially enhance the performance of nonstochastic control algorithms against stochastic inputs in practice (due to the fact that stochastic inputs are unlikely to cause systematic learning errors early in the control run) and appears to be present in their experiments. We also test Gradu et al. [2020] under a version of their implementation modified with controller-magnitude bounding to ameliorate divergence issues (still visible in some spiking). We have not been able to determine the source thereof, and we do not have access to the code used to generate the plots visible in Gradu et al. [2020], so we are unable to determine the source of these spikes. However, this modification strictly improves their performance on the benchmarks, thus maintaining fair comparison.

Moving-average losses are graphed for EBPC, BPC, and LQR for the above problem with the three perturbation types of Gradu et al. [2020]: Gaussian, $c \sin(rx) \begin{bmatrix} 1 \\ 1 \end{bmatrix}$ (with period 40), and Gaussian Random Walk. $H = 5$ was used for both memory algorithms.

We observe that while our method has higher initial error, it has long-term error substantially lower than that of competing methods in aggregate (except in the sanity-check case of Gaussian noise, where it quickly converges to the LQR error as desired). Critically, it is able to adapt effectively to trends in perturbations more effectively than previous higher-error-rate algorithms, allowing for constant or decreasing error in environments with constant-size or increasing perturbations.

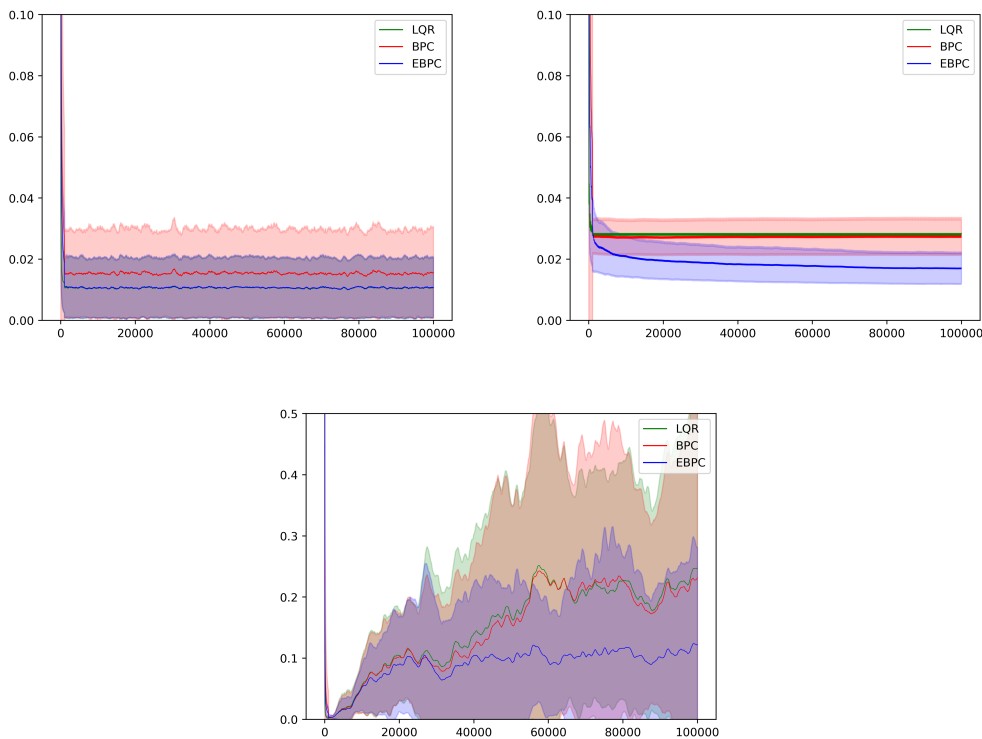

Figure 1: Loss ($y$-axis) of the three tested algorithms on Gaussian (left), sinusoidal (right), and Gaussian-walk (bottom) perturbation over time($x$-axis). Error bars indicate standard deviation across twelve draws of perturbation and controller randomness.

## C  Proof of `EBCO-M` Regret Guarantee

We prove the more general claim of Theorem 3.6.B, where the function $F_t$ is assumed to be conditionally $\sigma$-strongly convex. Denote $\bar{F}_t(x_{t-\bar{H}:t}) = \mathbb{E}[F_t(x_{t-\bar{H}:t}) \mid u_{1:t-H}, f_{H:t-H}]$.

Note that in Algorithm 1, with the delayed updates and the initialization $g_1 = \cdots = g_{\bar{H}} = 0$, we have $x_1 = \cdots = x_{2\bar{H}+1} = \arg\min_{x \in \mathcal{K}} R(x)$ and so learning begins only at the $2\bar{H} + 1$-th iteration. We can therefore decompose the regret against any $x \in \mathcal{K}$ as

$$\text{Regret}_T(x) = \underbrace{\left(\sum_{t=H}^{2\bar{H}} F_t(y_{t-\bar{H}:t}) - f_t(x)\right)}_{\text{(burn-in loss)}} + \underbrace{\left(\sum_{t=2\bar{H}+1}^{T} F_t(y_{t-\bar{H}:t}) - f_t(x)\right)}_{\text{(effective regret)}},$$

with burn-in loss crudely bounded by $HB$. We thus turn our attention in bounding the effective regret term.

The proof of the effective regret bound for Algorithm 1 consists of two main parts. In Section C.2, we show that the proposed gradient estimator $g_t$ has a bounded conditional bias. In Section C.3, we perform the analysis of a variant of the Regularized Follow-the-Leader (RFTL) algorithm, adding both a history component and a delayed update. Then, we show that together with the bounded conditional bias of our proposed gradient estimator, this yields an optimal regret bound for the bandit online convex optimization with memory algorithm outlined in Algorithm 1.

### C.1  Self-concordant barriers

The use of self-concordant barriers for bandit optimization is due to Abernethy et al. [2008], where the following properties are stated and used.

**Proposition C.1.** *$\nu$-self-concordant barriers over $\mathcal{K}$ satisfy the following properties:*

1. *Sum of two self-concordant functions is self-concordant. Linear and quadratic functions are self-concordant.*

2. *If $x, y \in \mathcal{K}$ satisfies $\|x - y\|_{\nabla^2 R(x)} < 1$, then the following inequality holds:*

$$(1 - \|x - y\|_{\nabla^2 R(x)})^2 \nabla^2 R(x) \preceq \nabla^2 R(y) \preceq \frac{1}{(1 - \|x - y\|_{\nabla^2 R(x)})^2} \nabla^2 R(x).$$

3. *The Dikin ellipsoid centered at any point in the interior of $\mathcal{K}$ w.r.t. a self-concordant barrier $R(\cdot)$ over $\mathcal{K}$ is completely contained in $\mathcal{K}$. Namely,*

$$\{y \in \mathbb{R}^n \mid \|y - x\|_{\nabla^2 R(x)} \leq 1\} \subset \mathcal{K}, \quad \forall x \in int(\mathcal{K}).$$

   *where*

$$\|v\|_{\nabla^2 R(x)} \overset{def}{=} \sqrt{v^\top \nabla^2 R(x) v}$$

4. *$\forall x, y \in int(\mathcal{K})$:*

$$R(y) - R(x) \leq \nu \log \frac{1}{1 - \pi_x(y)},$$

   *where $\pi_x(y) \overset{def}{=} \inf\{t \geq 0 : x + t^{-1}(y - x) \in \mathcal{K}\}$.*

## C.2 Gradient estimator

The goal of this section is to establish a bound on the conditional bias of the proposed gradient estimator $g_t$, formally given by the following proposition:

**Proposition C.2.** *The gradient estimator $g_t = n F_t(y_{t-\bar{H}:t}) \sum_{i=0}^{\bar{H}} A_{t-i}^{-1} u_{t-i}$ satisfies the following conditional bias bound in $\ell_2$: $\forall t \geq 2\bar{H} + 1$,*

$$\left\| \mathbb{E}[g_t \mid u_{1:t-H}, \mathcal{G}_{t-H}] - \nabla \bar{f}_t(x_t) \right\|_2 \leq \frac{16\sqrt{\eta}\beta n B H^3}{\sqrt{\sigma(t - 2\bar{H})}}.$$

**Lemma C.3.** *The gradient estimator $g_t$ is a conditionally unbiased estimator of the sum of the $H$ coordinate gradients of $F_t : \mathcal{K}^H \to \mathbb{R}$, i.e. $\forall t > H$,*

$$\mathbb{E}[g_t \mid u_{1:t-H}, \mathcal{G}_{t-H}] = \sum_{i=0}^{\bar{H}} \nabla_i \bar{F}_t(x_{t-\bar{H}:t}),$$

where $\nabla_i \bar{F}_t(z_1, \ldots, z_H) = \frac{\partial}{\partial z_i} \bar{F}_t(z_1, \ldots, z_H)$.

*Proof.* Let $q(x) = \frac{1}{2} x^\top A x + b^\top x + c$ be a (possibly random) quadratic function from $\mathbb{R}^n \to \mathbb{R}$ and $C$ be a (possibly random) symmetric, invertible matrix. Let $x_0 \in \mathbb{R}^n$ be a (possibly random) point of evaluation. Let $\mathcal{F}$ be a filtration such that $\{A, B, C, c, x_0\} \in \mathcal{F}$. Let $u \in \mathbb{R}^n$ be a random vector that is drawn from a symmetric distribution such that $\mathbb{E}[uu^\top] = \frac{r}{n} I_{n \times n}$ for some $r > 0$, and $u$ is independent of $\mathcal{F}$. Then,

$$\begin{aligned}
\mathbb{E}[C^{-1} u q(x_0 + Cu) \mid \mathcal{F}] &= \frac{1}{2} C^{-1} \mathbb{E}[u(x_0 + Cu)^\top A (x_0 + Cu) \mid \mathcal{F}] + C^{-1} \mathbb{E}[u b^\top (x_0 + Cu) \mid \mathcal{F}] \\
&= \frac{1}{2} C^{-1} \mathbb{E}[uu^\top] C (A + A^\top) x_0 + C^{-1} \mathbb{E}[uu^\top] C b \\
&= C^{-1} \mathbb{E}[uu^\top] C \left( \frac{1}{2}(A + A^\top) x_0 + b \right) \\
&= \frac{r}{n} \left( \frac{1}{2}(A + A^\top) x_0 + b \right) \\
&= \frac{r}{n} \nabla q(x_0).
\end{aligned}$$

Note that in Algorithm 1, $u_t$'s are sampled uniformly at random from the unit sphere in $\mathbb{R}^n$, so the distribution is symmetric and $\mathbb{E}[u_t u_t^\top] = \frac{1}{n} I_{n \times n}$, and thus $\mathbb{E}[u_{t-\bar{H}:t} u_{t-\bar{H}:t}^\top] = \frac{1}{n} I_{nH \times nH}$. Moreover, $\bar{F}_t, x_{t-\bar{H}:t}, A_{t-\bar{H}:t} \in \mathcal{F}_{t-H}$ and $u_{t-\bar{H}:t}$ are independent of $\mathcal{F}_{t-H}$. Let $\tilde{A}_t \stackrel{\text{def}}{=} \text{diag}(A_{t-\bar{H}}, \dots, A_t) \in \mathbb{R}^{nH \times nH}$ (i.e. the block matrix with diagonal blocks equal to $A_{t-\bar{H}}, \dots, A_t$). Then we have

$$\mathbb{E}[n\bar{F}_t(y_{t-\bar{H}:t}) \tilde{A}_t^{-1} u_{t-\bar{H}:t} \mid \mathcal{F}_{t-H}] = \nabla \bar{F}_t(x_{t-\bar{H}:t}).$$

Consider $\bar{g}_t = n\bar{f}_t(y_{t-\bar{H}:t}) \sum_{i=0}^{\bar{H}} A_{t-i}^{-1} u_{t-i}$. Note that $\tilde{A}_t^{-1} = \text{diag}(A_{t-\bar{H}}^{-1}, \dots, A_t^{-1})$ and by definition of $\bar{g}_t$, we have

$$\mathbb{E}[\bar{g}_t \mid \mathcal{F}_{t-H}] = \sum_{i=0}^{\bar{H}} \nabla_i \bar{F}_t(x_{t-\bar{H}:t}).$$

On the other hand, $x_t$ and $A_t$ are completely determined by $\{u_{1:t-H}\} \cup \{f_{H:t-H}\}$, and thus $\sum_{i=0}^{\bar{H}} A_{t-i}^{-1} u_{t-i}, y_{t-\bar{H}:t}$ is determined by $\{u_{1:t}\} \cup \{f_{H:t-H}\}$. Therefore,

$$
\begin{aligned}
\mathbb{E}[g_t \mid u_{1:t-H}, F_{H:t-H}] &= \mathbb{E}\left[ nF_t(y_{t-\bar{H}:t}) \sum_{i=0}^{\bar{H}} A_{t-i}^{-1} u_{t-i} \;\middle|\; u_{1:t-H}, \mathcal{G}_{t-H} \right] \\
&= \mathbb{E}\left[ \mathbb{E}\left[ nF_t(y_{t-\bar{H}:t}) \sum_{i=0}^{\bar{H}} A_{t-i}^{-1} u_{t-i} \;\middle|\; u_{1:t}, \mathcal{G}_{t-H} \right] \;\middle|\; u_{1:t-H}, \mathcal{G}_{t-H} \right] \\
&= \mathbb{E}\left[ n\mathbb{E}[F_t(y_{t-\bar{H}:t}) \mid u_{1:t}, \mathcal{G}_{t-H}] \sum_{i=0}^{\bar{H}} A_{t-i}^{-1} u_{t-i} \;\middle|\; u_{1:t-H}, \mathcal{G}_{t-H} \right] \\
&= \mathbb{E}\left[ n\bar{F}_t(y_{t-\bar{H}:t}) \sum_{i=0}^{\bar{H}} A_{t-i}^{-1} u_{t-i} \;\middle|\; u_{1:t-H}, \mathcal{G}_{t-H} \right] \\
&= \mathbb{E}[\bar{g}_t \mid u_{1:t-H}].
\end{aligned}
$$

We conclude that

$$\mathbb{E}[g_t \mid u_{1:t-H}, F_{H:t-H}] = \sum_{i=0}^{\bar{H}} \nabla_i \bar{F}_t(x_{t-\bar{H}:t}).$$

$\square$

**Definition C.4** (Local norms). *Denote the pair of dual norms $\| \cdot \|_t, \| \cdot \|_t^*$ on $\mathcal{K}$ as*

$$\|y\|_t \stackrel{\text{def}}{=} \|y\|_{A_t^{-2}} = \sqrt{y^\top (\nabla^2 R(x_t) + \eta \sigma t I) y} = \sqrt{y^\top A_t^{-2} y},$$

$$\|y\|_t^* \stackrel{\text{def}}{=} \|y\|_{A_t^2} = \sqrt{y^\top (\nabla^2 R(x_t) + \eta \sigma t I)^{-1} y} = \sqrt{y^\top A_t^2 y}.$$

*By Taylor expansion, $\forall R$ and $x, y \in \text{dom}(R)$, $\exists z = tx + (1-t)y$ for some $t = t(x, y, R) \in [0, 1]$ such that $D_R(x, y) := R(x) - R(y) - R(y)^\top(x - y) = \frac{1}{2}\|x - y\|_{\nabla^2 R(z)}^2$. We call $\| \cdot \|_{\nabla^2 R(z)}$ the induced norm by the Bregman divergence w.r.t. $R$ between $x$ and $y$. Denote as $\| \cdot \|_{t,t+1}$ the induced norm by the Bregman divergence w.r.t. $R_t(x) \stackrel{\text{def}}{=} R(x) + \frac{\eta \sigma}{2} \sum_{s=H}^{t} \|x - x_{s-\bar{H}}\|_2^2$ between $x_t$ and $x_{t+1}$. Denote its dual norm as $\| \cdot \|_{t,t+1}^*$.*

**Lemma C.5.** $\forall t \geq H$, *assuming $2\eta\|g_{t-\bar{H}}\|_t^* \leq 1$, then $\|x_t - x_{t+1}\|_t \leq 2\eta\|g_{t-\bar{H}}\|_t^*$.*

*Proof.* From Lemma 14 in Hazan and Levy [2014], $\|x - \arg\min_x h(x)\|_{\nabla^2 h(x)} \leq 2\|\nabla h(x)\|_{\nabla^2 h(x)}^*$, provided $h$ is self-concordant and $\|\nabla h(x)\|_{\nabla^2 h(x)}^* \leq 1$. Define $\Phi_t(x) \stackrel{\text{def}}{=} \eta \sum_{s=H}^{t} g_{s-\bar{H}}^\top x + R_t(x)$, where $R_t(x) = R(x) + \frac{\eta \sigma}{2} \sum_{s=H}^{t} \|x - x_{s-\bar{H}}\|_2^2$. $\Phi_t(\cdot)$ is self-concordant since it is the sum of a self-concordant function and sum of quadratic functions. Note that $x_{t+1} = \arg\min \Phi_t(x)$ by specification of Algorithm 1 and $\nabla^2 \Phi_t = \nabla^2 R_t$. Moreover, $\Phi_t(x) = \Phi_{t-1}(x) + \eta g_{t-\bar{H}}^\top x + \frac{\eta \sigma}{2}\|x - x_{t-\bar{H}}\|_2^2$. Since $x_t \in \text{int}(\mathcal{K})$ and minimizes $\Phi_{t-1}$, $\nabla \Phi_t(x) = \eta g_{t-\bar{H}}$. Applying Lemma 14 from Hazan and Levy [2014], $\|x_t - x_{t+1}\|_t \leq 2\eta\|g_{t-\bar{H}}\|_t^*$. $\square$

**Lemma C.6.** *If* $\eta \leq \frac{1}{8nH \log HB\sqrt{T}}$, *and assume that* $H = \text{poly}(\log T)$ *then the following inequalities hold deterministically* $\forall t \geq H$: $\forall((t - \bar{H}) \vee H) \leq s \leq t$,

$$\|g_{s-\bar{H}}\|_t^* \leq 2nBH \log H, \quad \|g_{s-\bar{H}}\|_{t,t+1}^* \leq 4nBH \log H.$$

*Proof.* We will show the joint hypothesis that: (1) $\left(\left(1 - \frac{1}{\sqrt{T}}\right) \wedge \frac{t}{t+1}\right) A_t \preceq A_{t+1} \preceq \frac{A_t}{\left(1 - \frac{1}{\sqrt{T}}\right) \wedge \frac{t}{t+1}}$;
(2) $\|g_{s-\bar{H}}\|_t^* \leq 2nBH \log H, \forall t - \bar{H} \leq s \leq t$; (3) $\|g_{s-\bar{H}}\|_{t,t+1}^* \leq 4nBH \log H, \forall t - \bar{H} \leq s \leq t$, for all $t$ by simultaneous induction on $t$. We divide our induction into two steps:

- **(1), (2), (3) hold for** $t = H, \ldots, 2\bar{H}$**:** note that $x_1 = \cdots = x_H = x = \underset{z \in \mathcal{K}}{\arg \min} R(z)$ and
  $g_1 = \cdots = g_{\bar{H}} = 0$, thus $x_{H+1} = \cdots = x_{2\bar{H}+1} = x$. Thus $\forall t = H, \ldots, 2\bar{H}, A_{t+1} \preceq A_t$
  holds trivially, to see the bound in the other direction, note that

  $$A_t = \sqrt{\frac{t+1}{t}} \left(\frac{t+1}{t} \nabla^2 R(x) + \eta\sigma(t+1)I\right)^{-\frac{1}{2}} \preceq \sqrt{\frac{t+1}{t}} A_{t+1}.$$

  $g_t = 0$ for $t = 1, \ldots, \bar{H}$, so (2), (3) follow.

- **Given that (1), (2), (3) hold for all** $t < T_0$**, show that (1), (2), (3) hold for** $t = T_0$**:**
  We first prove (2) for $s = t$. The bound holds identically up to constant factor $\leq 2$ for
  $s \in [t - \bar{H}, t)$ by induction hypothesis of $A_{t-\bar{H}:t}$. Assume $T_0 > 2\bar{H}$. Observe that
  $\left(1 - \frac{1}{\sqrt{T}}\right) \wedge \frac{t}{t+1} = \frac{t}{t+1}$ if and only if $t \leq \sqrt{T} - 1$. On the other hand, since by expression
  of $g_t = nF_t(y_{t-\bar{H}:t}) \sum_{i=0}^{\bar{H}} A_{t-i}^{-1} u_{t-i}$,

  $$\|g_{T_0-\bar{H}}\|_{T_0}^{*~2} = \|g_{T_0-\bar{H}}\|_{A_{T_0}^2}^2 \leq (nB)^2 \sum_{i,j=0}^{\bar{H}} u_{T_0-\bar{H}-i}^\top A_{T_0-\bar{H}-i}^{-1} A_{T_0}^2 A_{T_0-\bar{H}-j}^{-1} u_{T_0-\bar{H}-j}.$$

  Consider the induction hypothesis (1). For $T_0 \leq \sqrt{T}$, this implies that $\forall i \in [0, \bar{H}]$, there
  holds $\|A_{T_0-\bar{H}-i}^{-1} A_{T_0}\|_{\text{op}} \leq \frac{T_0}{T_0-\bar{H}-i}$, and thus

  $$\|g_{T_0-\bar{H}}\|_{T_0}^{*~2} \leq (nB)^2 \sum_{i,j=0}^{\bar{H}} \left(\frac{T_0}{T_0 - \bar{H} - i}\right) \left(\frac{T_0}{T_0 - \bar{H} - j}\right) = (nB)^2 \left(\sum_{i=\bar{H}}^{2\bar{H}} \frac{T_0}{T_0 - i}\right)^2,$$

  which is a decreasing function in $T_0$ and thus attains maximum at $T_0 = 2\bar{H} + 1$, giving that

  $$\|g_{T_0-\bar{H}}\|_{T_0}^{*~2} \leq (nB)^2 \left((2\bar{H} + 1) \sum_{i=1}^H \frac{1}{i}\right)^2 \leq 4(nBH)^2(\log(H))^2.$$

For $T_0 \geq \sqrt{T} + 2\bar{H} + 1$, $\|A_{T_0-\bar{H}-i}^{-1} A_{T_0}\|_{\mathrm{op}} \leq \left(1 - \frac{1}{\sqrt{T}}\right)^{-(\bar{H}+i)}$, so

$$\|g_{T_0-\bar{H}}\|_{T_0}^{*}{}^2 \leq (nB)^2 \sum_{i,j=0}^{\bar{H}} \left(1 - \frac{1}{\sqrt{T}}\right)^{-(2\bar{H}+i+j)}$$

$$= (nB)^2 \left(\left(1 - \frac{1}{\sqrt{T}}\right)^{-4\bar{H}} \sum_{i=0}^{\bar{H}} \left(1 - \frac{1}{\sqrt{T}}\right)^{i}\right)^2$$

$$= (nB)^2 \left(\left(1 - \frac{1}{\sqrt{T}}\right)^{-4\bar{H}} \sqrt{T}\left(1 - \left(1 - \frac{1}{\sqrt{T}}\right)^{\bar{H}}\right)\right)^2$$

$$\leq (nB)^2 \left(\left(1 - \frac{4\bar{H}}{\sqrt{T}}\right)^{-1} \sqrt{T}\left(1 - \left(1 - \frac{\bar{H}}{\sqrt{T}}\right)\right)\right)^2$$

$$\leq (nB)^2 \left(\frac{\sqrt{T}}{\sqrt{T} - 4\bar{H}}\right)^2 H^2$$

$$\leq 4(nBH)^2.$$

where the second inequality uses the inequality $(1+x)^r \geq 1 + rx$ for $x > -1$, integer $r \geq 1$, and the last inequality holds by assumption that $H = \mathrm{poly}(\log T)$.

For $T_0 \in (\sqrt{T}, \sqrt{T} + 2\bar{H} + 1)$,

$$\|A_{T_0} A_{T_0-\bar{H}-i}^{-1}\|_{\mathrm{op}} = \underbrace{\|A_{T_0} A_{T_0-1}^{-1} A_{T_0-1} \ldots A_{\sqrt{T}}^{-1}\|_{\mathrm{op}}}_{\leq \left(1 - \frac{1}{\sqrt{T}}\right)^{-(T_0-\sqrt{T})}} \underbrace{\|A_{\sqrt{T}} \ldots A_{T_0-\bar{H}-i}^{-1}\|_{\mathrm{op}}}_{\leq \frac{\sqrt{T}}{T_0-\bar{H}-i}}.$$

Thus, letting $\Delta \stackrel{\text{def}}{=} T_0 - \sqrt{T} \in [1, 2\bar{H}]$,

$$\|g_{T_0-\bar{H}}\|_{T_0}^{*}{}^2 \leq (nB)^2 \left(1 - \frac{1}{\sqrt{T}}\right)^{-\Delta} \left(\sum_{i=0}^{\bar{H}} \frac{\sqrt{T}}{\sqrt{T} + \Delta - \bar{H} - i}\right)^2$$

$$\leq (nB)^2 \underbrace{\left(1 - \frac{1}{\sqrt{T}}\right)^{-2\bar{H}}}_{\leq \left(1 - \frac{2\bar{H}}{\sqrt{T}}\right)^{-1}} \underbrace{H^2 \left(\frac{\sqrt{T}}{\sqrt{T} - 2\bar{H}}\right)^2}_{\leq 2H^2}$$

$$\leq 4(nBH)^2.$$

Then by Lemma C.5 and choice of $\eta$, $\|x_{T_0} - x_{T_0+1}\|_{T_0} \leq 2\eta \|g_{T_0-\bar{H}}\|_{T_0}^{*} \leq \frac{1}{\sqrt{T}}$. $R_{T_0}(x)$ is self-concordant, and $A_{T_0}^{-1} = \left(\nabla^2 R_{T_0}(x_{T_0})\right)^{\frac{1}{2}}$, so by the local Hessian bound in Proposition C.1,

$$\left(\left(1 - \frac{1}{\sqrt{T}}\right) \wedge \frac{t}{t+1}\right) A_{T_0}^{-1} \preceq (1 - \|x_{T_0} - x_{T_0+1}\|_{T_0}) A_{T_0}^{-1}$$

$$\preceq A_{T_0+1}^{-1}$$

$$\preceq \frac{A_{T_0}^{-1}}{1 - \|x_{T_0} - x_{T_0+1}\|_{T_0}}$$

$$\preceq \frac{1}{\left(1 - \frac{1}{\sqrt{T}}\right) \wedge \frac{t}{t+1}} A_{T_0}^{-1},$$

thus proving (1) for $t = T_0$.

To prove (4) for $t = T_0$, observe that if $z$ is a convex combination of $x_{T_0}$ and $x_{T_0+1}$, then

$$\|z - x_{T_0}\|_{\nabla^2 R_{T_0}(x_{T_0})} \leq \|x_{T_0+1} - x_{T_0}\|_{\nabla^2 R_{T_0}(x_{T_0})} \leq \frac{1}{\sqrt{T}},$$

and thus again by Proposition C.1,

$$(\nabla^2 R(z) + \eta\sigma tI)^{-1} \preceq \left(1 - \frac{1}{\sqrt{T}}\right)^{-2} (\nabla^2 R(x_t) + \eta\sigma tI)^{-1},$$

and thus since $\exists z$ convex combination of $x_{T_0}, x_{T_0+1}$: $\|g_{T_0-\bar{H}}\|^*_{T_0,T_0+1} = \|g_{T_0-\bar{H}}\|_{\nabla^{-2}R_{T_0}(z)}$ and thus $\|g_{T_0-\bar{H}}\|^*_{T_0,T_0+1} \leq \left(1 - \frac{1}{\sqrt{T}}\right)^{-2} \|g_{T_0-\bar{H}}\|^*_t \leq 4nBH \log H$.

$\qquad\qquad\qquad\qquad\qquad\qquad\qquad\qquad\qquad\qquad\qquad\qquad\qquad\qquad\qquad\qquad\square$

**Lemma C.7** (Iterate bound). $\forall t \geq H$, *the Euclidean distance between neighboring iterates is bounded by*

$$\|x_t - x_{t+1}\|_2 \leq \frac{4\sqrt{\eta}}{\sqrt{\sigma(t - \bar{H})}} \|g_{t-\bar{H}}\|^*_{t,t+1} \leq \frac{16\sqrt{\eta}nBH \log H}{\sqrt{\sigma(t - \bar{H})}}.$$

*Proof.* The second inequality follows from the previous lemma, so we prove the first. Recall $\Phi_t$ as defined in Lemma 14. By Taylor expansion, optimality condition and linearity of $\Phi_t(\cdot) - R_t(\cdot)$,

$$\Phi_t(x_t) = \Phi_t(x_{t+1}) + (x_t - x_{t+1})^\top \nabla\Phi_t(x_{t+1}) + D_{\Phi_t}(x_t, x_{t+1}) \geq \Phi_t(x_{t+1}) + D_{\tilde{R}_t}(x_t, x_{t+1}),$$

which by decomposing $\Phi_t$ implies

$$D_{R_t}(x_t, x_{t+1}) \leq [\Phi_{t-1}(x_t) - \Phi_{t-1}(x_{t+1})] + \eta g_{t-\bar{H}}^\top (x_t - x_{t+1}) \leq \eta g_{t-\bar{H}}^\top (x_t - x_{t+1}).$$

and thus for some $z = sx_t + (1 - s)x_{t+1}$, $s \in [0, 1]$, $\|x_t - x_{t+1}\|^2_{\nabla^2 R_t(z)} \leq 2\eta g_{t-\bar{H}}^\top (x_t - x_{t+1}) \leq 2\eta\|g_{t-\bar{H}}\|^*_{\nabla^2 R_t(z)}\|x_t - x_{t+1}\|_{\nabla^2 R_t(z)}$, thus establishing the bound $\|x_t - x_{t+1}\|_{\nabla^2 R_t(z)} \leq 2\eta\|g_{t-\bar{H}}\|^*_{\nabla^2 R_t(z)}$. Since $R_t(\cdot)$ is $\eta\sigma(t - \bar{H})$-strongly convex,

$$\|x_t - x_{t+1}\|_2 \leq \frac{2}{\sqrt{\eta\sigma(t - \bar{H})}}\|x_t - x_{t+1}\|_{\nabla^2 R_t(z)} \leq \frac{4\sqrt{\eta}}{\sqrt{\sigma(t - \bar{H})}}\|g_{t-\bar{H}}\|^*_{t,t+1}.$$

$\qquad\qquad\qquad\qquad\qquad\qquad\qquad\qquad\qquad\qquad\qquad\qquad\qquad\qquad\qquad\qquad\square$

**Corollary C.8.** *Define* $f_t : \mathcal{K} \to \mathbb{R}_+$ *by* $f_t(x) \overset{def}{=} F_t(x, \ldots, x)$. *We have that* $\forall t > 2\bar{H}$,

$$\left\|\mathbb{E}[g_t \mid u_{1:t-H}, \mathcal{G}_{t-H}] - \nabla\bar{f}_t(x_t)\right\|_2 \leq \frac{16\sqrt{\eta}\beta nBH^3}{\sqrt{\sigma(t - 2\bar{H})}}.$$

*Proof.* By the earlier bounds,

$$\left\|\mathbb{E}[g_t \mid u_{1:t-H}, \mathcal{G}_{t-H}] - \nabla \bar{f}_t(x_t)\right\|_2^2 = \left\|\sum_{i=0}^{\bar{H}} \nabla_i \bar{F}_t(x_{t-\bar{H}:t}) - \nabla \bar{f}_t(x_t)\right\|_2^2 \qquad \text{Lemma C.3}$$

$$= H\|\nabla \bar{F}_t(x_{t-\bar{H}:t}) - \nabla \bar{F}_t(x_t, \cdots, x_t)\|_2^2$$

$$\leq H\beta^2 \|(x_{t-\bar{H}}, \cdots, x_t) - (x_t, \cdots, x_t)\|_2^2$$

$$\leq H\beta^2 \sum_{i=1}^{\bar{H}} \|x_t - x_{t-i}\|_2^2$$

$$= H\beta^2 \sum_{i=1}^{\bar{H}} \sum_{j=1}^{i} \|x_{t-j+1} - x_{t-j}\|_2^2$$

$$\leq \frac{256\eta\beta^2 n^2 B^2 H^3 \log^2 H}{\sigma} \sum_{i=1}^{\bar{H}} \sum_{j=1}^{i} \frac{1}{t-j-\bar{H}} \qquad \text{Lemma C.7}$$

$$\leq \frac{256\eta\beta^2 n^2 B^2 H^3 \log^2 H}{\sigma} \frac{H^2}{2} \frac{1}{t-2\bar{H}}$$

$$= \frac{128\eta\beta^2 n^2 B^2 H^5 \log^2 H}{\sigma(t-2\bar{H})}.$$

Then taking the square root of each side yields the desired bound. $\qquad\square$

### C.3 Regret analysis

The previous section established a conditional bias bound on the gradient estimator $g_t$ used in Algorithm 1. In this section, we use this conditional bias bound together with an analysis on the subroutine algorithm, Regularized Follow-the-Leader with Delay (RFTL-D), to establish a regret guarantee for Algorithm 1.

**Decomposition of effective regret.** Letting $w = \arg\min_{x\in\mathcal{K}} \sum_{t=H}^{T} f_t(x)$, we divide the expected regret into three parts, which we will bound separately:

$$\text{Effective-Regret}_T = \mathbb{E}\left[\underbrace{\sum_{t=2\bar{H}+1}^{T} F_t(y_{t-\bar{H}:t}) - F_t(x_{t-\bar{H}:t})}_{\text{(1: estimator movement cost)}}\right] + \mathbb{E}\left[\underbrace{\sum_{t=2\bar{H}+1}^{T} F_t(x_{t-\bar{H}:t}) - f_t(x_t)}_{\text{(2: history movement cost)}}\right]$$

$$+ \mathbb{E}\left[\underbrace{\sum_{t=2\bar{H}+1}^{T} f_t(x_t) - f_t(w)}_{\text{(3: RFTL-D effective regret)}}\right].$$

To bound the estimator movement cost, note that $\|A_t^2\|_{\text{op}} = \|(\underbrace{\nabla^2 R(x_t)}_{\succeq 0} + \eta\sigma t I)^{-1}\|_{\text{op}} \leq \frac{1}{\eta\sigma t}$, and thus

$$(1) \leq \sum_{t=2\bar{H}+1}^{T} \mathbb{E}\left[\mathbb{E}\left[\nabla F_t(x_{t-\bar{H}:t})^T (A_{t-\bar{H}:t} u_{t-\bar{H}:t}) + \frac{\beta}{2}\|(A_{t-\bar{H}:t} u_{t-\bar{H}:t})\|_2^2 \;\Big|\; \mathcal{F}_{t-H}\right]\right]$$

$$= \frac{\beta}{2} \sum_{t=2\bar{H}+1}^{T} \mathbb{E}\left[\sum_{s=t-\bar{H}}^{t} \|A_s u_s\|_2^2\right] \leq \frac{\beta}{2} \sum_{t=2\bar{H}+1}^{T} \mathbb{E}\left[\sum_{s=t-\bar{H}}^{t} \|A_s^2\|_{\text{op}}\right] \leq \frac{\beta}{2\eta\sigma} \sum_{t=2\bar{H}+1}^{T} \sum_{s=t-\bar{H}}^{t} \frac{1}{s}$$

$$\leq \frac{\beta H \log T}{2\eta\sigma}.$$

To bound the history movement cost, note that by the iterate bound obtained in the analysis of Corollary C.8,

$$(2) = \mathbb{E}\left[\sum_{t=2\bar{H}+1}^{T} F_t(x_{t-\bar{H}:t}) - f_t(x_t)\right] \leq L \sum_{t=2\bar{H}+1}^{T} \|(x_{t-\bar{H}}, \ldots, x_t) - (x_t, \ldots, x_t)\|_2$$

$$\leq \frac{16\sqrt{\eta}nLBH^2\log H}{\sqrt{\sigma}} \sum_{t=2\bar{H}+1}^{T} \frac{1}{\sqrt{t-2\bar{H}}} \leq \frac{16\sqrt{\eta T}nLBH^2\log H}{\sqrt{\sigma}}.$$

It remains to bound the last term in the regret decomposition. For this, we analyze RFTL with delay (RFTL-D).

### C.3.1 RFTL with delay (RFTL-D)

The subroutine algorithm we used in Algorithm 1 is Regularized-Follow-the-Leader with delay (RFTL-D). We first analyze its regret bound in the full information setting. Consider a sequence of convex loss functions $\{\ell_t\}_{t=H}^{T}$ and the following algorithm.

---

**Algorithm 4** RFTL-D

---

1: Input: Bounded, convex, and closed set $\mathcal{K}$, time horizon $T$, delayed length $H$, step size $\eta > 0$, regularization function $R(\cdot)$.
2: Initialize $x_t = \arg\min_{x\in\mathcal{K}} R(x), \forall t = 1, \ldots, H$.
3: Set $\ell_t = 0, \forall t = 1, \ldots, \bar{H}$.
4: **for** $t = H, \ldots, T$ **do**
5:     Play $x_t$, observe and store cost function $\ell_t(x_t)$.
6:     Update $x_{t+1} = \arg\min_{x\in\mathcal{K}}\left\{\sum_{s=H}^{t} \ell_{s-\bar{H}}(x) + \frac{1}{\eta}R(x)\right\}$.
7: **end for**

---

Again, note that by design of Algorithm 4, the learning begins only after $2\bar{H}+1$-th iteration. Therefore, it suffices to bound effective regret Effective-Regret$_T \overset{\text{def}}{=} \sum_{t=2\bar{H}+1}^{T} \ell_t(x_t) - \min_{x\in\mathcal{K}} \sum_{t=2\bar{H}+1}^{T} \ell_t(x)$. First, we want to establish a regret inequality which is analogous to the standard regret inequality seen in the Regularized Follow-the-Leader algorithm without delay.

**Theorem C.9** (RFTL-D effective regret bound). *With convex loss functions bounded by $B$, Algorithm 4 guarantees the following regret bound for every $x \in \mathcal{K}$:*

$$\textit{Effective-Regret}_T(x) \leq 2\eta \sum_{t=2\bar{H}+1}^{T} \|\nabla_{t-\bar{H}}\|_{t,t+1,\Phi_t}^* \left\|\sum_{s=t-\bar{H}}^{t} \nabla_{s-\bar{H}}\right\|_{t,t+1,\Phi_t}^*$$

$$+ \frac{R(x) - R(x_{2\bar{H}+1})}{\eta} + 2HB,$$

*where $\|\cdot\|_{t,t+1,\Phi_t}$ and $\|\cdot\|_{t,t+1,\Phi_t}^*$ denote the local norm and its dual induced by the Bregman divergence w.r.t. the function $\Phi_t(x) \overset{\text{def}}{=} \eta \sum_{s=H}^{t} \ell_{s-\bar{H}}(x) + R(x)$ between $x_t$ and $x_{t+1}$.*

*Proof.* The proof of Theorem C.9 follows from the following lemma.

**Lemma C.10.** *Suppose the cost functions $\ell_t$ are bounded by $B$. Algorithm 4 guarantees the following regret bound:*

$$\textit{Effective-Regret}_T(x) \leq \sum_{t=2\bar{H}+1}^{T} \nabla_{t-\bar{H}}^{\top}(x_{t-\bar{H}} - x_{t+1}) + \frac{R(x) - R(x_{2\bar{H}+1})}{\eta} + 2HB.$$

*Proof of Lemma C.10.* Denote $h_{2\bar{H}}(x) \overset{\text{def}}{=} \frac{1}{\eta}R(x)$, $h_t(x) \overset{\text{def}}{=} \ell_{t-\bar{H}}(x), \forall t \geq 2\bar{H}+1$. Then, by the usual FTL-BTL analysis, $\forall x \in \mathcal{K}, T \geq 2\bar{H}, \sum_{t=2\bar{H}}^{T} h_t(x) \geq \sum_{t=2\bar{H}}^{T} h_t(x_{t+1})$. Thus, we can

bound regret by

$$\text{Regret}_T(x) \le \left( \sum_{t=H}^{T-\bar{H}} \ell_t(x_t) - \ell_t(x) \right) + 2HB$$

$$= \left( \sum_{t=H}^{T-\bar{H}} \ell_t(x_t) - \ell_t(x) \right) + 2HB$$

$$= \left( \sum_{t=H}^{T-\bar{H}} \ell_t(x_t) - \ell_t(x_{t+H}) \right) + \left( \sum_{t=H}^{T-\bar{H}} \ell_t(x_{t+H}) - \ell_t(x) \right) + 2HB$$

$$= \left( \sum_{t=2\bar{H}+1}^{T} \ell_{t-\bar{H}}(x_{t-\bar{H}}) - \ell_{t-\bar{H}}(x_{t+1}) \right) + \left( \sum_{t=2\bar{H}+1}^{T} \ell_{t-\bar{H}}(x_{t+1}) - \ell_{t-\bar{H}}(x) \right) + 2HB$$

$$\le \sum_{t=2\bar{H}+1}^{T} \nabla_{t-\bar{H}}^{\top}(x_{t-\bar{H}} - x_{t+1}) + \left( \sum_{t=2\bar{H}+1}^{T} h_t(x_{t+1}) - h_t(x) \right) + 2HB$$

$$\le \sum_{t=2\bar{H}+1}^{T} \nabla_{t-\bar{H}}^{\top}(x_{t-\bar{H}} - x_{t+1}) + \frac{R(x) - R(x_{2\bar{H}+1})}{\eta} + 2HB,$$

where the last inequality follows from the inequality $\sum_{t=2\bar{H}}^{T} h_t(x) \ge \sum_{t=2\bar{H}}^{T} h_t(x_{t+1}), \forall x \in \mathcal{K}$. $\qquad\square$

Consider the function $\Phi_t(x) \stackrel{\text{def}}{=} \eta \sum_{s=2\bar{H}+1}^{t} \ell_{s-\bar{H}}(x) + R(x), t \ge 2\bar{H} + 1$. By Taylor expansion and optimality condition, we have that $\forall t \ge 2\bar{H} + 1$,

$$\Phi_t(x_{t-\bar{H}}) = \Phi_t(x_{t+1}) + (x_{t-\bar{H}} - x_{t+1})^{\top} \nabla \Phi_t(x_{t+1}) + D_{\Phi_t}(x_{t-\bar{H}}, x_{t+1})$$
$$\ge \Phi_t(x_{t+1}) + D_{\Phi_t}(x_{t-\bar{H}}, x_{t+1}),$$

which implies a bound on the Bregman divergence between $x_{t-\bar{H}}$ and $x_{t+1}$ with respect to $\Phi_t$,

$$D_{\Phi_t}(x_{t-\bar{H}}, x_{t+1}) \le \Phi_t(x_{t-\bar{H}}) - \Phi_t(x_{t+1})$$

$$\le \underbrace{\Phi_{t-H}(x_{t-\bar{H}}) - \Phi_{t-H}(x_{t+1})}_{\le 0} + \eta \sum_{s=t-\bar{H}}^{t} \nabla_{s-\bar{H}}^{\top}(x_{t-\bar{H}} - x_{t+1})$$

$$\le \eta \left( \left\| \sum_{s=t-\bar{H}}^{t} \nabla_{s-\bar{H}} \right\|_{t,t+1,\Phi_t}^{*} \right) \|x_{t-\bar{H}} - x_{t+1}\|_{t,t+1,\Phi_t}$$

$$= \eta \left( \left\| \sum_{s=t-\bar{H}}^{t} \nabla_{s-\bar{H}} \right\|_{t,t+1,\Phi_t}^{*} \right) \sqrt{2 D_{\Phi_t}(x_{t-\bar{H}}, x_{t+1})},$$

which gives the bound on both the Bregman divergence and the iterate distance in terms of Bregman divergence induced norm between $x_{t-\bar{H}}$ and $x_{t+1}$,

$$D_{\Phi_t}(x_{t-\bar{H}}, x_{t+1}) \le 2\eta^2 \left\| \sum_{s=t-\bar{H}}^{t} \nabla_{s-\bar{H}} \right\|_{t,t+1,\Phi_t}^{*2},$$

$$\|x_{t-\bar{H}} - x_{t+1}\|_{t,t+1,\Phi_t} \le 2\eta \left\| \sum_{s=t-\bar{H}}^{t} \nabla_{s-\bar{H}} \right\|_{t,t+1,\Phi_t}^{*}.$$

Following the expression of the regret bound established in Lemma C.10, we bound

$$\text{Effective-Regret}_T(x) \leq \sum_{t=2\bar{H}+1}^{T} \nabla_{t-\bar{H}}^{\top}(x_{t-\bar{H}} - x_{t+1}) + \frac{R(x) - R(x_{2\bar{H}+1})}{\eta} + 2HB$$

$$\leq \sum_{t=2\bar{H}+1}^{T} \|\nabla_{t-\bar{H}}\|_{t,t+1,\Phi_t}^* \|x_{t-\bar{H}} - x_{t+1}\|_{t,t+1,\Phi_t}$$

$$+ \frac{R(x) - R(x_{2\bar{H}+1})}{\eta} + 2HB$$

$$\leq 2\eta \sum_{t=2\bar{H}+1}^{T} \|\nabla_{t-\bar{H}}\|_{t,t+1,\Phi_t}^* \left\| \sum_{s=t-\bar{H}}^{t} \nabla_{s-\bar{H}} \right\|_{t,t+1,\Phi_t}^*$$

$$+ \frac{R(x) - R(x_{2\bar{H}+1})}{\eta} + 2HB.$$

$\square$

**Corollary C.11.** *In Algorithm 4, if the loss functions are assumed to be $\sigma$-strongly smooth and bounded by B, and the updates are given by*

$$x_{t+1} = \arg\min_{x \in \mathcal{K}} \left\{ \left( \sum_{s=H}^{t} \nabla_{s-\bar{H}}^{\top} x + \frac{\sigma}{2} \|x - x_{s-\bar{H}}\|_2^2 \right) + \frac{1}{\eta} R(x) \right\},$$

*then Algorithm 4 guarantees the following regret bound:*

$$\text{Effective-Regret}_T \leq 2\eta \sum_{t=2\bar{H}+1}^{T} \|\nabla_{t-\bar{H}}\|_{t,t+1}^* \left\| \sum_{s=t-\bar{H}}^{t} \nabla_{s-\bar{H}} \right\|_{t,t+1}^* + \frac{R(x) - R(x_{2\bar{H}+1})}{\eta} + HB,$$

*with the local norms defined as in Definition C.4.*

*Proof.* We make use of a lemma of Zinkevich [2003] and Hazan et al. [2007]:

**Lemma C.12.** *The following inequality holds for two sequences of convex loss functions* $\{\ell_t\}_{t=1}^{T}, \{\tilde{\ell}_t\}_{t=1}^{T}$ *if* $\tilde{\ell}_t(x_t) = \ell_t(x_t)$ *and* $\tilde{\ell}_t(x) \leq \ell_t(x), \forall x \in \mathcal{K}$:

$$\sum_{t=1}^{T} \ell_t(x_t) - \min_{x \in \mathcal{K}} \sum_{t=1}^{T} \ell_t(x) \leq \sum_{t=1}^{T} \tilde{\ell}_t(x_t) - \min_{x \in \mathcal{K}} \sum_{t=1}^{T} \tilde{\ell}_t(x).$$

Since we assume $\ell_t$s to be $\sigma$-strongly convex, we can construct $\tilde{\ell}_t$ that satisfies $\tilde{\ell}_t(x_t) = \ell_t(x_t)$ and $\tilde{\ell}_t(x) \leq \ell_t(x), \forall x \in \mathcal{K}$ as the following:

$$\tilde{\ell}_t(x) \stackrel{\text{def}}{=} \ell_t(x_t) + \nabla\ell_t(x_t)^{\top}(x - x_t) + \frac{\sigma}{2}\|x - x_t\|_2^2.$$

The update then becomes

$$x_{t+1} = \arg\min_{x \in \mathcal{K}} \left\{ \left( \sum_{s=H}^{t} \nabla_{s-\bar{H}}^{\top} x + \frac{\sigma}{2} \|x - x_{s-\bar{H}}\|_2^2 \right) + \frac{1}{\eta} R(x) \right\}$$

$$= \arg\min_{x \in \mathcal{K}} \left\{ \sum_{s=H}^{t} \tilde{\ell}_{s-\bar{H}}(x) + \frac{1}{\eta} R(x) \right\}.$$

Note that $\nabla\tilde{\ell}_t(x_t) = \nabla\ell_t(x_t)$. Let $\|x\|_{t,t+1} = \|x\|_{t,t+1,R_t}$, where $R_t(x) = R(x) + \frac{\eta\sigma}{2}\sum_{s=H}^{t}\|x - x_{s-\bar{H}}\|_2^2$ Then from Theorem C.9 and linearity of $\Phi_t - R_t$,

$$\text{Effective-Regret}_T \leq 2\eta \sum_{t=2\bar{H}+1}^{T} \|\nabla_{t-\bar{H}}\|_{t,t+1,\Phi_t}^* \left\|\sum_{s=t-\bar{H}}^{t} \nabla_{s-\bar{H}}\right\|_{t,t+1,\Phi_t}^*$$

$$+ \frac{R(x) - R(x_{2\bar{H}+1})}{\eta} + HB$$

$$= 2\eta \sum_{t=2\bar{H}+1}^{T} \|\nabla_{t-\bar{H}}\|_{t,t+1}^* \left\|\sum_{s=t-\bar{H}}^{t} \nabla_{s-\bar{H}}\right\|_{t,t+1}^* + \frac{R(x) - R(x_{2\bar{H}+1})}{\eta} + HB.$$

$\square$

Corollary C.11 implies that the above regret bound holds if we run RFTL-D with the true gradient of $\{f_t\}_{t=H}^{T}$ in the full information setting. In the bandit setting, Algorithm 1 is run with the gradient estimators $g_t$ in place of the actual gradient $\nabla f_t(x_t)$. We introduce the following lemma that bounds the regret of a first-order OCO algorithm $\mathcal{A}$ when using gradient estimators in place of the true gradient:

**Lemma C.13.** *Let $\ell_1, \ldots, \ell_T : \mathcal{K} \to \mathbb{R}_+$ be a sequence of differentiable convex loss functions. Let $\mathcal{A}$ be a first-order OCO algorithm over $\mathcal{K}$ with regret bound*

$$Regret_T^{\mathcal{A}} \leq D_{\mathcal{A}}(\nabla\ell_1(x_1), \ldots, \nabla\ell_T(x_T)).$$

*Define $x_1 \leftarrow \mathcal{A}(\emptyset)$, $x_t \leftarrow \mathcal{A}(g_1, \ldots, g_{t-1})$ for $t \leq T$. Suppose $\exists B(t)$ such that the gradient estimator $g_t$ satisfies $\|\mathbb{E}[g_t \mid \mathcal{G}_t] - \nabla\ell_t(x_t)\|_2 \leq B(t)$, where $\mathcal{G}_t$ is any filtration such that $\ell_t, x_t \in \mathcal{G}_t$. Then $\forall x \in \mathcal{K}$,*

$$\mathbb{E}\left[\sum_{t=1}^{T} \ell_t(x_t) - \ell_t(x)\right] \leq \mathbb{E}[D_{\mathcal{A}}(g_1, \ldots, g_T)] + D\sum_{t=1}^{T} B(t).$$

*Proof.* Define $q_t(x) \stackrel{\text{def}}{=} \ell_t(x) + (g_t - \nabla\ell_t(x_t))^\top x$. Then $\nabla q_t(x_t) = g_t$. Since $\mathcal{A}$ is a first-order OCO algorithm, $\mathcal{A}(q_1, \ldots, q_{t-1}) = \mathcal{A}(g_1, \ldots, g_{t-1})$, $\forall t$. Moreover, $\forall x \in \mathcal{K}$,

$$\sum_{t=1}^{T} q_t(x_t) - q_t(x) \leq D_{\mathcal{A}}(g_1, \ldots, g_T).$$

By assumption, $\forall t, x$,

$$\mathbb{E}[q_t(x_t) - q_t(x)] = \mathbb{E}[\ell_t(x_t) - \ell_t(x)] - \mathbb{E}[(g_t - \nabla\ell_t(x_t))^\top(x - x_t)]$$
$$= \mathbb{E}[\ell_t(x_t) - \ell_t(x)] - \mathbb{E}[\mathbb{E}[(g_t - \nabla\ell_t(x_t))^\top(x - x_t) \mid \mathcal{G}_t]]$$
$$= \mathbb{E}[\ell_t(x_t) - \ell_t(x)] - \mathbb{E}[(\mathbb{E}[g_t \mid \mathcal{G}_t] - \nabla\ell_t(x_t))^\top(x - x_t)]$$
$$\geq \mathbb{E}[\ell_t(x_t) - \ell_t(x)] - DB(t).$$

Then

$$\mathbb{E}\left[\sum_{t=1}^{T} \ell_t(x_t) - \ell_t(x)\right] \leq \mathbb{E}\left[\sum_{t=1}^{T} q_t(x_t) - q_t(x)\right] + D\sum_{t=1}^{T} B(t)$$

$$\leq \mathbb{E}[D_{\mathcal{A}}(g_1, \ldots, g_T)] + D\sum_{t=1}^{T} B(t).$$

$\square$

With Corollary C.11 and Lemma C.13, we are ready to bound the last term in the regret decomposition.

**Lemma C.14.** *For any sequence of loss functions $\{F_t\}_{t=H}^T$ satisfying assumptions in 3.1.1, the sequence $\{x_t\}_{t=H}^T$ returned by Algorithm 1 satisfies $\forall x \in \mathcal{K}$,*

$$\mathbb{E}\left[\sum_{t=2\bar{H}+1}^T f_t(x_t) - f_t(x)\right] \le 16\eta n^2 B^2 H^3 \log^2 HT + \frac{\nu \log T}{\eta} + 2HB + \frac{16\sqrt{\eta T}\beta n BDH^4}{\sqrt{\sigma}}.$$

*Proof.* Recall the definition of the function $\pi_w$ with respect to $w \in \text{int}(\mathcal{K})$ in Proposition C.1. For a given $w \in \text{int}(\mathcal{K})$, $\pi_w : \mathcal{K} \to \mathbb{R}_+$ is given by $\pi_w(y) = \inf\{t \ge 0 : w + t^{-1}(y-w) \in \mathcal{K}\}$. Note that we can assume without loss of generality that $\pi_{x_{2\bar{H}+1}}(x) \le 1 - T^{-1}$. Since $F_t$ is $L$-Lipschitz, if $x$ violates this assumption, i.e. $\pi_{x_{2\bar{H}+1}}(x) > 1 - T^{-1}$, $\exists x' \in \mathcal{K}$ with $\|x - x'\|_2 \le \mathcal{O}(T^{-1})$ and $\pi_{x_H}(x') \le 1 - T^{-1}$, and if total loss playing $x'$ is at most $O(1)$ away from playing $x$. With this assumption, Proposition C.1 readily bounds the quantity $R(w) - R(x_{2\bar{H}+1})$, which is always non-negative since $x_{2\bar{H}+1} = x_1 = \arg\min_{x \in \mathcal{K}} R(x)$.

Let $\mathcal{A}$ be the RFTL-D algorithm with updates for $\sigma$-strongly convex functions. Then, the effective regret of bandit RFTL-D with respect to any $x \in \mathcal{K}$ is bounded by

$$\mathbb{E}\left[\sum_{t=2\bar{H}+1}^T f_t(x_t) - f_t(x)\right]$$

$$= \mathbb{E}\left[\sum_{t=2\bar{H}+1}^T \mathbb{E}\left[f_t(x_t) - f_t(w) \mid u_{1:t}, \mathcal{G}_{t-H}\right]\right]$$

$$= \mathbb{E}\left[\sum_{t=2\bar{H}+1}^T \bar{f}_t(x_t) - \bar{f}_t(w)\right]$$

$$\le \mathbb{E}[D_\mathcal{A}(g_H, \ldots, g_T)] + \frac{16\sqrt{\eta}\beta n BDH^3}{\sqrt{\sigma}} \sum_{t=2\bar{H}+1}^T \frac{1}{\sqrt{t-2\bar{H}}} \quad \text{(Corollary C.8, Lemma C.13)}$$

$$\le 2\eta \sum_{t=2\bar{H}+1}^T \mathbb{E}\left[\|g_{t-\bar{H}}\|_{t,t+1}^* \left\|\sum_{s=t-\bar{H}}^t g_{s-\bar{H}}\right\|_{t,t+1}^*\right] + \frac{R(x) - R(x_{2\bar{H}+1})}{\eta} + 2HB$$

$$\quad + \frac{16\sqrt{\eta T}\beta n BDH^4}{\sqrt{\sigma}} \quad \text{(Corollary C.11)}$$

$$\le 16\eta n^2 B^2 H^3 \log^2 HT + \frac{R(x) - R(x_{2\bar{H}+1})}{\eta} + 2HB + \frac{16\sqrt{\eta T}\beta n BDH^4}{\sqrt{\sigma}} \quad \text{(Lemma C.6)}$$

$$\le 16\eta n^2 B^2 H^3 \log^2 HT + \frac{\nu \log T}{\eta} + 2HB + \frac{16\sqrt{\eta T}\beta n BDH^4}{\sqrt{\sigma}} \quad \text{(Proposition C.1)}.$$

$\square$

Lemma C.14 establishes the bound on the expected bandit RFTL-D regret. Combining the above bounds, with $H = \text{poly}(\log T)$ we have the following expected regret bound for Algorithm 1:

$$\text{Effective-Regret}_T \le \underbrace{\frac{\beta H \log T}{2\eta\sigma}}_{\text{bound on (1)}} + \underbrace{\frac{16\sqrt{\eta T}nLBH^2 \log H}{\sqrt{\sigma}}}_{\text{bound on (2)}}$$

$$+ \underbrace{16\eta n^2 B^2 H^3 \log^2 T + \frac{\nu \log T}{\eta} + 2HB + \frac{16\sqrt{\eta T}\beta n BDH^4}{\sqrt{\sigma}}}_{\text{bound on (3)}}$$

$$\le \mathcal{O}\left(\frac{\beta}{\sigma}n\text{poly}(H)\sqrt{T}\right) = \mathcal{O}\left(\frac{\beta}{\sigma}n\text{poly}(\log T)\sqrt{T}\right),$$

by taking $\eta = \mathcal{O}\left(\frac{1}{nBH\log H\sqrt{T}}\right)$, with $\mathcal{O}(\cdot)$ hiding polynomials in $D, L, B$.

# D Proof of `EBPC` Regret Guarantee for Known Systems

This section proves the regret bound in Theorem 4.1 for the BCO-M based controller outlined in Algorithm 2. We will reduce the regret analysis of our proposed bandit LQR/LQG controller to that of BCO-M by designing with-history loss functions $F_t : \mathcal{M}(H, R)^H \to \mathbb{R}_+$ that well-approximates $c_t(\cdot, \cdot)$ for stable systems. In Section D.1, we provide the precise definitions of the with-history loss functions and proceed to check their regularity conditions as required by Theorem 3.6.B in Section D.2. In Section D.3, we analyze the regret of Algorithm 2 by bounding both the regret with respect to the with-history loss functions and the approximation error of the with-history loss functions to the true cost functions when evaluating on a single control policy parametrized by some $M \in \mathcal{M}(H, R)$.

## D.1 Construction of with-history loss functions

In the bandit control task using our proposed bandit controller outlined in Algorithm 2, there are two independent sources of noise: the gradient estimator $g_t$ used in Algorithm 2 and the perturbation sequence $\{(\mathbf{w}_t^{\text{stoch}}, \mathbf{e}_t^{\text{stoch}})\}_{t=1}^T$ injected to the partially observable linear dynamical system. Formally, we define the following filtrations generated by these two sources of noises.

**Definition D.1** (Noise filtrations). *For all $1 \leq t \leq T$, let $\mathcal{F}_t \overset{def}{=} \sigma(\{\varepsilon_s\}_{0 \leq s \leq t})$ be the filtration generated by the noises sampled to create the gradient estimator in the algorithm up to time $t$. Let $\mathcal{G}_t \overset{def}{=} \sigma(\{(\mathbf{w}_s^{\text{stoch}}, \mathbf{e}_s^{\text{stoch}})\}_{0 \leq s \leq t})$ be the filtration generated by the stochastic part of the semi-adversarial perturbation to the linear systems up till time $t$.*

The main insight in the analysis of online nonstochastic control algorithms is the reduction of the control problem to an online learning with memory problem. To this end, we construct the with-history loss functions as follows:

**Definition D.2** (With-history loss functions for known systems). *Given a Markov operator $G$ of a partially observable linear dynamical system and an incidental cost function $c_t : \mathbb{R}^{d_\mathbf{y}} \times \mathbb{R}^{d_\mathbf{u}} \to \mathbb{R}_+$ at time $t$, its corresponding with-history loss function at time $t$ is given a (random) function $F_t : \mathcal{M}(H, R)^H \to \mathbb{R}$ of the form*

$$F_t(N_1, \ldots, N_H) \overset{def}{=} c_t \left( \mathbf{y}_t^{\mathbf{nat}} + \sum_{i=1}^{\bar{H}} G^{[i]} \sum_{j=0}^{\bar{H}} N_{H-i}^{[j]} \mathbf{y}_{t-i-j}^{\mathbf{nat}} + \sum_{i=H}^{t} G^{[i]} \sum_{j=0}^{\bar{H}} \widetilde{M}_{t-i}^{[j]} \mathbf{y}_{t-i-j}^{\mathbf{nat}}, \sum_{j=0}^{\bar{H}} N_H^{[j]} \mathbf{y}_{t-j}^{\mathbf{nat}} \right).$$

*Additionally, denote the unary form $f_t : \mathcal{M}(H, R) \to \mathbb{R}_+$ induced by $F_t$ as $f_t(N) \overset{def}{=} F_t \underbrace{(N, \ldots, N)}_{N \text{ in all } H \text{ indices}}$.*

We immediately note a connection of the with-history loss functions constructed in Definition D.2 to the cost functions. Observe that by expression $\mathbf{y}_t, \mathbf{u}_t$ resulted from running Algorithm 2 explicitly,

$$c_t(\mathbf{y}_t, \mathbf{u}_t) = c_t \left( \mathbf{y}_t^{\mathbf{nat}} + \sum_{i=1}^{t} G^{[i]} \sum_{j=0}^{\bar{H}} \widetilde{M}_{t-i}^{[j]} \mathbf{y}_{t-i-j}^{\mathbf{nat}}, \sum_{j=0}^{\bar{H}} \widetilde{M}_t^{[j]} \mathbf{y}_{t-j}^{\mathbf{nat}} \right) = F_t(\widetilde{M}_{t-\bar{H}:t}).$$

**Remark D.3.** *Note that $\mathbf{y}_t^{\mathbf{nat}}$ is independent of $\mathcal{F}_T$. Therefore, by construction, $F_t$ is a $\mathcal{F}_{t-H} \cup \mathcal{G}_t$-measurable random function that is independent of $\varepsilon_{t-\bar{H}:t}$. In particular, Assumption 3.3 on the adversary is satisfied.*

It is left to check the regularity assumptions of $F_t$, which we defer to Section D.2.

## D.2 Regularity condition of with-history loss functions

The goal of this section is to establish the other conditions to apply the result of Theorem 3.6.B. The following table summarizes the results in this section.

| Parameter | Definition | Magnitude |
|---|---|---|
| $R_{\mathbf{y}}$ | $\ell_2$ bound on observations | $R_{\mathrm{nat}}(1 + RR_G)$ |
| $R_{\mathbf{u}}$ | $\ell_2$ bound on controls based on $\mathcal{M}(H,R)$ | $R_{\mathrm{nat}}R$ |
| $B$ | diameter bound on $c_t, F_t, f_t$ | $L_c R_{\mathrm{nat}}^2 ((1 + RR_G)^2 + R^2)$ |
| $D$ | diameter bound on $\mathcal{M}(H,R)$ | $2\sqrt{d_{\mathbf{u}} \wedge d_{\mathbf{y}}} R$ |
| $\sigma_f$ | conditional strong convexity parameter of $f_t$ | $\sigma_c \left( \sigma_{\mathbf{e}}^2 + \sigma_{\mathbf{w}}^2 \frac{\sigma_{\min}(C)}{1 + \|A\|_{\mathrm{op}}^2} \right)$ |
| $\beta_F$ | smoothness parameter of $F_t$ | $4\beta_c R_{\mathrm{nat}}^2 R_G^2 H$ |
| $L_F$ | Lipschitz parameter of $F_t$ | $2L_c \sqrt{(1 + RR_G)^2 + R^2} R_G R_{\mathrm{nat}}^2 \sqrt{H}$ |

We start with bounding $\ell_2$-norm on the observed signals $\mathbf{y}_t$ and controls $\mathbf{u}_t$ played by Algorithm 2.

**Lemma D.4** (Observation and control norm bounds)**.** *Denote $R_{\mathbf{y}} := \sup_t \|\mathbf{y}_t\|_2$ and $R_{\mathbf{u}} := \sup_t \|\mathbf{u}_t\|_2$. Then, the following bounds hold deterministically:*

$$R_{\mathbf{y}} \leq R_{\mathrm{nat}}(1 + RR_G), \quad R_{\mathbf{u}} \leq R_{\mathrm{nat}}R.$$

*Proof.* By algorithm specification, $\mathbf{y}_t, \mathbf{u}_t$ allow the following expansions:

$$\|\mathbf{u}_t\|_2 = \left\| \sum_{j=0}^{\bar{H}} \widetilde{M}_t^{[j]} \mathbf{y}_{t-j}^{\mathbf{nat}} \right\|_2 \leq \max_{0 \leq j \leq \bar{H}} \|\mathbf{y}_{t-j}^{\mathbf{nat}}\|_2 \left\| \sum_{j=0}^{\bar{H}} \widetilde{M}_t^{[j]} \right\|_{\mathrm{op}} \leq_1 R_{\mathrm{nat}}R,$$

$$\|\mathbf{y}_t\|_2 = \left\| \mathbf{y}_t^{\mathbf{nat}} + \sum_{i=1}^{t} G^{[i]} \mathbf{u}_{t-i} \right\|_2 \leq R_{\mathrm{nat}} + \max_{1 \leq i \leq t} \|\mathbf{u}_{t-i}\|_2 \left\| \sum_{i=1}^{t} G^{[i]} \right\|_{\mathrm{op}} \leq R_{\mathrm{nat}} + R_{\mathrm{nat}}RR_G,$$

where $\leq_1$ follows from $\widetilde{M}_t \in \mathcal{M}(H,R)$ for all $t$ by Remark 3.6. $\qquad\square$

**Lemma D.5** (Diameter bounds)**.** *Given a Markov operator $G$ of a stable partially observable linear dynamical system. Let $\mathcal{U} := \left\{ \sum_{j=0}^{\bar{H}} M^{[j]} \zeta_j : M \in \mathcal{M}(H,R), \zeta_j \in \mathbb{R}^{d_{\mathbf{y}}}, \|\zeta_j\|_2 \leq R_{\mathrm{nat}} \right\}$ and $\mathcal{Y} := \left\{ \zeta + \sum_{i=1}^{T-1} G^{[i]} \xi_i : \zeta \in \mathbb{R}^{d_{\mathbf{y}}}, \|\zeta\|_2 \leq R_{\mathrm{nat}}, \xi_i \in \mathcal{U} \right\}$. Denote $B = \sup_{\mathbf{y} \in \mathcal{Y}} \sup_{\mathbf{u} \in \mathcal{U}} \sup_{1 \leq t \leq T} c_t(\mathbf{y}, \mathbf{u})$. Denote $D = \sup_{M, M' \in \mathcal{M}(H,R)} \|M - M'\|_F$. Then,*

$$B \leq L_c R_{\mathrm{nat}}^2 ((1 + RR_G)^2 + R^2), \quad D \leq 2\sqrt{d_{\mathbf{u}} \wedge d_{\mathbf{y}}} R.$$

*Proof.* Recall the quadratic and Lipschitz assumption on $c_t$. $\forall \mathbf{y} \in \mathcal{Y}, \mathbf{u} \in \mathcal{U}, 1 \leq t \leq T$,

$$c_t(\mathbf{y}, \mathbf{u}) \leq L_c \|(\mathbf{y}, \mathbf{u})\|_2^2 = L_c(\|\mathbf{y}\|_2^2 + \|\mathbf{u}\|_2^2) \leq L_c R_{\mathrm{nat}}^2 ((1 + RR_G)^2 + R^2).$$

For any $M, M' \in \mathcal{M}(H,R)$, we have

$$\|M - M'\|_F \leq \sum_{j=0}^{\bar{H}} \|M^{[j]} - M'^{[j]}\|_F \leq \sqrt{d_{\mathbf{u}} \wedge d_{\mathbf{y}}} \sum_{j=0}^{\bar{H}} \|M^{[j]} - M'^{[j]}\|_{\mathrm{op}} \leq 2\sqrt{d_{\mathbf{u}} \wedge d_{\mathbf{y}}} R.$$

$$\square$$

In particular, Lemma D.5 implies the diameter bound for $c_t(\mathbf{y}_t, \mathbf{u}_t)$, $\forall t$, and $c_t(\mathbf{y}_t^M, \mathbf{u}_t^M)$, $\forall t, \forall M \in \mathcal{M}(H,R)$ as well as $F_t$ on $\mathcal{M}(H,R)^H$ and $f_t$ on $\mathcal{M}(H,R)$, $\forall t$. We proceed to check other regularity conditions for $F_t$ and $f_t$.

**Lemma D.6** (Regularity conditions of $F_t$ and $f_t$)**.** *Let $F_t$ and $f_t$ be given as in Definition D.2, and $G$ be the Markov operator of a partially observable linear dynamical system. $F_t$ and $f_t$ satisfy the following regularity conditions $\forall t$:*

- *The function $\mathbb{E}[f_t(\cdot) \mid \mathcal{F}_{t-H} \cup \mathcal{G}_{t-H}]$ defined on $\mathcal{M}(H,R)$ is $\sigma_f$-strongly convex with strong convexity parameter $\sigma_f = \sigma_c \left( \sigma_{\mathbf{e}}^2 + \sigma_{\mathbf{w}}^2 \frac{\sigma_{\min}(C)}{1 + \|A\|_{\mathrm{op}}^2} \right)$.*

- $F_t$ is quadratic and $\beta_F$-smooth with $\beta_F = 4\beta_c R_{\text{nat}}^2 R_G^2 H$.

- $F_t$ is $L_F$-Lipschitz with $L_F = 2L_c\sqrt{(1 + RR_G)^2 + R^2}R_G R_{\text{nat}}^2\sqrt{H}$.

*Proof.* First, we show the conditional strong convexity. Recall that $c_t$ is quadratic, therefore $c_t(\mathbf{y}_t, \mathbf{u}_t) = \mathbf{y}_t^\top Q_t \mathbf{y}_t + \mathbf{u}_t^\top R_t \mathbf{u}_t$. Consider the following quantities:

$$\mathbf{S}_M \stackrel{\text{def}}{=} \mathbf{y}_t^{\mathbf{nat}} + \sum_{i=1}^{\bar{H}} G^{[i]} \sum_{j=0}^{\bar{H}} M^{[j]}\mathbf{y}_{t-i-j}^{\mathbf{nat}}, \quad \mathbf{F} \stackrel{\text{def}}{=} \sum_{i=H}^{t} G^{[i]} \sum_{j=0}^{\bar{H}} \widetilde{M}_{t-i}^{[j]}\mathbf{y}_{t-i-j}^{\mathbf{nat}}, \mathbf{C}_M \stackrel{\text{def}}{=} \sum_{j=0}^{\bar{H}} M^{[j]}\mathbf{y}_{t-j}^{\mathbf{nat}}.$$

Note that $\mathbf{S}_M, \mathbf{C}_M$ are independent of $\mathcal{F}_{t-H}$ and $\mathbf{F} \in \mathcal{F}_{t-H} \cup \mathcal{G}_{t-H}$. Thus,

$$\mathbb{E}\left[f_t(M) \mid \mathcal{F}_{t-H} \cup \mathcal{G}_{t-H}\right] = \mathbb{E}\left[c_t(\mathbf{S}_M + \mathbf{F}, \mathbf{C}_M) \mid \mathcal{F}_{t-H} \cup \mathcal{G}_{t-H}\right]$$
$$= \mathbb{E}[\mathbf{S}_M^\top Q_t \mathbf{S}_M \mid \mathcal{G}_{t-H}] + \mathbf{F}^\top(Q_t + Q_t^\top)\mathbb{E}[\mathbf{S}_M \mid \mathcal{G}_{t-H}]$$
$$+ \mathbf{F}^\top Q_t \mathbf{F} + \mathbb{E}[\mathbf{C}_M^\top R_t \mathbf{C}_M \mid \mathcal{G}_{t-H}]$$
$$= \mathbb{E}[c_t(\mathbf{S}_M, \mathbf{C}_M) \mid \mathcal{G}_{t-H}] + \ell(M),$$

where $\ell(M) \stackrel{\text{def}}{=} \mathbf{F}^\top(Q_t + Q_t^\top)\mathbb{E}[\mathbf{S}_M \mid \mathcal{G}_{t-H}] + \mathbf{F}^\top Q_t \mathbf{F}$ is affine in $M$. The strong convexity of $\mathbb{E}[c_t(\mathbf{S}_M, \mathbf{C}_M) \mid \mathcal{G}_{t-H}]$ is established by the following lemma from Simchowitz et al. [2020]:

**Lemma D.7** (Lemma J.10 and Lemma J.15 in [Simchowitz et al., 2020]). $\forall M \in \mathcal{M}(H, R)$,

$$\mathbb{E}\left[\left\|(\mathbf{S}_M, \mathbf{C}_M) - (\mathbf{y}_t^{\mathbf{nat}}, \mathbf{0}_{d_{\mathbf{u}}})\right\|_2^2 \mid \mathcal{G}_{t-H}\right] \geq \left(\sigma_{\mathbf{e}}^2 + \sigma_{\mathbf{w}}^2 \frac{\sigma_{\min}(C)}{1 + \|A\|_{\text{op}}^2}\right)\|M\|_F^2.$$

The above lemma implies that $\mathbb{E}[c_t(\mathbf{S}_M, \mathbf{C}_M) \mid \mathcal{G}_{t-H}]$ is $\sigma_f$-strongly convex for $\sigma_f = \sigma_c\left(\sigma_{\mathbf{e}}^2 + \sigma_{\mathbf{w}}^2 \frac{\sigma_{\min}(C)}{1 + \|A\|_{\text{op}}^2}\right)$ on $\mathcal{M}(H, R)$.

By assumption, $c_t(\cdot, \cdot)$ is $\beta_c$-smooth.

$$F_t(N_1, \ldots, N_H) = (\mathbf{S}_{N_{1:H}} + \mathbf{F})^\top Q_t(\mathbf{S}_{N_{1:H}} + \mathbf{F}) + \mathbf{C}_{N_H}^\top R_t \mathbf{C}_{N_H},$$

where $\mathbf{S}_{N_{1:H}} \stackrel{\text{def}}{=} \mathbf{y}_t^{\mathbf{nat}} + \sum_{i=1}^{\bar{H}} G^{[i]} \sum_{j=0}^{\bar{H}} N_{H-i}^{[j]}\mathbf{y}_{t-i-j}^{\mathbf{nat}}$ and $\mathbf{C}_{N_H} \stackrel{\text{def}}{=} \sum_{j=0}^{\bar{H}} N_H^{[j]}\mathbf{y}_{t-j}^{\mathbf{nat}}$ are linear in $N_{1:H}$. $F_t$ is quadratic by the above expression. Moreover, $F_t$ is $\beta_F$-smooth if and only if $c_t(\mathbf{S}_{N_{1:H}}, \mathbf{C}_{N_H})$ is $\beta_F$-smooth as a function of $N_{1:H}$. We proceed to bound $\beta_F$. Consider the linear operator $v : \mathcal{M}(H, R)^H \to \mathbb{R}^{2d_{\mathbf{y}}}$ given by $v(N_{1:H}) = (\mathbf{S}_{N_{1:H}}, \mathbf{C}_{N_H})$. Then $\forall N_{1:H}, N'_{1:H} \subset \mathcal{M}(H, R)$,

$$\|v(N_{1:H}) - v(N'_{1:H})\|_2 = \left\|\left(\sum_{i=1}^{\bar{H}} G^{[i]} \sum_{j=0}^{\bar{H}} (N_{H-i}^{[j]} - N_{H-i}'^{[j]})\mathbf{y}_{t-i-j}^{\mathbf{nat}}, \sum_{j=0}^{\bar{H}} (N_H^{[j]} - N_H'^{[j]})\mathbf{y}_{t-j}^{\mathbf{nat}}\right)\right\|_2$$

$$\leq \left\|\sum_{i=1}^{\bar{H}} G^{[i]} \sum_{j=0}^{\bar{H}} (N_{H-i}^{[j]} - N_{H-i}'^{[j]})\mathbf{y}_{t-i-j}^{\mathbf{nat}}\right\|_2 + \left\|\sum_{j=0}^{\bar{H}} (N_H^{[j]} - N_H'^{[j]})\mathbf{y}_{t-j}^{\mathbf{nat}}\right\|_2$$

$$\leq \sum_{i=1}^{\bar{H}} \|G^{[i]}\|_{\text{op}} \left\|\sum_{j=0}^{\bar{H}} (N_{H-i}^{[j]} - N_{H-i}'^{[j]})\mathbf{y}_{t-i-j}^{\mathbf{nat}}\right\|_2 + \left\|\sum_{j=0}^{\bar{H}} (N_H^{[j]} - N_H'^{[j]})\mathbf{y}_{t-j}^{\mathbf{nat}}\right\|_2$$

$$\leq R_G \max_{1 \leq i \leq \bar{H}} \left\|\sum_{j=0}^{\bar{H}} (N_i^{[j]} - N_i'^{[j]})\mathbf{y}_{t-(H-i)-j}^{\mathbf{nat}}\right\|_2 + \left\|\sum_{j=0}^{\bar{H}} (N_H^{[j]} - N_H'^{[j]})\mathbf{y}_{t-j}^{\mathbf{nat}}\right\|_2$$

$$\leq 2R_G R_{\text{nat}} \max_{1 \leq i \leq H} \|N_i - N_i'\|_{\ell_1, \text{op}}$$

$$\leq 2R_G R_{\text{nat}}\sqrt{H}\|N_{1:H} - N'_{1:H}\|_F,$$

which bounds $\|\mathbf{D}v(N_{1:H})\|_2 \leq 2R_G R_{\text{nat}}\sqrt{H}$ and thus

$$\|\nabla^2(c_t(v(N_{1:H})))\|_{\text{op}} = \|\mathbf{D}v(N_{1:H})(\nabla^2 c_t)(v(N_{1:H}))\mathbf{D}v(N_{1:H})^\top\|_{\text{op}}$$
$$\leq \beta_c\|\mathbf{D}v(N_{1:H})\|_2^2$$
$$\leq 4\beta_c R_{\text{nat}}^2 R_G^2 H.$$

$F_t$ is $\beta_F \overset{\text{def}}{=} 4\beta_c R_{\text{nat}}^2 R_G^2 H$-smooth since $c_t(\mathbf{S}_{N_{1:H}}, \mathbf{C}_{N_H})$ is $4\beta_c R_{\text{nat}}^2 R_G^2 H$-smooth. It is left to bound the gradient for $F_t$. Note that

$$\begin{aligned}
\|\nabla F_t(N_1, \ldots, N_H)\|_2 &= \|(\nabla c_t)(v(N_{1:H}))\|_2 \|\mathbf{D}v(N_{1:H})\|_2 \\
&\leq 2L_c \sqrt{(1 + RR_G)^2 + R^2} R_G R_{\text{nat}}^2 \sqrt{H} \\
&= L_F.
\end{aligned}$$

$\square$

## D.3 Controller regret decomposition and analysis

Recall the definition of regret for the controller algorithm:

$$\begin{aligned}
\text{Regret}_T(\texttt{controller}) &= J_T(\texttt{controller}) - \inf_{M \in \mathcal{M}(H,R)} J_T(\pi_M) \\
&= \sum_{t=1}^{T} c_t(\mathbf{y}_t, \mathbf{u}_t) - \inf_{M \in \mathcal{M}(H,R)} \sum_{t=1}^{T} c_t(\mathbf{y}_t^M, \mathbf{u}_t^M),
\end{aligned}$$

where $\mathbf{u}_t$ is the control played by the controller algorithm at time $t$ and $\mathbf{y}_t$ is the observation attained by the algorithm's history of controls at time $t$. $(\mathbf{y}_t^M, \mathbf{u}_t^M)$ is the observation-control pair that would have been returned if the DRC policy $M$ were executed from the beginning of the time. The above regret can be decomposed in the following way.

$$\text{Regret}_T(\texttt{controller}) = \underbrace{\left( \sum_{t=1}^{2\bar{H}} c_t(\mathbf{y}_t, \mathbf{u}_t) \right)}_{\text{(burn-in loss)}} + \underbrace{\left( \sum_{t=2\bar{H}+1}^{T} F_t(\widetilde{M}_{t-\bar{H}:t}) - \inf_{M \in \mathcal{M}(H,R)} \sum_{t=2\bar{H}+1}^{T} f_t(M) \right)}_{\text{(effective BCO-M regret)}}$$

$$+ \underbrace{\left( \inf_{M \in \mathcal{M}(H,R)} \sum_{t=2\bar{H}+1}^{T} f_t(M) - \inf_{M \in \mathcal{M}(H,R)} \sum_{t=2\bar{H}+1}^{T} c_t(\mathbf{y}_t^M, \mathbf{u}_t^M) \right)}_{\text{(control truncation loss)}}$$

The first term is the loss incurred by the initialization stage of the algorithm. The second term entails the regret guarantee with respect to the with-history loss functions defined in Section D.1, which we bound by a combination of the result of Theorem 3.6.B and the regularity conditions established in Section D.2. The third term is a truncation loss of the comparator used in the regret analysis. In particular, $c_t(\cdot, \cdot)$ has history of length $t$, but the constructed $f_t$ only has history of length $H$. Therefore, each term in the summand of the first term in the control truncation loss measures the counterfactual cost at time $t$ had $M$ been used in constructing the control since $H$ steps back, while each term in the summand of the second term in the control truncation loss measures the counterfactual cost at time $t$ had $M$ been applied to construct the controls from the beginning of the time. The control truncation loss is bounded by the decaying behavior of stable systems, where effects of past controls decay exponentially over time.

We bound each term separately. First, the burn-in loss can be crudely bounded by the diameter bound $B$ of $c_t(\cdot, \cdot)$, which is established by Lemma D.5 in Section D.2 by the Lipschitz assumption of $c_t(\cdot, \cdot)$. In particular, applying the diameter bound and under the assumption that $H = \text{poly}(\log T)$,

$$\text{(burn-in loss)} \leq 2HB \leq 2HL_c R_{\text{nat}}^2 (R^2 + (1 + RR_G)^2) = \tilde{\mathcal{O}}(1),$$

Then, we bound the control truncation loss. By the decaying behavior of stable systems, $\psi_G(H) \leq O(T^{-1})$ for $H$ taken to be $\text{poly}(\log T)$.

$$(\text{control truncation loss}) \leq \sup_{M \in \mathcal{M}(H,R)} \left\{ \sum_{t=2\bar{H}+1}^{T} f_t(M) - c_t(\mathbf{y}_t^M, \mathbf{u}_t^M) \right\}$$

$$= \sup_{M \in \mathcal{M}(H,R)} \left\{ \sum_{t=2\bar{H}+1}^{T} c_t\left( \mathbf{y}_t^{\mathbf{nat}} + \sum_{i=1}^{\bar{H}} G^{[i]} \sum_{j=0}^{\bar{H}} M^{[j]} \mathbf{y}_{t-i-j}^{\mathbf{nat}} \right. \right.$$

$$\left. \left. + \sum_{i=H}^{t} G^{[i]} \sum_{j=0}^{\bar{H}} \widetilde{M}_{t-i}^{[j]} \mathbf{y}_{t-i-j}^{\mathbf{nat}}, \mathbf{u}_t^M \right) - c_t\left( \mathbf{y}_t^{\mathbf{nat}} + \sum_{i=1}^{\bar{H}} G^{[i]} \sum_{j=0}^{\bar{H}} M^{[j]} \mathbf{y}_{t-i-j}^{\mathbf{nat}}, \mathbf{u}_t^M \right) \right\}$$

$$\leq L_c B \sup_{M \in \mathcal{M}(H,R)} \sum_{t=2\bar{H}+1}^{T} \left\| \sum_{i=H}^{t} G^{[i]} \sum_{j=0}^{\bar{H}} (\widetilde{M}_{t-i}^{[j]} - M^{[j]}) \mathbf{y}_{t-i-j}^{\mathbf{nat}} \right\|_2$$

$$\leq L_c B T \psi_G(H) \cdot \sup_t \sup_{M \in \mathcal{M}(H,R)} \left\| \sum_{j=0}^{\bar{H}} (\widetilde{M}_{t-i}^{[j]} - M^{[j]}) \mathbf{y}_{t-i-j}^{\mathbf{nat}} \right\|_2$$

$$\leq 2 L_c B T \psi_G(H) R R_{\text{nat}}$$

$$\leq \mathcal{O}(1).$$

It is left to bound the effective BCO-M regret. By construction of $F_t$ in Section D.1 and algorithm specification of our proposed bandit controller in Algorithm 2, we are essentially running the `EBCO-M` algorithm with the sequence of loss functions $\{F_t\}_{t=H}^T$ on the constraint set $\mathcal{K} = \mathcal{M}(H, R)$. Note that further by the analysis in Section D.2 and Remark D.3 and substituting the parameters $\sigma = \sigma_f$, $\beta = \beta_F$, $L = L_F$, and diameter bounds $B, D$ defined in Lemma D.5, Theorem 3.6.B immediately implies that

$$\mathbb{E}[(\text{effective BCO-M regret})] \leq \tilde{\mathcal{O}}\left( \frac{\beta d_{\mathbf{u}} d_{\mathbf{y}}}{\sigma_c} \sqrt{T} \right),$$

since all the parameters for $F_t$ obtained in Section D.2 differ from the parameters of $c_t$ by factors of at most logarithmic in $T$. Putting together, the regret of the bandit controller is bounded by

$$\mathbb{E}[\text{Regret}_T(\texttt{controller})] \leq \tilde{\mathcal{O}}\left( \frac{\beta_c d_{\mathbf{u}} d_{\mathbf{y}}}{\sigma_c} \sqrt{T} \right).$$

# E  Proof of `EBPC` Regret Guarantee for Unknown Systems

When the system is unknown, we run an estimation algorithm outlined in Algorithm 3, followed by our proposed BCO-M based control algorithm with slightly modified parameters. In particular, to compare with the single best policy in the DRC policy class parametrized by $\mathcal{M}(H, R)$, we let $\mathcal{K} = \mathcal{M}(H^+, R^+)$, where $H^+ = 3H$ and $R^+ = 2R$ and set history parameter to be $H^+$. Subsequently, we denote $\overline{H^+} \stackrel{\text{def}}{=} H^+ - 1$. This section will be organized as the following: Section E.1 introduces a previously known error guarantee for the estimation algorithm outlined in Algorithm 3; Section E.2 defines the estimated with-history loss functions;

## E.1  System estimation error guarantee

When the system is unknown, we would need to first run a system estimation algorithm to obtain an estimator $\hat{G}$ for the Markov operator $G$, which we use as an input to our control algorithm outlined in Algorithm 2. It is known that the estimation algorithm we outlined in Algorithm 3 has high probability error guarantee in its estimated Markov operator, formally given by the following theorem.

**Theorem E.1** (Theorem 7, Simchowitz et al. [2020]). *With probability at least $1 - \delta - N^{-(\log N)^2}$, Algorithm 3 guarantees that with $\varepsilon_G(N, \delta) \asymp \frac{1}{\sqrt{N}} H^2 R_{\text{nat}} \sqrt{(d_{\mathbf{y}} \vee d_{\mathbf{u}}) + \log\left(\frac{1}{\delta}\right) + \log(1 + R_{\text{nat}})}$, the following inequalities hold:*

*1. $\|\mathbf{u}_t\|_2 \leq R_{\mathbf{u}, \delta} \stackrel{\text{def}}{=} 5\sqrt{d_{\mathbf{u}} + 2\log\left(\frac{3}{\delta}\right)}, \forall t \in [N]$.*

*2. $\|\hat{G} - G\|_{\ell_1, \text{op}} \leq \varepsilon_G(N, \delta) \leq \frac{1}{2\max\{RR_G, R_{\mathbf{u}, \delta}\}}$.*

**Remark E.2.** *Denote $E$ as the event where the two inequalities of Theorem E.1 hold. We are interested in the expected regret of our proposed bandit controller, which is*

$$\mathbb{E}[Regret_T(\texttt{controller}) \mid E]\mathbb{P}(E) + \mathbb{E}[Regret_T(\texttt{controller}) \mid E^C]\mathbb{P}(E^C)$$

$$\leq \mathbb{E}[Regret_T(\texttt{controller}) \mid E] + (\delta + N^{-(\log N)^2})\hat{B}T,$$

*where $\hat{B}$ denotes the bound on the cost $c_t$ when performing controls assuming $\hat{G}$ is the true Markov operator. We will show in Section E.4 that $\hat{B} \lesssim B$. Therefore, when $\delta \leq \frac{1}{\sqrt{T}}$ and $N \geq \sqrt{T}$, we have $(\delta + N^{-(\log N)^2})BT \leq \mathcal{O}(\sqrt{T})$. Therefore, from now on we make the following assumption:*

**Assumption E.3** (Estimation error). *The estimation sample size $N$ and error parameter $\delta$ are set to be $N = \lceil\sqrt{T}\rceil$ and $\delta = \frac{1}{\sqrt{T}}$. The estimated Markov operator $\hat{G}$ obtained from Algorithm 3 satisfies the following two inequalities with $\varepsilon_G \asymp \frac{1}{\sqrt{N}} H^2 R_{\text{nat}} \sqrt{(d_{\mathbf{y}} \vee d_{\mathbf{u}}) + \log\left(\frac{1}{\delta}\right) + \log(1 + R_{\text{nat}})}$:*

*1. $\|\mathbf{u}_t\|_2 \leq R_{\mathbf{u}, \delta}, \forall t \in [N]$.*

*2. $\|\hat{G} - G\|_{\ell_1, \text{op}} \leq \varepsilon_G \leq \frac{1}{2\max\{RR_G, R_{\mathbf{u}, \delta}\}}$.*

*Additionally, without loss of generality we assume that $\varepsilon_G \leq R_G$.*

## E.2  Construction of estimated with-history loss functions

Once we obtain $\hat{G}$ from Algorithm 3 for $N$ iterations, we invoke Algorithm 2 treating $\hat{G}$ as the input Markov operator on $\mathcal{K} = M(H^+, R^+)$ with history parameter $H^+$. In this case, the cost functions $c_t(\mathbf{y}_t, \mathbf{u}_t)$ evaluated by the $(\mathbf{y}_t, \mathbf{u}_t)$ resulted from playing Algorithm 2 allows the following two equivalent expressions:

$$c_t(\mathbf{y}_t, \mathbf{u}_t) = c_t\left(\mathbf{y}_t^{\mathbf{nat}} + \sum_{i=1}^{t} G^{[i]}\mathbf{u}_{t-i}, \mathbf{u}_t\right) = c_t\left(\mathbf{y}_t^{\mathbf{nat}} + \sum_{i=1}^{t} G^{[i]} \sum_{j=0}^{\overline{H^+}} \widetilde{M}_{t-i}^{[j]}\hat{\mathbf{y}}_{t-i-j}^{\mathbf{nat}}, \sum_{j=0}^{\overline{H^+}} \widetilde{M}_t^{[j]}\hat{\mathbf{y}}_{t-j}^{\mathbf{nat}}\right)$$

$$= c_t\left(\hat{\mathbf{y}}_t^{\mathbf{nat}} + \sum_{i=1}^{t} \hat{G}^{[i]}\mathbf{u}_{t-i}, \mathbf{u}_t\right) = c_t\left(\hat{\mathbf{y}}_t^{\mathbf{nat}} + \sum_{i=1}^{t} \hat{G}^{[i]} \sum_{j=0}^{\overline{H^+}} \widetilde{M}_{t-i}^{[j]}\hat{\mathbf{y}}_{t-i-j}^{\mathbf{nat}}, \sum_{j=0}^{\overline{H^+}} \widetilde{M}_t^{[j]}\hat{\mathbf{y}}_{t-j}^{\mathbf{nat}}\right),$$

where $\hat{\mathbf{y}}_t^{\mathbf{nat}}$ is the nature's $\mathbf{y}$ calculated by the algorithm at time $t$ using the estimated Markov operator $\hat{G}$. The last inequality follows from $\hat{G}^{[i]} = 0$ for $i \geq H^+$. We construct the two estimated with-history loss functions. First, we construct with-history loss functions analogous to the $F_t$ constructed in Section D.1 for the known system.

**Remark E.4.** *By specification in the bandit controller outlined in Algorithm 2, $\hat{\mathbf{y}}_t^{\mathbf{nat}}$ is obtained by the formula $\hat{\mathbf{y}}_t^{\mathbf{nat}} \leftarrow \mathbf{y}_t - \sum_{i=1}^{t-1} \hat{G}^{[i]} \mathbf{u}_{t-1-i}$, and thus $\hat{\mathbf{y}}_t^{\mathbf{nat}} \in \mathcal{F}_{t-H} \cup \mathcal{G}_t$ with the filtrations defined in Definition D.1.*

**Definition E.5** (With-history losses for unknown system). *Given an estimated Markov operator $\hat{G}$ of a partially observable linear dynamical system and an incidental cost function $c_t : \mathbb{R}^{d_{\mathbf{y}}} \times \mathbb{R}^{d_{\mathbf{u}}} \to \mathbb{R}_+$ at time $t$, define its with-history loss at time $t$ to be $\hat{F}_t : \mathcal{M}(H^+, R^+)^{H^+} \to \mathbb{R}_+$, given by*

$$\hat{F}_t(N_{1:H^+}) \stackrel{def}{=} c_t \left( \hat{\mathbf{y}}_t^{\mathbf{nat}} + \sum_{i=1}^{\overline{H^+}} \hat{G}^{[i]} \sum_{j=0}^{\overline{H^+}} N_{H^+-i}^{[j]} \hat{\mathbf{y}}_{t-i-j}^{\mathbf{nat}}, \sum_{j=0}^{\overline{H^+}} N_{H^+}^{[j]} \hat{\mathbf{y}}_{t-j}^{\mathbf{nat}} \right).$$

*Define $\hat{f}_t : \mathcal{M}(H, R) \to \mathbb{R}_+$ to be the unary form induced by $\hat{F}_t$, given by $\hat{f}_t(N) \stackrel{def}{=} \hat{F}_t \underbrace{(N, \dots, N)}_{N \text{ in all } H \text{ indices}}.$*

Note that $\hat{F}_t(\widetilde{M}_{t-\overline{H^+}:t}) = c_t(\mathbf{y}_t, \mathbf{u}_t)$. Moreover, $\hat{F}_t$ is a $\mathcal{F}_{t-H} \cup \mathcal{G}_t$-measurable random function by Remark E.4 that is independent of $\varepsilon_{t-\overline{H^+}:t}$. In particular, Assumption 3.3 is satisfied. In addition to the with-history losses, we introduce a new pseudo loss function $\mathring{F}_t : \mathcal{M}(H^+, R^+)^{H^+} \to \mathbb{R}_+$ as the following.

**Definition E.6** (With-history pseudo losses for unknown system). *Given a partially observable linear dynamical system with Markov operator $G$ and an incidental cost function $c_t : \mathbb{R}^{d_{\mathbf{y}}} \times \mathbb{R}^{d_{\mathbf{u}}} \to \mathbb{R}_+$ at time $t$. define its with-history pseudo loss at tiem $t$ to be $\mathring{F}_t : \mathcal{M}(H^+, R^+)^{H^+} \to \mathbb{R}_+$, given by*

$$\mathring{F}_t(N_{1:H^+}) \stackrel{def}{=} c_t \left( \mathbf{y}_t^{\mathbf{nat}} + \sum_{i=1}^{\overline{H^+}} G^{[i]} \sum_{j=0}^{\overline{H^+}} N_{H^+-i}^{[j]} \hat{\mathbf{y}}_{t-i-j}^{\mathbf{nat}}, \sum_{j=0}^{\overline{H^+}} N_{H^+}^{[j]} \hat{\mathbf{y}}_{t-j}^{\mathbf{nat}} \right).$$

*Define $\mathring{f}_t : \mathcal{M}(H, R) \to \mathbb{R}_+$ to be the unary form induced by $\mathring{F}_t$, given by $\mathring{f}_t(N) \stackrel{def}{=} \mathring{F}_t \underbrace{(N, \dots, N)}_{N \text{ in all } H \text{ indices}}.$*

While the learner has no access to $\mathring{F}_t$, it is useful for regret analysis: we will show that the gradient of $\mathring{f}_t$ is sufficiently close to the gradient of $\hat{f}_t$ and therefore the running bandit-RFTL-D on the loss functions $\{\hat{f}_t\}_{t=N+H^+}^T$ is nearly equivalent to running bandit RFTL-D with erroneous gradients on the loss functions $\{\mathring{f}_t\}_{t=N+H^+}^T$.

We analyze how error in the computed gradient affects the final regret guarantee in Section E.3. In Section E.4, we prove the regularity conditions needed for both Section E.3 and the downstream regret analysis performed in Section E.5, which gives our desired final regret bound.

## E.3 RFTL-D with erroneous gradients

We establish a regret guarantee for RFTL-with-delay (RFTL-D) with erroneous gradient against loss functions that are conditionally strongly convex and satisfying other regularity conditions stated below in Assumption E.7 and E.8. The proof follows similarly to that in Simchowitz et al. [2020], where they proved a similar regret guarantee for Online Gradient Descent (OGD). In particular, we establish that when run with conditionally strongly convex loss functions, (1) the error in gradient propagates quadratically in the regret bound, and (2) the regret bound has a negative movement cost term.

We begin with the working assumptions on the feasible set $\mathcal{K} \subset \mathbb{R}^d$ and the sequence of loss functions $\{f_t\}_{t=H}^T$ defined on $\mathcal{K}$.

**Assumption E.7** (Conditional strong convexity). *Let $\{f_t\}_{t=H}^T$ be a sequence of loss functions mapping from $\mathcal{K} \to \mathbb{R}$. Letting $\mathcal{H}_t$ be the filtration generated by algorithm history up till time $t$ for all $t \geq H$, assume that $f_{t;H}(\cdot) \stackrel{def}{=} \mathbb{E}[f_t(\cdot) \mid \mathcal{H}_{t-H}]$ is $\sigma$-strongly convex on $\mathcal{K}$.*

**Assumption E.8** (Diameter). *Assume that* $\mathrm{diam}(\mathcal{K}) = \sup_{z, z' \in \mathcal{K}} \|z - z'\|_2 \le D$. *Moreover, assume that* $\{f_t\}_{t=H}^T$ *obeys the range diameter bound* $\sup_{z, z' \in \mathcal{K}} |f_t(z) - f_t(z')| \le B$.

**Assumption E.9** (Gradient error). *Let* $\{\delta_t\}_{t=H}^T$ *denote the sequence of errors injected to the gradients.* $\{\delta_t\}_{t=H}^T$ *satisfies that for* $\tilde{\nabla}_t \stackrel{def}{=} \nabla f_t(z_t) + \delta_t$, *where* $z_t$ *is the algorithm's decision at time* $t$, $\|\tilde{\nabla}_t\|_{(t)} \le L_{\tilde{f}}$ *for some norm* $\|\cdot\|_{(t)}$ *with dual* $\|\cdot\|_{(t),*}$ *possibly varying with* $t$.

We consider RFTL-D run with erroneous gradients, outlined by Algorithm 5.

---

**Algorithm 5** RFTL-D with erroneous gradients

---

1: Input: feasible set $\mathcal{K} \subset \mathbb{R}^d$, time horizon $T$, history parameter $H$, strong convexity parameter $\sigma$, step size $\eta > 0$, regularization function $R(\cdot) : \mathcal{K} \to \mathbb{R}$.
2: Initialize: $\tilde{\nabla}_1 = \cdots = \tilde{\nabla}_{\bar{H}} = 0$, $z_1 = \cdots = z_H \in \mathcal{K}$.
3: **for** $t = H, \ldots, T$ **do**
4:  Play $z_t$, incur loss $f_t(z_t)$, receive gradient with error $\tilde{\nabla}_t = \nabla f_t(z_t) + \delta_t$.
5:  Update $z_{t+1} = \arg\min_{z \in \mathcal{K}} \left( \sum_{s=H}^t \left( \tilde{\nabla}_{s-\bar{H}}^\top z + \frac{\sigma}{4} \|z - z_{s-\bar{H}}\|_2^2 \right) + \frac{1}{\eta} R(z) \right)$.
6: **end for**

---

**Lemma E.10** (Conditional regret inequality for RFTL-D). *Under Assumption E.7, E.8, and E.9, let* $\Delta_t \stackrel{def}{=} \nabla f_t(z_t) - \nabla f_{t;H}(z_t)$ *denote the difference between the true gradient and the conditional gradient. Then we have that,* $\forall z \in \mathcal{K}$, $\{z_t\}_{t=H}^T$ *output by Algorithm 5 satisfies the following regret inequality:*

$$\sum_{t=H}^T f_{t;H}(z_t) - f_{t;H}(z) \le L_{\tilde{f}} \sum_{t=2\bar{H}+1}^T \|z_{t-\bar{H}} - z_{t+1}\|_{(t),*} + 2HB + \frac{R(z)}{\eta}$$

$$- \frac{\sigma}{4} \sum_{t=H}^{T-\bar{H}} \|z_t - z\|_2^2 - \sum_{t=H}^{T-\bar{H}} (\delta_t + \Delta_t)^\top (z_t - z).$$

*Proof.* Define $h_t(z) \stackrel{def}{=} \tilde{\nabla}_{t-\bar{H}}^\top z + \frac{\sigma}{4} \|z_{t-\bar{H}} - z\|_2^2$ for $t \ge 2\bar{H}+1$ and $\frac{1}{\eta} R(z)$ otherwise. By standard FTL-BTL lemma, $\sum_{t=2\bar{H}}^T h_t(z) \ge \sum_{t=2\bar{H}}^T h_t(z_{t+1})$, $\forall z \in \mathcal{K}$. Then

$$\sum_{t=H}^{T-\bar{H}} f_{t;H}(z_t) - f_{t;H}(z) \le \sum_{t=H}^{T-\bar{H}} \nabla f_{t;H}(z_t)^\top (z_t - z) - \frac{\sigma}{2} \sum_{t=H}^{T-\bar{H}} \|z_t - z\|_2^2$$

$$= \sum_{t=H}^{T-\bar{H}} \left( \tilde{\nabla}_t^\top (z_t - z) - \frac{\sigma}{2} \|z_t - z\|_2^2 \right) - \sum_{t=H}^{T-\bar{H}} (\delta_t + \Delta_t)^\top (z_t - z)$$

$$= \underbrace{\sum_{t=2\bar{H}+1}^T h_t(z_{t-\bar{H}}) - h_t(z)}_{(*)} - \sum_{t=H}^{T-\bar{H}} (\delta_t + \Delta_t)^\top (z_t - z) - \frac{\sigma}{4} \sum_{t=H}^{T-\bar{H}} \|z_t - z\|_2^2.$$

Applying the FTL-BTL lemma, the first part on the right hand side is bounded by

$$(*) \le \left( \sum_{t=2\bar{H}+1}^T h_t(z_{t-\bar{H}}) - h_t(z_{t+1}) \right) + \left( \sum_{t=\bar{H}}^{2\bar{H}} h_t(z_{t+1}) - h_t(z) \right)$$

$$\le \sum_{t=2\bar{H}+1}^T \underbrace{\nabla h_t(z_{t-\bar{H}})^\top}_{\tilde{\nabla}_{t-\bar{H}}} (z_{t-\bar{H}} - z_{t+1}) + HB + \frac{R(z)}{\eta}$$

$$\le L_{\tilde{f}} \sum_{t=2\bar{H}+1}^T \|z_{t-\bar{H}} - z_{t+1}\|_{(t),*} + HB + \frac{R(z)}{\eta}.$$

Combining,

$$\sum_{t=H}^{T} f_{t;H}(z_t) - f_{t;H}(z) \leq L_{\tilde{f}} \sum_{t=2\bar{H}+1}^{T} \|z_{t-\bar{H}} - z_{t+1}\|_{(t),*} + 2HB + \frac{R(z)}{\eta}$$
$$- \frac{\sigma}{4} \sum_{t=H}^{T-\bar{H}} \|z_t - z\|_2^2 - \sum_{t=H}^{T-\bar{H}} (\delta_t + \Delta_t)^\top (z_t - z).$$

$\square$

**Lemma E.11** (Regret inequality for bandit RFTL-D)**.** *Suppose RFTL-D is run with gradient estimators $g_t$ such that $g_t$ satisfies $\|\mathbb{E}[g_t \mid \mathcal{G}_t] - \mathbb{E}[\tilde{\nabla}_t \mid \mathcal{G}_t]\|_2 \leq B(t)$, where $\mathcal{G}_t$ is any filtration such that $z_t \in \mathcal{G}_t$, then $\forall z \in \mathcal{K}$,*

$$\mathbb{E}\left[ \sum_{t=2\bar{H}+1}^{T} f_t(z_t) - f_t(z) \right] \leq L_g \mathbb{E}\left[ \sum_{t=2\bar{H}+1}^{T} \|z_{t-\bar{H}} - z_{t+1}\|_{(t),*} \right] + 3HB + \frac{R(z)}{\eta}$$
$$- \frac{\sigma}{6} \mathbb{E}\left[ \sum_{t=H}^{T-\bar{H}} \|z_t - z\|_2^2 \right] + \frac{3}{\sigma} \mathbb{E}\left[ \sum_{t=H}^{T-\bar{H}} \|\delta_t\|_2^2 \right] + 2D \sum_{t=H}^{T-\bar{H}} B(t),$$

*where $L_g = \sup_t \|g_t\|_{(t)}$.*

*Proof.* Define $q_t(z) \overset{\text{def}}{=} f_t(z) + (g_t - \tilde{\nabla}_t)^\top z + \delta_t, \forall t \geq H$, and note that $\nabla q_t(z_t) = g_t$ by construction. Then since RFTL-D is a first-order OCO algorithm, we have RFTL-D$(q_H, \ldots, q_{t-1})$ = RFTL-D$(g_H, \ldots, g_{t-1}), \forall t$. Moreover, by Lemma 6.3.1 in Hazan [2016], $\forall z \in \mathcal{K}$, we have

$$\sum_{t=2\bar{H}+1}^{T} q_t(z_t) - q_t(z) \leq \text{Regret}_T^{\text{RFTL-D}}(g_H, \ldots, g_T) + \sum_{t=H}^{2\bar{H}} q_t(z) - q_t(z_t)$$
$$\leq \text{Regret}_T^{\text{RFTL-D}}(g_H, \ldots, g_T) + \sum_{t=H}^{2\bar{H}} f_t(z) - f_t(z_t)$$
$$\leq \text{Regret}_T^{\text{RFTL-D}}(g_H, \ldots, g_T) + HB,$$

where the second inequality follows from $\forall H \leq t \leq 2\bar{H}$, $g_t = \tilde{\nabla}_t = 0$ and thus $q_t(z) = f_t(z) + \delta_t$, $\forall z$. Additionally, $\forall t \geq 2\bar{H} + 1, z \in \mathcal{K}$,

$$\mathbb{E}[q_t(z_t) - q_t(z)] = \mathbb{E}[f_t(z_t) - f_t(z)] - \mathbb{E}[(g_t - \tilde{\nabla}_t)^\top (z_t - z)]$$
$$= \mathbb{E}[f_t(z_t) - f_t(z)] - \mathbb{E}[\mathbb{E}[(g_t - \tilde{\nabla}_t)^\top (z_t - z) \mid \mathcal{G}_t]]$$
$$= \mathbb{E}[f_t(z_t) - f_t(z)] - \mathbb{E}[(\mathbb{E}[g_t \mid \mathcal{G}_t] - \mathbb{E}[\tilde{\nabla}_t \mid \mathcal{G}_t])^\top (z_t - z)]$$
$$\geq \mathbb{E}[f_t(z_t) - f_t(z)] - DB(t).$$

Moreover, by Lemma E.10, $\forall z \in \mathcal{K}$, since

$$\mathbb{E}[\Delta_t^\top (z_t - z)] = \mathbb{E}[\mathbb{E}[\Delta_t^\top (z_t - z) \mid \mathcal{H}_{t-H}]] = \mathbb{E}[\mathbb{E}[\Delta_t \mid \mathcal{H}_{t-H}]^\top (z_t - z)] = 0,$$

the expected regret is bounded by

$$\mathbb{E}\left[ \text{Regret}_T^{\text{RFTL-D}}(g_1, \ldots, g_T) \right] \leq L_g \mathbb{E}\left[ \sum_{t=2\bar{H}+1}^{T} \|z_{t-\bar{H}} - z_{t+1}\|_{(t),*} \right] + 2HB + \frac{R(z)}{\eta}$$
$$- \frac{\sigma}{4} \mathbb{E}\left[ \sum_{t=H}^{T-\bar{H}} \|z_t - z\|_2^2 \right] - \underbrace{\mathbb{E}\left[ \sum_{t=H}^{T-\bar{H}} (g_t - \nabla f_t(z_t)))^\top (z_t - z) \right]}_{(*)},$$

where we can further decouple $(*)$ as

$$(*) = \underbrace{\mathbb{E}\left[\sum_{t=H}^{T-\bar{H}} (g_t - \tilde{\nabla}_t)^\top (z_t - z)\right]}_{(1)} + \underbrace{\mathbb{E}\left[\sum_{t=H}^{T-\bar{H}} \delta_t^\top (z_t - z)\right]}_{(2)},$$

$$(1) = \sum_{t=H}^{T-\bar{H}} (\mathbb{E}[g_t \mid \mathcal{G}_t] - \tilde{\nabla}_t)^\top (z_t - z) \le D \sum_{t=H}^{T-\bar{H}} B(t), \quad (2) \le \mathbb{E}\left[\sum_{t=H}^{T-\bar{H}} \frac{3}{\sigma}\|\delta_t\|_2^2 + \frac{\sigma}{12}\|z_t - z\|_2^2\right].$$

Combining, $\forall z \in \mathcal{K}$,

$$\mathbb{E}\left[\sum_{t=2\bar{H}+1}^{T} f_t(z_t) - f_t(z)\right] \le L_g \mathbb{E}\left[\sum_{t=2\bar{H}+1}^{T} \|z_{t-\bar{H}} - z_{t+1}\|_{(t),*}\right] + 3HB + \frac{R(z)}{\eta}$$
$$- \frac{\sigma}{6}\mathbb{E}\left[\sum_{t=H}^{T-\bar{H}} \|z_t - z\|_2^2\right] + \frac{3}{\sigma}\mathbb{E}\left[\sum_{t=H}^{T-\bar{H}} \|\delta_t\|_2^2\right] + 2D \sum_{t=H}^{T-\bar{H}} B(t).$$

$\square$

## E.4    Regularity conditions for estimated with-history loss functions and iterates

This section is analogous to Section D.2, and establishes regularity conditions for $\hat{F}_t, \mathring{F}_t, \mathcal{K} = \mathcal{M}(H^+, R^+)$.

The following table summarizes the results in this section.

| Parameter | Definition | Magnitude |
|---|---|---|
| $\hat{R}_{\text{nat}}$ | $\ell_2$ bound on the signals | $2R_{\text{nat}}$ |
| $R_{\hat{\mathbf{y}}}$ | $\ell_2$ bound on observations | $2R_{\text{nat}} + 4R_G \max\{R_{\mathbf{u},\delta}, RR_{\text{nat}}\}$ |
| $R_{\hat{\mathbf{u}}}$ | $\ell_2$ bound on controls based on $\mathcal{M}(H^+, R^+)$ | $2\max\{R_{\mathbf{u},\delta}, RR_{\text{nat}}\}$ |
| $\hat{B}$ | diameter bound on $c_t$ | $4L_c((R_{\text{nat}}^2 + 3R_G \max\{R_{\mathbf{u},\delta}, RR_{\text{nat}}\})^2)$ |
| $\hat{D}$ | diameter bound on $\mathcal{M}(H, R)$ | $2\sqrt{d_{\mathbf{u}} \wedge d_{\mathbf{y}}} R^+$ |
| $\sigma_{\mathring{f}}$ | conditional strong convexity parameter of $\mathring{f}_t$ | $\frac{\sigma_c}{4}\left(\sigma_{\mathbf{e}}^2 + \sigma_{\mathbf{w}}^2 \frac{\sigma_{\min}(C)}{1+\|A\|_{\text{op}}^2}\right)$ |
| $\sigma_{\hat{f}}$ | conditional strong convexity parameter of $\hat{f}_t$ | $\frac{\sigma_c}{4}\left(\sigma_{\mathbf{e}}^2 + \sigma_{\mathbf{w}}^2 \frac{\sigma_{\min}(C)}{1+\|A\|_{\text{op}}^2}\right)$ |
| $\beta_{\mathring{F}}$ | smoothness parameter of $\mathring{F}_t$ | $16\beta_c R_G^2 R_{\text{nat}}^2 H^+$ |
| $\beta_{\hat{F}}$ | smoothness parameter of $\hat{F}_t$ | $64\beta_c R_G^2 R_{\text{nat}}^2 H^+$ |
| $L_{\mathring{F}}$ | Lipschitz parameter of $\mathring{F}_t$ | $4L_c\sqrt{R_{\hat{\mathbf{y}}}^2 + R_{\hat{\mathbf{u}}}^2} R_{\text{nat}} R_G \sqrt{H^+}$ |
| $L_{\hat{F}}$ | Lipschitz parameter of $\hat{F}_t$ | $8L_c\sqrt{R_{\hat{\mathbf{y}}}^2 + R_{\hat{\mathbf{u}}}^2} R_{\text{nat}} R_G \sqrt{H^+}$ |

We start with proving $\ell_2$ bounds on the observations and controls.

**Lemma E.12** (Control, signal, and observation norm bounds for unknown systems). *Under Assumption E.3 on the obtained estimator $\hat{G}$ for the Markov operator and suppose the bandit controller outlined in Algorithm 2 is run with $\hat{G}$. Denote $\hat{R}_{\text{nat}} := \sup_t \|\mathbf{y}_t^{\text{nat}}\|_2$, $R_{\hat{\mathbf{y}}} := \sup_t \|\mathbf{y}_t\|_2$ and $R_{\hat{\mathbf{u}}} := \sup_t \|\mathbf{u}_t\|_2$, where $(\mathbf{y}_t, \mathbf{u}_t)$ are the observation-control pair resulted by executing the bandit controller, then the following bounds hold deterministically:*

$$\hat{R}_{\text{nat}} \le 2R_{\text{nat}}, \quad R_{\hat{\mathbf{y}}} \le 2R_{\text{nat}} + 4R_G \max\{R_{\mathbf{u},\delta}, RR_{\text{nat}}\}, \quad R_{\hat{\mathbf{u}}} \le 2\max\{R_{\mathbf{u},\delta}, RR_{\text{nat}}\},$$

*Proof.* By Assumption E.3, $\|\hat{G}\|_{\ell_1,\mathrm{op}} \leq 2R_G$ and $\forall t$,

$$\max_{s \leq t} \|\mathbf{u}_s\|_2 \leq \max\left\{ R_{\mathbf{u},\delta}, \max_{s \leq t} \left\| \sum_{i=0}^{\overline{H^+}} \widetilde{M}_s^{[i]} \hat{\mathbf{y}}_{s-i}^{\mathbf{nat}} \right\|_2 \right\}$$

$$\leq \max\left\{ R_{\mathbf{u},\delta}, R \max_{s \leq t} \max_{0 \leq i \leq \overline{H^+}} \|\hat{\mathbf{y}}_{s-i}^{\mathbf{nat}}\|_2 \right\}$$

$$\leq \max\left\{ R_{\mathbf{u},\delta}, R \left( R_{\mathrm{nat}} + \max_{s \leq t} \max_{0 \leq i \leq \overline{H^+}} \|\hat{\mathbf{y}}_{s-i}^{\mathbf{nat}} - \mathbf{y}_{s-i}^{\mathbf{nat}}\|_2 \right) \right\}$$

$$\leq \max\left\{ R_{\mathbf{u},\delta}, R \left( R_{\mathrm{nat}} + \max_{s \leq t} \max_{0 \leq i \leq \overline{H^+}} \left\| \sum_{j=1}^{s-i} (G^{[i]} - \hat{G}^{[i]}) \mathbf{u}_{s-i-j} \right\|_2 \right) \right\}$$

$$\leq \max\left\{ R_{\mathbf{u},\delta}, R \left( R_{\mathrm{nat}} + \varepsilon_G \max_{s \leq t-1} \|\mathbf{u}_s\|_2 \right) \right\}$$

$$\leq \max\{R_{\mathbf{u},\delta}, RR_{\mathrm{nat}}\} + \frac{\max_{s \leq t} \|\mathbf{u}_s\|_2}{2},$$

where the last inequality follows from $\varepsilon_G \leq \frac{1}{2\max\{RR_G, R_{\mathbf{u},\delta}\}}$ in Assumption E.3. The above inequality implies $R_{\hat{\mathbf{u}}} \leq 2\max\{R_{\mathbf{u},\delta}, RR_{\mathrm{nat}}\}$. Immediately, $\forall t$,

$$\|\hat{\mathbf{y}}_t^{\mathbf{nat}}\|_2 \leq R_{\mathrm{nat}} + \|\hat{\mathbf{y}}_t^{\mathbf{nat}} - \mathbf{y}_t^{\mathbf{nat}}\|_2 = R_{\mathrm{nat}} + \left\| \sum_{j=1}^{t} (G^{[i]} - \hat{G}^{[i]}) \mathbf{u}_{t-j} \right\|_2 \leq R_{\mathrm{nat}} + \varepsilon_G R_{\hat{\mathbf{u}}} \leq 2R_{\mathrm{nat}},$$

$$\|\mathbf{y}_t\|_2 = \left\| \hat{\mathbf{y}}_t^{\mathbf{nat}} + \sum_{i=1}^{t} \hat{G}^{[i]} \mathbf{u}_{t-i} \right\|_2 \leq 2R_{\mathrm{nat}} + 2R_G R_{\hat{\mathbf{u}}} \leq 2R_{\mathrm{nat}} + 4R_G \max\{R_{\mathbf{u},\delta}, RR_{\mathrm{nat}}\}.$$

$\square$

**Lemma E.13** (Diameter bounds). *Consider the following sets*

$$\hat{\mathcal{U}} \stackrel{def}{=} \left\{ \sum_{j=0}^{\overline{H^+}} M^{[j]} \zeta_j : M \in \mathcal{M}(H^+, R^+), \zeta_j \in \mathbb{R}^{d_{\mathbf{y}}}, \|\zeta_j\|_2 \leq \hat{R}_{\mathrm{nat}} \right\},$$

$$\hat{\mathcal{Y}} \stackrel{def}{=} \left\{ \zeta + \sum_{i=1}^{T-1} G^{[i]} \xi_i : \zeta \in \mathbb{R}^{d_{\mathbf{y}}}, \|\zeta\|_2 \leq \hat{R}_{\mathrm{nat}}, \xi_i \in \hat{\mathcal{U}} \right\},$$

*and* $\hat{B} := \sup_t \sup_{\mathbf{y} \in \hat{\mathcal{Y}}, \mathbf{u} \in \hat{\mathcal{U}}} c_t(\mathbf{y}, \mathbf{u})$. *Let* $\hat{D} := \sup_{M, M' \in \mathcal{M}(H^+, R^+)} \|M - M'\|_F$. *Then,*

$$\hat{B} \leq 4L_c((R_{\mathrm{nat}}^2 + 3R_G \max\{R_{\mathbf{u},\delta}, RR_{\mathrm{nat}}\})^2), \quad \hat{D} \leq 2\sqrt{d_{\mathbf{u}} \wedge d_{\mathbf{y}}} R^+.$$

*Proof.* First, we calculate the bound on $\hat{D}$. $\forall M, M' \in \mathcal{M}(H^+, R^+)$,

$$\|M - M'\|_F \leq \sum_{j=0}^{\overline{H^+}} \|M^{[j]} - M'^{[j]}\|_F \leq \sqrt{d_{\mathbf{u}} \wedge d_{\mathbf{y}}} \sum_{j=0}^{\overline{H^+}} \|M^{[j]} - M'^{[j]}\|_{\mathrm{op}} \leq 2\sqrt{d_{\mathbf{u}} \wedge d_{\mathbf{y}}} R^+.$$

To see the bound on $\hat{B}$, note that $\forall t, \forall \mathbf{y} \in \hat{\mathcal{Y}}, \mathbf{u} \in \hat{\mathcal{U}}$, by the quadratic and Lipschitz condition on $c_t$,

$$c_t(\mathbf{y}, \mathbf{u}) \leq L_c(\|\mathbf{y}\|_2^2 + \|\mathbf{u}\|_2^2) \leq 4L_c((R_{\mathrm{nat}}^2 + 3R_G \max\{R_{\mathbf{u},\delta}, RR_{\mathrm{nat}}\})^2).$$

$\square$

**Lemma E.14** (Regularity conditions for $\mathring{F}_t, \mathring{f}_t$ and $\hat{F}_t, \hat{f}_t$). *$\mathring{F}_t, \mathring{f}_t$ and $\hat{F}_t, \hat{f}_t$ follow the following regularity conditions under the assumption that* $\varepsilon_G \leq \frac{1}{4R_G R_{\hat{\mathbf{u}}} \sqrt{H^+}} \sqrt{\frac{\sigma_f}{\sigma_c}}$,

- $\mathring{F}_t$ is $L_{\mathring{F}}$-Lipschitz with $L_{\mathring{F}} = L_c\sqrt{R_{\hat{\mathbf{y}}}^2 + R_{\hat{\mathbf{u}}}^2}(4R_{\mathrm{nat}}R_G\sqrt{H^+})$; $\hat{F}_t$ is $L_{\hat{F}}$-Lipschitz with $L_{\hat{F}} = L_c\sqrt{R_{\hat{\mathbf{y}}}^2 + R_{\hat{\mathbf{u}}}^2}(8R_{\mathrm{nat}}R_G\sqrt{H^+})$.

- $\mathring{F}_t$ is $\beta_{\mathring{F}}$-smooth with $\beta_{\mathring{F}} = 16\beta_c R_G^2 R_{\mathrm{nat}}^2 H^+$; $\hat{F}_t$ is $\beta_{\hat{F}}$-smooth with $\beta_{\hat{F}} = 64\beta_c R_G^2 R_{\mathrm{nat}}^2 H^+$.

- $\mathring{f}_t, \hat{f}_t$ are $\sigma_{\mathring{f}}, \sigma_{\hat{f}}$-conditionally strongly convex with $\sigma_{\mathring{f}} = \sigma_{\hat{f}} = \frac{\sigma_f}{4}$.

*Proof.* Consider the following quantities:

$$\mathring{\mathbf{S}}_{N_{1:H^+}} \stackrel{\mathrm{def}}{=} \mathbf{y}_t^{\mathbf{nat}} + \sum_{i=1}^{\overline{H^+}} G^{[i]} \sum_{j=0}^{\overline{H^+}} N_{H^+-i}^{[j]} \hat{\mathbf{y}}_{t-i-j}^{\mathbf{nat}}, \quad \mathring{\mathbf{C}}_{N_{H^+}} = \sum_{j=0}^{\overline{H^+}} N_{H^+}^{[j]} \hat{\mathbf{y}}_{t-j}^{\mathbf{nat}},$$

$$\hat{\mathbf{S}}_{N_{1:H^+}} \stackrel{\mathrm{def}}{=} \hat{\mathbf{y}}_t^{\mathbf{nat}} + \sum_{i=1}^{\overline{H^+}} \hat{G}^{[i]} \sum_{j=0}^{\overline{H^+}} N_{H^+-i}^{[j]} \hat{\mathbf{y}}_{t-i-j}^{\mathbf{nat}}, \quad \hat{\mathbf{C}}_{N_{H^+}} = \mathring{\mathbf{C}}_{N_{H^+}}.$$

Consider the linear operator $\mathring{v}, \hat{v} : \mathcal{M}(H^+, R)^{H^+} \to \mathbb{R}^{d_{\mathbf{y}}} \times \mathbb{R}^{d_{\mathbf{u}}}$ given by $\mathring{v}(N_{1:H^+}) = (\mathring{\mathbf{S}}_{N_{1:H^+}}, \mathring{\mathbf{C}}_{N_{H^+}})$, $\hat{v}(N_{1:H^+}) = (\hat{\mathbf{S}}_{N_{1:H^+}}, \hat{\mathbf{C}}_{N_{H^+}})$. Similar to the analysis in Section D.2, $\forall N_{1:H^+}, N'_{1:H^+} \in \mathcal{M}(H^+, R^+)^{H^+}$,

$$\|\mathring{v}(N_{1:H^+}) - \mathring{v}(N'_{1:H^+})\|_2 = \left\| \left( \sum_{i=1}^{\overline{H^+}} G^{[i]} \sum_{j=0}^{\bar{H}} (N_{H^+-i}^{[j]} - N'^{[j]}_{H^+-i})\hat{\mathbf{y}}_{t-i-j}^{\mathbf{nat}}, \sum_{j=0}^{\overline{H^+}} (N_{H^+}^{[j]} - N'^{[j]}_{H^+})\hat{\mathbf{y}}_{t-j}^{\mathbf{nat}} \right) \right\|_2$$

$$\leq \left\| \sum_{i=1}^{\overline{H^+}} G^{[i]} \sum_{j=0}^{\overline{H^+}} (N_{H^+-i}^{[j]} - N'^{[j]}_{H^+-i})\hat{\mathbf{y}}_{t-i-j}^{\mathbf{nat}} \right\|_2 + \left\| \sum_{j=0}^{\overline{H^+}} (N_{H^+}^{[j]} - N'^{[j]}_{H^+})\hat{\mathbf{y}}_{t-j}^{\mathbf{nat}} \right\|_2$$

$$\leq \sum_{i=1}^{\overline{H^+}} \|G^{[i]}\|_{\mathrm{op}} \left\| \sum_{j=0}^{\overline{H^+}} (N_{H^+-i}^{[j]} - N'^{[j]}_{H^+-i})\hat{\mathbf{y}}_{t-i-j}^{\mathbf{nat}} \right\|_2 + \left\| \sum_{j=0}^{\overline{H^+}} (N_{H^+}^{[j]} - N'^{[j]}_{H^+})\hat{\mathbf{y}}_{t-j}^{\mathbf{nat}} \right\|_2$$

$$\leq R_G \max_{1 \leq i \leq H^+} \left\| \sum_{j=0}^{\overline{H^+}} (N_i^{[j]} - N'^{[j]}_i)\hat{\mathbf{y}}_{t-i-j}^{\mathbf{nat}} \right\|_2 + \left\| \sum_{j=0}^{\overline{H^+}} (N_{H^+}^{[j]} - N'^{[j]}_{H^+})\hat{\mathbf{y}}_{t-j}^{\mathbf{nat}} \right\|_2$$

$$\leq 4R_G R_{\mathrm{nat}} \max_{1 \leq i \leq H^+} \|N_i - N'_i\|_{\ell_1,\mathrm{op}}$$

$$\leq 4R_G R_{\mathrm{nat}} \sqrt{H^+} \|N_{1:H^+} - N'_{1:H^+}\|_F,$$

which bounds $\|\mathbf{D}\mathring{v}(N_{1:H^+})\|_2 \leq 4R_G R_{\mathrm{nat}}\sqrt{H^+}$. Similarly, we can bound $\|\mathbf{D}\hat{v}(N_{1:H^+})\|_2 \leq 8R_G R_{\mathrm{nat}}\sqrt{H^+}, \forall N_{1:H^+} \in \mathcal{M}(H^+, R^+)^{H^+}$. The gradient bounds $L_{\mathring{F}}, L_{\hat{F}}$ are thus given by

$$\|\nabla \mathring{F}_t(N_1, \ldots, N_{H^+})\|_2 = \|(\nabla c_t)(\mathring{v}(N_{1:H^+}))\|_2 \|\mathbf{D}\mathring{v}(N_{1:H^+})\|_2 \leq L_c\sqrt{R_{\hat{\mathbf{y}}}^2 + R_{\hat{\mathbf{u}}}^2}(4R_{\mathrm{nat}}R_G\sqrt{H^+}),$$

$$\|\nabla \hat{F}_t(N_1, \ldots, N_{H^+})\|_2 = \|(\nabla c_t)(\hat{v}(N_{1:H^+}))\|_2 \|\mathbf{D}\hat{v}(N_{1:H^+})\|_2 \leq L_c\sqrt{R_{\hat{\mathbf{y}}}^2 + R_{\hat{\mathbf{u}}}^2}(8R_{\mathrm{nat}}R_G\sqrt{H^+}).$$

The smoothness parameters $\beta_{\mathring{F}}, \beta_{\hat{F}}$ is given by

$$\|\nabla^2 c_t(\mathring{v}(N_{1:H^+}))\|_{\mathrm{op}} = \|\mathbf{D}\mathring{v}(N_{1:H^+})(\nabla^2 c_t)(\mathring{v}(N_{1:H^+}))\mathbf{D}\mathring{v}(N_{1:H^+})^\top\|_{\mathrm{op}} \leq 16\beta_c R_G^2 R_{\mathrm{nat}}^2 H^+,$$

$$\|\nabla^2 c_t(\hat{v}(N_{1:H^+}))\|_{\mathrm{op}} = \|\mathbf{D}\hat{v}(N_{1:H^+})(\nabla^2 c_t)(\hat{v}(N_{1:H^+}))\mathbf{D}\hat{v}(N_{1:H^+})^\top\|_{\mathrm{op}} \leq 64\beta_c R_G^2 R_{\mathrm{nat}}^2 H^+.$$

To bound the conditional strong convexity parameters $\sigma_{\mathring{f}}, \sigma_{\hat{f}}$, it suffices to show an analogue to Lemma D.7 that $\forall M \in \mathcal{M}(H^+, R^+)$,

$$\underbrace{\mathbb{E}\left[\|(\mathring{\mathbf{S}}_M, \mathring{\mathbf{C}}_M) - (\mathbf{y}_t^{\mathbf{nat}}, \mathbf{0}_{d_{\mathbf{u}}})\|_2^2 \mid \mathcal{F}_{t-H} \cup \mathcal{G}_{t-H}\right]}_{(1)} \geq \frac{\sigma_{\mathring{f}}}{\sigma_c}\|M\|_F^2,$$

$$\underbrace{\mathbb{E}\left[\|(\hat{\mathbf{S}}_M, \hat{\mathbf{C}}_M) - (\mathbf{y}_t^{\mathbf{nat}}, \mathbf{0}_{d_{\mathbf{u}}})\|_2^2 \mid \mathcal{F}_{t-H} \cup \mathcal{G}_{t-H}\right]}_{(2)} \geq \frac{\sigma_{\hat{f}}}{\sigma_c}\|M\|_F^2.$$

As $(a-b)^2 \geq \frac{1}{2}a^2 - b^2, \forall a, b \in \mathbb{R}$,

$$(1) = \mathbb{E}\left[\left\|\left(\sum_{i=1}^{\overline{H^+}} G^{[i]} \sum_{j=0}^{\overline{H^+}} M^{[j]}\hat{\mathbf{y}}_{t-i-j}^{\mathbf{nat}}, \sum_{j=0}^{\overline{H^+}} M^{[j]}\hat{\mathbf{y}}_{t-j}^{\mathbf{nat}}\right)\right\|_2^2 \middle| \mathcal{F}_{t-H} \cup \mathcal{G}_{t-H}\right]$$

$$\geq -\mathbb{E}\left[\left\|\left(\sum_{i=1}^{\overline{H^+}} G^{[i]} \sum_{j=0}^{\overline{H^+}} M^{[j]}(\hat{\mathbf{y}}_{t-i-j}^{\mathbf{nat}} - \mathbf{y}_{t-i-j}^{\mathbf{nat}}), \sum_{j=0}^{\overline{H^+}} M^{[j]}(\hat{\mathbf{y}}_{t-j}^{\mathbf{nat}} - \mathbf{y}_{t-j}^{\mathbf{nat}})\right)\right\|_2^2 \middle| \mathcal{F}_{t-H} \cup \mathcal{G}_{t-H}\right]$$

$$+ \frac{1}{2}\mathbb{E}\left[\left\|\left(\sum_{i=1}^{\overline{H^+}} G^{[i]} \sum_{j=0}^{\overline{H^+}} M^{[j]}\mathbf{y}_{t-i-j}^{\mathbf{nat}}, \sum_{j=0}^{\overline{H^+}} M^{[j]}\mathbf{y}_{t-j}^{\mathbf{nat}}\right)\right\|_2^2 \middle| \mathcal{F}_{t-H} \cup \mathcal{G}_{t-H}\right]$$

$$\geq \left(\frac{\sigma_f}{2\sigma_c} - R_G^2 \varepsilon_G^2 R_{\hat{\mathbf{u}}}^2 H^+\right)\|M\|_F^2$$

$$\geq \frac{\sigma_f}{4\sigma_c}\|M\|_F^2,$$

and similarly $(2) \geq \left(\frac{\sigma_f}{2\sigma_c} - 4R_G^2 \varepsilon_G^2 R_{\mathbf{u}}^2 H^+\right) \geq \frac{\sigma_f}{4\sigma_c}\|M\|_F^2.$ $\qquad\square$

## E.5 Unknown system regret analysis

Before the decomposition of regret, we introduce a result from Simchowitz et al. [2020]. Define $\varphi : \mathcal{M}(H, R) \to \mathbb{R}^{d_{\mathbf{y}} \times d_{\mathbf{u}} \times H^+}$ such that $\varphi(M) = (M, \mathbf{0}_{d_{\mathbf{y}} \times d_{\mathbf{u}}}, \dots, \mathbf{0}_{d_{\mathbf{y}} \times d_{\mathbf{u}}})$. Note that $\varphi(\mathcal{M}(H, R)) \subset \mathcal{M}(H^+, R^+)$.

**Proposition E.15** (Proposition F.8 in Simchowitz et al. [2020]). *$\exists M_0 \in \mathcal{M}(H, R)$ such that $\forall \tau > 0$,*

$$\sum_{t=N+2\overline{H^+}+1}^{T} \mathring{f}_t(\varphi(M_0)) - \inf_{M \in \mathcal{M}(H, R)} \sum_{t=N+2\overline{H^+}+1}^{T} f_t(M \mid G, \mathbf{y}_{1:t}^{\mathbf{nat}})$$

$$\leq 36(H^+)^2 R_G^4 R_{\mathrm{nat}}^4 (R^+)^3 (H^+ + T\varepsilon_G^2)\left(L_c \vee \frac{L_c^2}{\tau}\right) + \tau \sum_{t=N+2\overline{H^+}+1}^{T} \|M_t - \varphi(M_0)\|_F^2,$$

*where $f_t(\cdot \mid G, \mathbf{y}_{1:t}^{\mathbf{nat}}) : \mathcal{M}(H, R) \to \mathbb{R}_+$ is given by*

$$f_t(M \mid G, \mathbf{y}_{1:t}^{\mathbf{nat}}) \stackrel{def}{=} c_t\left(\mathbf{y}_t^{\mathbf{nat}} + \sum_{i=1}^{H} G^{[i]} \sum_{j=0}^{\bar{H}} M^{[j]}\mathbf{y}_{t-i-j}^{\mathbf{nat}}, \sum_{j=0}^{\bar{H}} M^{[j]}\mathbf{y}_{t-j}^{\mathbf{nat}}\right).$$

Let $M_0 \in \mathcal{M}(H, R)$ satisfy the inequality in Proposition E.15, and consider the decomposition of regret into four parts, which we will proceed to bound each separately.

$$\text{Regret}_T = \underbrace{\left(\sum_{t=1}^{N+2\bar{H}} c_t(\mathbf{y}_t, \mathbf{u}_t)\right)}_{\text{(burn-in loss)}} + \underbrace{\left(\sum_{t=N+2\overline{H^+}+1}^{T} c_t(\mathbf{y}_t, \mathbf{u}_t) - \mathring{F}_t(\widetilde{M}_{t-\overline{H^+}:t})\right)}_{\text{(algorithm estimation loss)}}$$

$$+ \underbrace{\left(\sum_{t=N+2\overline{H^+}+1}^{T} \mathring{F}_t(\widetilde{M}_{t-\overline{H^+}:t}) - \mathring{f}_t(\varphi(M_0))\right)}_{(\mathring{f}_t\text{-BCO-M-regret})}$$

$$+ \underbrace{\left(\sum_{t=N+2\overline{H^+}+1}^{T} \mathring{f}_t(\varphi(M_0)) - \inf_{M \in \mathcal{M}(H,R)} \sum_{t=N+2\overline{H^+}+1}^{T} c_t(\mathbf{y}_t^M, \mathbf{u}_t^M)\right)}_{\text{(comparator estimation loss)}}.$$

The choice of $M_0$ and Proposition E.15 directly allows us to bound the comparator estimation loss. In particular, note that

$$\inf_{M \in \mathcal{M}(H,R)} \sum_{t=N+2\overline{H^+}+1}^{T} f_t(M \mid G, \mathbf{y}_{1:t}^{\mathbf{nat}}) - \inf_{M \in \mathcal{M}(H,R)} \sum_{t=N+2\overline{H^+}+1}^{T} c_t(\mathbf{y}_t^M, \mathbf{u}_t^M)$$

$$\leq \sup_{M \in \mathcal{M}(H,R)} \sum_{t=N+2\overline{H^+}+1}^{T} f_t(M \mid G, \mathbf{y}_{1:t}^{\mathbf{nat}}) - c_t(\mathbf{y}_t^M, \mathbf{u}_t^M)$$

$$\leq \sup_{M \in \mathcal{M}(H,R)} \sum_{t=N+2\overline{H^+}+1}^{T} c_t\left(\mathbf{y}_t^{\mathbf{nat}} + \sum_{i=1}^{H} G^{[i]} \sum_{j=0}^{\bar{H}} M^{[j]} \mathbf{y}_{t-i-j}^{\mathbf{nat}}, \sum_{j=0}^{\bar{H}} M^{[j]} \mathbf{y}_{t-j}^{\mathbf{nat}}\right) -$$

$$c_t\left(\mathbf{y}_t^{\mathbf{nat}} + \sum_{i=1}^{t} G^{[i]} \sum_{j=0}^{\bar{H}} M^{[j]} \mathbf{y}_{t-i-j}^{\mathbf{nat}}, \sum_{j=0}^{\bar{H}} M^{[j]} \mathbf{y}_{t-j}^{\mathbf{nat}}\right)$$

$$\leq L_c \sqrt{(1 + R_G R)^2 + R^2 R_{\text{nat}}^2} R \psi_G(H) T \leq \mathcal{O}(1).$$

Therefore, combining terms and taking $H = \text{poly}(\log T)$, we have that for some constants $C_{\text{param}}^0, C_{\text{param}}^1$ depending on the natural parameters and universal constants $C$, $\forall \tau > 0$,

$$\text{(comparator estimation loss)} - \tau \sum_{t=N+2\overline{H^+}+1}^{T} \|M_t - M_0\|_F^2 \leq \tilde{\mathcal{O}}(1)\frac{\varepsilon_G T}{\tau} + \tilde{\mathcal{O}}(1)$$

$$\leq \frac{1}{\tau}\tilde{\mathcal{O}}(\sqrt{T}) + \tilde{\mathcal{O}}(1),$$

where the last inequality comes from taking $N = \lceil \sqrt{T} \rceil$, $\delta = \frac{1}{\sqrt{T}}$ as in Assumption E.3, and $\varepsilon_G \asymp \frac{1}{\sqrt{N}} H^2 R_{\text{nat}} \sqrt{(d_{\mathbf{y}} \vee d_{\mathbf{u}}) + \log\frac{1}{\delta} + \log(1 + R_{\text{nat}})}$ as in Proposition E.15.

Then, we proceed to bound the burn-in loss and the algorithm estimation loss. The burn-in loss can be crudely bounded by the diameter bound on $c_t$ established in Section E.4. Take $N = \lceil \sqrt{T} \rceil$,

$$\text{(burn-in loss)} \leq (N + 2\overline{H^+})\hat{B} \leq \mathcal{O}(\sqrt{T}) + \tilde{\mathcal{O}}(1).$$

The algorithm estimation loss can be bounded as follows:

$$
\begin{aligned}
\text{(algorithm estimation loss)} &\leq L_c\sqrt{R_{\hat{\mathbf{y}}}^2 + R_{\hat{\mathbf{u}}}^2} \sum_{t=N+2\overline{H^+}+1}^{T} \left\| \sum_{i=H}^{t} G^{[i]} \sum_{j=0}^{\overline{H^+}} \widetilde{M}_{t-i}^{[j]} \hat{\mathbf{y}}_{t-i-j}^{\mathbf{nat}} \right\|_2 \\
&\leq L_c\sqrt{R_{\hat{\mathbf{y}}}^2 + R_{\hat{\mathbf{u}}}^2} R_{\hat{\mathbf{u}}} \psi_G(H) T \\
&\leq \tilde{\mathcal{O}}(1).
\end{aligned}
$$

It is left to bound the $\mathring{f}_t$-BCO-M regret term, which is given by the following lemma:

**Proposition E.16.** *The BCO-M regret against the estimated with-history unary functions $\hat{f}_t$ has the following bound in expectation:*

$$
\mathbb{E}[(\mathring{f}_t\text{-}BCO\text{-}M\text{-}regret)] \leq \tilde{\mathcal{O}}(\sqrt{T}) + \frac{1}{\sigma_{\mathring{f}}}\tilde{\mathcal{O}}(\sqrt{T}) - \frac{\sigma_{\mathring{f}}}{6}\mathbb{E}\left[ \sum_{t=N+2\overline{H^+}+1}^{T} \|M_t - \varphi(M_0)\|_F^2 \right].
$$

*Proof.* First, we decompose the regret with respect to $\hat{F}_t$ into two parts:

$$
\underbrace{\left( \sum_{t=N+2\overline{H^+}+1}^{T} \mathring{F}_t(\widetilde{M}_{t-\bar{H}:t}) - \mathring{f}_t(M_t) \right)}_{\text{(estimation + movement cost)}} + \underbrace{\left( \sum_{t=N+2\overline{H^+}+1}^{T} \mathring{f}_t(M_t) - \sum_{t=N+2\overline{H^+}+1}^{T} \mathring{f}_t(\varphi(M_0)) \right)}_{\text{(bandit RFTL-D regret with erroneous gradient)}}.
$$

The movement cost is bounded similarly as in the analysis of Algorithm 1. In particular,

$$
\begin{aligned}
&\mathbb{E}\left[ \sum_{t=N+2\overline{H^+}+1}^{T} \mathring{F}_t(\widetilde{M}_{t-\bar{H}:t}) - \mathring{F}_t(M_{t-\bar{H}:t}) \right] \\
&\leq \sum_{t=N+2\overline{H^+}+1}^{T} \mathbb{E}\left[ \mathbb{E}\left[ \langle \nabla\mathring{F}_t(M_{t-\overline{H^+}:t}), A_{t-\overline{H^+}:t}\varepsilon_{t-\overline{H^+}:t} \rangle + \frac{\beta_{\mathring{F}}}{2} \sum_{s=t-\overline{H^+}}^{t} \|A_s\varepsilon_s\|_2^2 \mid \mathcal{F}_{t-H^+} \cup \mathcal{G}_{t-H^+} \right] \right] \\
&\leq \frac{\beta_{\mathring{F}}}{2} \sum_{t=N+2\overline{H^+}+1}^{T} \sum_{s=t-\overline{H^+}}^{t} \|A_s^2\|_{\text{op}} \leq \frac{\beta_{\mathring{F}}}{\eta\sigma_{\mathring{f}}} \sum_{t=N+2\overline{H^+}+1}^{T} \sum_{s=t-\overline{H^+}}^{t} \frac{1}{s} \leq \frac{\beta_{\mathring{F}} H^+ \log T}{\eta\sigma_{\mathring{f}}} \leq \tilde{\mathcal{O}}\left( \frac{\beta_c}{\sigma_c}\sqrt{T} \right),
\end{aligned}
$$

and

$$
\begin{aligned}
\mathbb{E}\left[ \sum_{t=N+2\overline{H^+}+1}^{T} \mathring{F}_t(M_{t-\overline{H^+}:t}) - \mathring{f}_t(M_t) \right] &\leq L_{\mathring{F}} \sum_{t=N+2\overline{H^+}+1}^{T} \|M_{t-\overline{H^+}:t} - (M_t, \ldots, M_t)\|_F \\
&\leq \frac{16\sqrt{\eta}d_{\mathbf{y}}d_{\mathbf{u}}L_{\mathring{F}}\hat{B}(H^+)^3}{\sqrt{\sigma_{\mathring{f}}}} \sum_{t=N+2\overline{H^+}+1}^{T} \frac{1}{\sqrt{t-2\overline{H^+}}} \\
&\leq \frac{16\sqrt{\eta T}d_{\mathbf{y}}d_{\mathbf{u}}L_{\mathring{F}}\hat{B}(H^+)^3}{\sqrt{\sigma_{\mathring{f}}}}.
\end{aligned}
$$

Combining, we have a bound on the estimation and movement cost:

$$
\text{(estimation + movement cost)} \leq \tilde{\mathcal{O}}\left( \frac{\beta_c}{\sigma_c}d_{\mathbf{y}}d_{\mathbf{u}}\sqrt{T} \right).
$$

To bound the second term, we first bound the gradient error $\|\nabla\hat{f}_t(M_t) - \nabla\mathring{f}_t(M_t)\|_F$, which measures the gradient error in using $\hat{f}_t$ to approximate $\mathring{f}_t$ in the algorithm.

**Lemma E.17** (Gradient error in estimating pseudo-losses)**.** *Let $\hat{f}_t$ and $\mathring{f}_t$ be given as in Definition E.5 and E.6. Let $M_t \in \mathcal{M}(H^+, R^+)$ played by Algorithm 2. Then, $\forall t$,*

$$
\left\| \nabla\mathring{f}_t(M_t) - \nabla\hat{f}_t(M_t) \right\|_F \leq \tilde{\mathcal{O}}(1)\varepsilon_G.
$$

*Proof.* Consider the function $\hat{v}(\cdot \mid G) : \mathcal{M}(H^+, R^+) \to \mathbb{R}^{d_\mathbf{y}+d_\mathbf{u}}$ parametrized by $G$ given by $\hat{v}(N \mid G) = (\hat{\mathbf{S}}_{N,G}, \hat{\mathbf{C}}_N)$, where $\hat{\mathbf{S}}_{N,G} = \mathbf{y}_t^\mathbf{nat} + \sum_{i=1}^{\bar{H}} G^{[i]} \sum_{j=0}^{\overline{H^+}} N^{[j]} \hat{\mathbf{y}}_{t-i-j}^\mathbf{nat}$ and $\hat{\mathbf{C}}_N = \sum_{j=0}^{\overline{H^+}} N^{[j]} \hat{\mathbf{y}}_{t-j}^\mathbf{nat}$. Then the gradient difference can be decomposed as

$$\nabla \mathring{f}_t(M_t) - \nabla \hat{f}_t(M_t) = \mathbf{D}\hat{v}(M_t \mid \hat{G}) \cdot (\nabla c_t)(\hat{\mathbf{S}}_{M_t,\hat{G}}, \hat{\mathbf{C}}_{M_t}) - \mathbf{D}\hat{v}(M_t \mid G) \cdot (\nabla c_t)(\hat{\mathbf{S}}_{M_t,G}, \hat{\mathbf{C}}_{M_t})$$

$$= \underbrace{\mathbf{D}\hat{v}(M_t \mid \hat{G}) \cdot ((\nabla c_t)(\hat{\mathbf{S}}_{M_t,\hat{G}}, \hat{\mathbf{C}}_{M_t})) - (\nabla c_t)(\hat{\mathbf{S}}_{M_t,G}, \hat{\mathbf{C}}_{M_t})))}_{(1)}$$

$$+ \underbrace{(\mathbf{D}\hat{v}(M_t \mid \hat{G}) - \mathbf{D}\hat{v}(M_t \mid G)) \cdot (\nabla c_t)(\hat{\mathbf{S}}_{M_t,G}, \hat{\mathbf{C}}_{M_t})}_{(2)},$$

where

$$(1) \leq \|\mathbf{D}\hat{v}(M_t \mid \hat{G})\|_2 \beta_c \left\| \sum_{i=H}^{t} (G^{[i]} - \hat{G}^{[i]}) \sum_{j=0}^{\overline{H^+}} M_t^{[j]} \hat{\mathbf{y}}_{t-i-j}^\mathbf{nat} \right\|_2 \leq \|\mathbf{D}\hat{v}(M_t \mid \hat{G})\|_2 \varepsilon_G R_{\hat{\mathbf{u}}},$$

$$(2) \leq \|\mathbf{D}\hat{v}(M_t \mid \hat{G}) - \mathbf{D}\hat{v}(M_t \mid G)\|_2 L_c \sqrt{R_{\hat{\mathbf{y}}}^2 + R_{\hat{\mathbf{u}}}^2},$$

where $R_{\hat{\mathbf{y}}}, R_{\hat{\mathbf{u}}}$ are established in Lemma E.12. We may further bound $\|\mathbf{D}\hat{v}(M \mid \hat{G})\|_2 \leq 4 R_G R_\mathrm{nat} \sqrt{H^+}$ and $\|\mathbf{D}\hat{v}(M \mid \hat{G} - G)\|_2 \leq 4\varepsilon_G R_\mathrm{nat} \sqrt{H^+}$ by identical analysis as in Lemma E.14. Combining, we have established the bound on the gradient difference between the true and pseudo-loss functions:

$$\left\| \nabla \hat{f}_t(M_t) - \nabla \mathring{f}_t(M_t) \right\|_F \leq \tilde{\mathcal{O}}(1)\varepsilon_G.$$

$\square$

With Lemma E.17, we are ready to establish the following corollary to Lemma E.11 that gives the regret inequality with respect to $\mathring{f}_t$:

**Corollary E.18** (Pseudo-loss regret inequality). *Let $L_g = \sup_t \|g_t\|_{t,t+1}^*$, where $g_t$ is the gradient estimator used in the bandit controller outlined in Algorithm 2. Then the following regret inequality holds:*

$$\mathbb{E}\left[ \sum_{t=N+2\overline{H^+}+1}^{T} \mathring{f}_t(M_t) - \mathring{f}_t(\varphi(M_0)) \right] \leq L_g \mathbb{E}\left[ \sum_{t=N+2\overline{H^+}+1}^{T} \left\| M_{t-\overline{H^+}} - M_{t+1} \right\|_{t,t+1} \right] + 3H^+ \hat{B}$$

$$+ \frac{\nu \log(T)}{\eta} - \frac{\sigma_{\hat{f}}}{6} \mathbb{E}\left[ \sum_{t=N+H^+}^{T-\overline{H^+}} \|M_t - M\|_F^2 \right]$$

$$+ \frac{3}{\sigma_{\hat{f}}} \tilde{\mathcal{O}}(1)\varepsilon_G^2 T + \frac{32\sqrt{\eta T}\beta_{\hat{F}} d_\mathbf{u} d_\mathbf{y} \hat{B}\hat{D}(H^+)^4}{\sqrt{\sigma_{\hat{f}}}}.$$

*Proof.* By Proposition C.2, the gradient estimator $g_t$ constructed in Algorithm 2 satisfies the following bias guarantee $\forall t \geq N + 2\overline{H^+} + 1$,

$$\left\| \mathbb{E}[g_t \mid \mathcal{F}_{t-H^+} \cup \mathcal{G}_{t-H^+}] - \mathbb{E}[\nabla \hat{f}_t(M_t) \mid \mathcal{F}_{t-H^+} \cup \mathcal{G}_{t-H^+}] \right\|_2 \leq \frac{16\sqrt{\eta}\beta_{\hat{f}} d_\mathbf{u} d_\mathbf{y} \hat{B}(H^+)^3}{\sqrt{\sigma_{\hat{f}}(t-N-2\overline{H^+})}}. \quad \text{(E.1)}$$

In the setting of Section E.3, take $\delta_t = \nabla \hat{f}_t(M_t) - \nabla \mathring{f}_t(M_t)$. The gradient estimator $g_t$ used in Algorithm 2 obeys $\|\mathbb{E}[g_t \mid \mathcal{F}_{t-H} \cup \mathcal{G}_{t-H}] - \mathbb{E}[\nabla \hat{f}_t(M_t) \mid \mathcal{F}_{t-H} \cup \mathcal{G}_{t-H}]\|_F \leq B(t)$ with $B(t) = \frac{16\sqrt{\eta}\beta_{\hat{f}} d_\mathbf{u} d_\mathbf{y} \hat{B}(H^+)^3}{\sqrt{\sigma_{\hat{f}}(t-N-2\overline{H^+})}}$ as given by E.1. Take $\|\cdot\|_{(t)} = \|\cdot\|_{t,t+1}^*$ and $\|\cdot\|_{(t),*} = \|\cdot\|_{t,t+1}$,

$B = \hat{B}$, $H = H^+$, $\sigma = \sigma_{\mathring{f}}$, Lemma E.11 and Lemma E.17 imply

$$\mathbb{E}\left[\sum_{t=N+2\overline{H^+}+1}^{T} \mathring{f}_t(M_t) - \mathring{f}_t(\varphi(M_0))\right] \leq L_g \mathbb{E}\left[\sum_{t=N+2\overline{H^+}+1}^{T} \left\|M_{t-\overline{H^+}} - M_{t+1}\right\|_{t,t+1}\right] + 3H^+\hat{B}$$

$$+ \frac{\nu \log(T)}{\eta} - \frac{\sigma_{\mathring{f}}}{6}\mathbb{E}\left[\sum_{t=N+H^+}^{T-\overline{H^+}} \|M_t - \varphi(M_0)\|_F^2\right]$$

$$+ \frac{3}{\sigma_{\mathring{f}}}\tilde{\mathcal{O}}(1)\varepsilon_G^2 T + \frac{32\sqrt{\eta T}\beta_{\hat{F}}d_{\mathbf{u}}d_{\mathbf{y}}\hat{B}\hat{D}(H^+)^4}{\sqrt{\sigma_{\mathring{f}}}}.$$

$\square$

We proceed to establish a local norm bound for $H$-steps-apart iterates. Let $\Phi : \mathcal{M}(H^+, R^+)^{H^+} \to \mathbb{R}$ be given by $\Phi_t(M) \overset{\text{def}}{=} \eta\left(\sum_{s=N+H^+}^{t}\langle g_{s-\overline{H^+}}, M\rangle + \frac{\sigma_{\mathring{f}}}{4}\|M - M_{s-\overline{H^+}}\|_F^2\right) + R(M)$. Recall that $R_t(M) \overset{\text{def}}{=} R(M) + \frac{\eta\sigma_{\mathring{f}}}{4}\sum_{s=N+H^+}^{t}\|M - M_{s-\overline{H^+}}\|_F^2$. By definition of Algorithm 2, the optimality condition, and linearity of $\Phi_t - R_t$, we have

$$\frac{1}{2}\|M_{t-\overline{H^+}} - M_{t+1}\|_{t,t+1}^2 = D_{R_t}(M_{t-\overline{H^+}}, M_{t+1})$$

$$\leq \Phi_t(M_{t-\overline{H^+}}) - \Phi_t(M_{t+1})$$

$$= \underbrace{(\Phi_{t-H^+}(M_{t-\overline{H^+}}) - \Phi_{t-H^+}(M_{t+1}))}_{\leq 0} + \eta\sum_{s=t-\overline{H^+}}^{t}\langle g_{s-\overline{H^+}}, M_{t-\overline{H^+}} - M_{t+1}\rangle$$

$$+ \frac{\eta\sigma_{\mathring{f}}}{4}\sum_{s=t-\overline{H^+}}^{t}\|M_{t-\overline{H^+}} - M_{s-\overline{H^+}}\|_F^2 - \|M_{t+1} - M_{s-\overline{H^+}}\|_F^2$$

$$\leq \eta\|M_{t-\overline{H^+}} - M_{t+1}\|_{t,t+1}\left\|\sum_{s=t-\overline{H^+}}^{t} g_{s-\overline{H^+}}\right\|_{t,t+1}^* + \frac{\eta\sigma_{\mathring{f}}\hat{D}^2 H^+}{4},$$

which implies

$$\|M_{t-\overline{H^+}} - M_{t+1}\|_{t,t+1} \leq \max\left\{4\eta\left\|\sum_{s=t-\overline{H^+}}^{t} g_{s-\overline{H^+}}\right\|_{t,t+1}^*, 2\sqrt{\eta\sigma_{\mathring{f}}H^+}\hat{D}\right\}.$$

Lemma C.6 established that $\forall t - \overline{H^+} \leq s \leq t$, $\|g_{s-\overline{H^+}}\|_{t,t+1}^* \leq 8d_{\mathbf{u}}d_{\mathbf{y}}\hat{B}H^{+4}$ and $\|g_t\|_{t,t+1}^* \leq 4d_{\mathbf{u}}d_{\mathbf{y}}\hat{B}H^{+4}$ hold deterministically. Plugging $L_g = 4d_{\mathbf{u}}d_{\mathbf{y}}\hat{B}H^{+4}$ and the iterate bounds into the bound obtained in Corollary E.18 and take step size $\eta = \mathcal{O}(\frac{1}{d_{\mathbf{y}}d_{\mathbf{u}}\hat{B}H^3\sqrt{T}})$, we have

$$\mathbb{E}[(\mathring{f}_t\text{-BCO-M-regret})] \leq \tilde{\mathcal{O}}(\sqrt{T}) + \frac{1}{\sigma_{\mathring{f}}}\tilde{\mathcal{O}}(1)\varepsilon_G^2 T - \frac{\sigma_{\mathring{f}}}{6}\mathbb{E}\left[\sum_{t=N+H^+}^{T-\overline{H^+}}\|M_t - \varphi(M_0)\|_F^2\right]$$

$$= \tilde{\mathcal{O}}(\sqrt{T}) + \frac{1}{\sigma_{\mathring{f}}}\tilde{\mathcal{O}}(\sqrt{T}) - \frac{\sigma_{\mathring{f}}}{6}\mathbb{E}\left[\sum_{t=N+2\overline{H^+}+1}^{T}\|M_t - \varphi(M_0)\|_F^2\right].$$

$\square$

Combining the bounds on burn-in loss, algorithm estimation loss, $\mathring{f}_t$-BCO-M regret, and comparator estimation loss and taking $\tau = \frac{\sigma_{\mathring{f}}}{6}$,

$$\mathbb{E}\left[\text{Regret}_T(\texttt{controller})\right] \leq \underbrace{\tilde{\mathcal{O}}(\sqrt{T})}_{\text{(burn-in loss)}} + \underbrace{\tilde{\mathcal{O}}(1)}_{\text{(algorithm estimation loss)}} +$$

$$\underbrace{\left(\tilde{\mathcal{O}}(\sqrt{T}) + \frac{1}{\sigma_{\mathring{f}}}\tilde{\mathcal{O}}(\sqrt{T}) - \frac{\sigma_{\mathring{f}}}{6}\mathbb{E}\left[\sum_{t=N+2\overline{H^+}+1}^{T} \|M_t - \varphi(M_0)\|_F^2\right]\right)}_{(\mathring{f}_t\text{-BCO-M-regret})} +$$

$$\underbrace{\left(\tilde{\mathcal{O}}\left(\frac{\beta_c}{\sigma_c}d_{\mathbf{y}}d_{\mathbf{u}}\sqrt{T}\right) + \frac{1}{\tau}\tilde{\mathcal{O}}(\sqrt{T}) + \tau\mathbb{E}\left[\sum_{t=N+2\overline{H^+}+1}^{T} \|M_t - \varphi(M_0)\|_F^2\right]\right)}_{\text{(comparator estimation loss)}}$$

$$\leq \tilde{\mathcal{O}}\left(\frac{\beta_c}{\sigma_c}d_{\mathbf{y}}d_{\mathbf{u}}\sqrt{T}\right).$$