# OpenReview forum: "Optimal Rates for Bandit Nonstochastic Control"
_NeurIPS.cc/2023/Conference — NeurIPS 2023 poster_

### Official Review · Reviewer_FiLK · 2023-07-03

**Soundness:** 3 good
**Presentation:** 3 good
**Contribution:** 3 good
**Rating:** 5
**Confidence:** 1

**Summary:**

The authors present an algorithm fro the LQG problem, in which a linear system has to be controlled in the presence of semi-adversarial perturbations in order to minimise an adversarially-chosen cost, with the controller only having access to a projection of the real dynamics onto a lower-dimensionality subspace. The authors show that their proposed method achieves optimal regret in time horizon of $O(\sqrt{T})$.

**Strengths:**

The paper's introduction of the problem is quite readable to a non-specialist, though specialist knowledge is required to understand most of the details and the theorem proof.

**Weaknesses:**

The paper is extremely theoretical and deals with a very specialised problem. This problem has connections with more mainstream AI problems (Optimal Control and Reinforcement Learning are closely related), and the paper does present it in a way that makes it understandable to a non-specialist reader. Still, the paper does not present any task-oriented AI-based framework involving the training of a Neural Net. While this might not be enough to disqualify the paper from publication at NeurIPS, it does severely restrict the number of potential readers.

**Questions:**

My understanding of the paper is too limited to provide any meaningful suggestions for improvement.

**Limitations:**

The paper is extremely theoretical and can only be understood by a very specialised audience. The relevance of the author's solution for the presented problem (or even of the problem itself) to more mainstream AI research, if any, is not at all discussed.

Edit: In their rebuttal, the authors challenged my assessment of their work's limitations by pointing out a connection between nonstochastic control and hyperparameter tuning in large-scale optimization, which was remarked in “A Nonstochastic Control Approach to Optimization” by Chen & Hazan (2023). While the connection is definitely present and fascinating, I still felt that it was not enough of an established research track for me to change my assessment. I therefore kept my initial score.

---

> ### Author Rebuttal · Authors · 2023-08-07
>
> Thank you for taking the time to read and review our paper. We tried to address your concerns in the following paragraph:
>
> Weakness:
>
> **FiLK**: The paper is extremely theoretical and deals with a very specialised problem. This problem has connections with more mainstream AI problems (Optimal Control and Reinforcement Learning are closely related), and the paper does present it in a way that makes it understandable to a non-specialist reader. Still, the paper does not present any task-oriented AI-based framework involving the training of a Neural Net. While this might not be enough to disqualify the paper from publication at NeurIPS, it does severely restrict the number of potential readers.
>
> **Response**: Thank you for your feedback. We agree that this is indeed a theoretical work, but the recent emerging line of work, as cited in our paper, in online nonstochastic control makes NeurIPS an appropriate venue to submit our theoretical work. Therefore, we respectfully disagree with the reviewer’s comment about the lack of relevance of our work to “more mainstream AI research”.
>
> There is a direct connection between online control and training deep neural networks, as per the recent work “A Nonstochastic Control Approach to Optimization” by Chen & Hazan (2023), which establishes a connection between nonstochastic control and hyperparameter tuning in large-scale optimization.This establishes a connection between control, RL and optimization for deep learning, which renders the study of online control relevant to mainstream AI research. We will mention this in our paper, but feel that direct implementation of this recent advancement is out of scope (as it is the topic of another (almost) concurrent paper).
>
> In general, we want to emphasize that (1) this is a work, although theoretical, on the topic that has become increasingly popular in the online learning community in the recent years, and (2) our work has the potential to be applied to DNN related research, but since it is out of the scope of our current paper, we believe that our current experiments are sufficient for supporting the theory. If this changes your perception of our work, please consider raising your score. Thank you!

---

> > ### Comment · Reviewer_FiLK · 2023-08-19
> > **Acknowledgement of rebuttal**
> >
> > I thank the authors for their rebuttal and their response to my only question about their paper.
> > I think that the single reference that the authors point out as an argument for the relevance of their results is not quite enough to change my assessment; also, and perhaps more importantly, the authors did not add this reference and the underlying context to a revision of the paper itself, which would have been beneficial to the readers.
> > I therefore shall keep my score unchanged.

---

### Official Review · Reviewer_iuZo · 2023-07-06

**Soundness:** 3 good
**Presentation:** 2 fair
**Contribution:** 3 good
**Rating:** 7
**Confidence:** 4

**Summary:**

This paper studies LQG with semi-adversarial perturbations and time varying adversarial bandit loss functions.
The best known bound for such problems has a T^{3/4} time horizon dependence.
They provide an algorithm for LQR/LQG problems that attains $\tilde{\mathcal{O}}(\sqrt{T})$ regret.
Their algorithm goes through even if the underlying system dynamics are unknown.
Additionally, their method also provides a new scheme for bandit convex optimization with memory.


**Strengths:**

This paper addresses an important problem in the field of control and online learning.

**Weaknesses:**


I must express my concerns about its presentation and structure. The writing is not as clear as it needs to be, making it quite challenging to follow the paper.
The argument seems to lack the rigorousness and the self-containment.

A crucial element that seems to be missing is detailed explanations for the conclusions drawn. Without these, it becomes arduous to justify the results. Enhancing the clarity and providing in-depth explanations of the methodologies and results will significantly improve the comprehension and overall impact of the study. I list my comments, questions and suggestions below.


**Questions:**


## Major comments:

1) Can you elaborate more on the DRC policy class? To me it seems like you are concentrating on the causal (non-anticipatory) policies that are linear. Is it possible to show that a linear policy will be optimal for your setting because you are concentrating on the quadratic cost function? I believe that would require to show that the estimation of the state given observations would be linear in observations as in the classical LQG setting.

2) I think nature's y seems like the purified the purified output control method in [R1]. Is that true? If yes can you please provide the relevant reference there?

3) I find it very confusing that $y$ suddenly becomes a decision variable in Section 3.1 while it represents the observations of the state in the previous chapter and also in the subsequent section. This is truly confusing if I did not miss something. Following that I do not understand why $u$ is the noise in Assumption 3.3.


4) What does $A_t$ represent in Algorithm 1? It seems completely irrelevant to the state transition matrix $A$ to me. Why do you choose the same letter? I find this very confusing.


5) Where is the proof of Theorem 3.7?

6) Line 529, I do not understand why $x_1, \ldots, x_{2\bar H +1} = argmin_{x \in \mathcal K} R(x)$ because it seems to me $x_1, \ldots x_{H } = argmin_{x \in \mathcal K} R(x)$ but $x_{H+1}, \ldots, x_{2 \bar H + 1}  = argmin_{x \in \mathcal K} \sum_{s= H}^t \frac{\sigma}{2} \| x - x_{s- \bar H}\|^2) + \eta^{-1} R(x)$. What am I missing here?

7) Can you comment on if it is possible to remove the logarithmic term even from the regret or is it fundamentally not possible? (in Theorem 4.1 and Theorem 4.2)

8) Why this rate is optimal? $\sqrt{T}$ regret rate would be optimal as it is tight. By not $\tilde{\mathcal{O}}(\sqrt(T))$. Can you please elaborate on that.

## Minor comments:

1) Typo in Page 4, Line 136: Time step instead of timestep.

2) Please provide references for every claim that is mentioned in the paper without a proof. For example, in Page 4, Line 145, I suggest to provide examples of the papers that make us of the system's counterfactual signal. Same goes for Page 5, Line 159.
Another example is in Page 5, Line 181. The sentence there reads "These conditions are relatively standard for bandit convex optimization algorithms, and are needed for the ..". Please provide references for the bandit convex optimization algorithms that make these standard assumptions.
Please provide references for Line 215 (which previous works).

3) Typo in Page 5, Line 157, "that that" should be "that".

4) Line 444: "empirical" instead of "emprical"
5) $W_t$ is not introduced in Line 235.
6) Line 197, what is $\Theta(H)$?
7) It's mandatory to include the code for the experiment section in Neurlps submissions. Your code seems to be missing. Please rectify this immediately by sharing your code via an anonymous GitHub repository
8) I'm having trouble understanding the meaning of the paragraph found on Line 196. Could you please clarify?
9) In Line 510, which spiking? I do not understand the paragraph in Line 509.
10) Why there is a divergence problem in the third plot of Figure 1?
11) I must express difficulty in comprehending the conclusion about the initial phase of EBPC possessing lower stability. The relationship with Figure 1 is unclear and requires further explanation. I am also perplexed by the decision to choose an extraordinarily high starting multiplier on ε - a justification is needed for this choice.
It's important to stress that the writing in the experiment section does not meet the expected standards for rigor. As it stands, the line of reasoning is opaque, and the process by which the conclusions were reached is indiscernible.
Further compounding the issue, the code has not been made available for examination. This, as previously emphasised, hampers any possibility of a comprehensive understanding or verification of the results. I strongly recommend addressing these concerns in a revised submission.
12) Line 504- Why this is a barely-stable system?
13) Line 261, if the key point is to use ellipsoidal gradient estimator, where is its definition?
14) In algorithm 1, $S^{n-1}$ is not defined.
15) Line 810- There is a typo.


References:

[R1] A. Ben-Tal, S. Boyd, and A. Nemirovski, Control of Uncertainty-Affected Discrete Time Linear Systems via Convex Programming. Technical Report Minerva Optimization Center, Technion, Haifa,
Israel, 2005 [Online]. Available: http://www.stanford.edu/~boyd/papers/pur_out_control.html


**Limitations:**

Yes.

---

> ### Author Rebuttal · Authors · 2023-08-07
>
> Thank you very much for your review and valuable feedback. Here are our responses to the points you raised:
>
> Weaknesses:
>
> **iuZo**: I must express my concerns about its presentation and structure. The writing is not as clear as it needs to be, making it quite challenging to follow the paper. The argument seems to lack the rigorousness and the self-containment.
> A crucial element that seems to be missing is detailed explanations for the conclusions drawn. Without these, it becomes arduous to justify the results. Enhancing the clarity and providing in-depth explanations of the methodologies and results will significantly improve the comprehension and overall impact of the study. I list my comments, questions and suggestions below.
>
> **Response**: Thank you for raising your concerns. In general, we agree that the presentation of our paper can be improved. We will add in more explanations on the intuitions and main ideas of our algorithms and theorems.
>
> Questions:
>
> Major comments:
>
> 1. The linear dynamic controller (LDC) policy class is optimal for this problem (proved by Kalman in the 1960s). The DRC policy class contains (or approximates up to arbitrary accuracy) the class of LDC (see Theorem 1 of “Improper Learning for Non-Stochastic Control” by Simchowitz et al. (2020)). We plan to clarify this context in the paper – thank you for pointing out the deficit.
>
>
> 2. We were not aware of this – thank you for providing the link to the paper! To the best of our knowledge, our nature’s y is the same as the purified output defined by 1.2 in [R1]. We will add this reference (which is also called the “Youla reparametrization” from the 1970`s literature in control, see references in Simchowitz et al.).
>
> 3. We wholeheartedly agree. This came about because we attempt to maintain consistency with the standard notations in both online bandit optimization and nonstochastic control, which causes overlapping use of letters. We differentiate the notations with bolded letters in the control setting and standard letters in the bandit settings, but we will change notations to make the presentation more clear, and avoid overloading.
>
> 4. Thank you for pointing this out. $A_t$ is not related to the state transition matrix (we are in the time-invariant setting, therefore $A$ is the state transition matrix) and is constructed in line 12 of Algorithm 1. We will change this for clarity.
>
> 5. The proof of Theorem 3.7 is in Appendix C.
>
> 6. You are right about $x_{H+1},\dots,x_{2\bar{H}+1}=argmin_{x\in\mathcal{K}}\sum_{s=H}^t \frac{\sigma}{2}|x-x_{s-\bar{H}}|^2+\eta^{-1}R(x)$. However, the equality $x_1,\dots,x_{2\bar{H}+1}=argmin_{x\in\mathcal{K}}R(x)$ still holds because the first $H$ $x$’s are the same. In particular, let us denote $x^*=argmin_{x\in\mathcal{K}}R(x)$, then we have for each $s=H+1,\dots,2\bar{H}+1$, $x_s=argmin_{x\in\mathcal{K}}\sum_{s=H}^t \frac{\sigma}{2}|x-x^*|^2+\eta^{-1}R(x)=x^*$.
>
> 7. No, it is not possible for any OCO-M based control algorithm.
>
> 8. You are correct; by “tight” we mean up to logarithmic terms in T, as with most ONC literature. We will make sure to clarify.
>
> Response to minor comments:
>
> For comments 1, 2, 3, 4, 15, thank you for pointing these issues out, and we will change according to your suggestions.
>
> 5: $W_t$ is the same as the $W_t$ in line 233.
>
> 6: $\Theta(\cdot)$ denotes asymptotic equivalence, as defined in line 482.
>
> 7: Thank you for asking. The link to the anonymous GitHub will be sent to the AC as instructed by the guidelines. Updated code for the modified experiments will follow with the modifications.
>
> 8: Previous work by Simchowitz et al. (2020) showed that the DRC policy class is a very rich policy class. See discussion to reviewer iuZo’s comment 1.
>
> 9: The spiking is not reflected visibly in the graphs (due to averaging across many runs). We will attempt to clarify this paragraph in the revision: it deals with a) the empirically motivated learning rate modification by Gradu et al. (see point 11) and b) adverse behavior observed in their code when the modification is made.
>
> 10: There is not a divergence problem: as we use DRC, the error in a generic system will reflect noise magnitude, which is increasing under random-walk noise. Note the essentially identical behavior in figure (1f) of Gradu et al.
>
> 11: The lower stability is reflected by a substantially slower decline to the minimum than the Gradu et al. algorithm under the empirically optimized multiplier – we will attempt to clarify this in the plots. The addition of the starting multiplier is not a novel addition for empirical tests, but was present in the tests of Gradu et al. (and is visible in their source) – we apologize if this was unclear. We intend to clarify the discussion of this choice (which was made by Gradu et al. to demonstrate empirically meaningful convergence results over a reasonable time threshold in the non-adversarial noise cases tested in that paper and here), as well as clarifying the presentation of the evidence for the conclusions drawn. For code, see point 7.
>
> 12: The system is barely stable because its eigenvalues have absolute values near 1. A system is stable in the strong sense if and only if its eigenvalues all have absolute values <1.
>
> 13: It is the $g_t$ defined in Algorithm 1. We will make sure we clarify that when introducing Algorithm 1.
>
> 14: Thank you for pointing this out. $S^{n-1}$ denotes the unit sphere in $\mathbb{R}^{n}$. We will add this into our notation section starting in line 119.
>
> Thank you again for your various comments, criticisms, and suggestions for improvement. We hope that our proposed changes address them to your satisfaction – please let us know if you have further questions or concerns.

---

> > ### Comment · Reviewer_iuZo · 2023-08-17
> >
> > Thanks a lot for the response.
> > About the 7th question in my major comments, can you explain why it is impossible for any OCO-M-based control algorithm? A reference would also work. I suggest including this in the paper for readers not working in this field.
> >
> > If the presentation of the paper is improved, it will have a better chance of being understood and used by other people. I certainly encourage the authors to do that in the revision.

---

> > > ### Author Response · Authors · 2023-08-17
> > >
> > > Thank you for the follow up question! The original OCO-M paper [1] is the fundamental framework that a long line of nonstochastic control (cited in our paper) is based on. While [1] does not explicitly provide a lower bound, following similar analysis as in [1], the memory length would factor in the regret bound in polynomial order. In the nonstochastic control setting, the memory length is taken to be $\log T$ due to the decaying effect of the past states in stable systems. This is why in this line of work, $\log T$ appears in the final bound. We will clarify this in the paper.
> > >
> > > [1]: Anava, Oren, Elad Hazan, and Shie Mannor. "Online learning for adversaries with memory: price of past mistakes." Advances in Neural Information Processing Systems 28 (2015).

---

> > > > ### Comment · Reviewer_iuZo · 2023-08-20
> > > >
> > > > Thank you very much for your detailed response!
> > > > I I will keep my score.

---

### Official Review · Reviewer_2R7J · 2023-07-06

**Soundness:** 3 good
**Presentation:** 3 good
**Contribution:** 3 good
**Rating:** 7
**Confidence:** 2

**Summary:**

This paper presents an algorithm to perform control in the semi-adversarial setting. After presenting the gap in the literature they want to fill, the authors present an algorithm that shows optimal regret ($O(\sqrt{T})$) for this setting, considerably improving the previous bound known from the literature ($O(T^{3/4})$, Gradu et al., 2020). The authors perform an experimental campaign to validate their solution in the appendix.

**Strengths:**

- The gap in the literature that the authors fill is clearly stated.

- The setting and the assumptions are clearly presented.

- The work, in general, is clearly presented (except for the algorithms, see Weaknesses).

- The authors face both the cases of known and unknown systems.

- The bounds presented are tight, given the lower bounds known from the literature.



**Weaknesses:**

- Algorithms 1, 2, and 3 should be explained instead of only presenting the pseudo-code. Some step may not be clear to all the readers only from the pseudo-code.

**Questions:**

No questions.

**Limitations:**

No limitations.

---

> ### Author Rebuttal · Authors · 2023-08-07
>
> Thank you very much for your review and feedback. Here is the response to your concern, with which we hope to further boost your confidence in our work:
>
> **2R7J**: Algorithms 1, 2, and 3 should be explained instead of only presenting the pseudo-code. Some steps may not be clear to all the readers only from the pseudo-code.
> **Response**: Thank you for pointing out this issue. We agree that the current presentation of the algorithms lacks an intuitive explanation and therefore limits the presentation clarity to a broader audience. We will add more explanations before introducing the pseudocode for every algorithm.

---

> > ### Comment · Reviewer_2R7J · 2023-08-19
> >
> > Thank you for your response. I will keep my score for now, waiting for the reviewers' discussion period.

---

### Official Review · Reviewer_7wFb · 2023-07-08

**Soundness:** 2 fair
**Presentation:** 2 fair
**Contribution:** 2 fair
**Rating:** 5
**Confidence:** 3

**Summary:**

This paper studies LQG with semi-adversarial disturbances and bandit feedback. The authors proposed a novel algorithm with $\sqrt T$ regret bound, which improves the $T^{3/4}$ regret bound in the literature. The algorithm utilizes the quadratic cost function structure and the ellipsoidal gradient estimator.

**Strengths:**

The paper follows the classic disturbance-action policy and online nonstochastic control and OCO-M framework. The novel part is the Elllipsoida BCO with Memory algorithm and its regret analysis. The algorithm motivation and the key steps are explained clearly.

The paper solves an interesting open question that whether $\sqrt T$ regret can be achieved for LQG under bandit feedback, under the assumption that the disturbances consist of both a large enough i.i.d. term and an adversarial term.

**Weaknesses:**

1. Simulation. The paper lacks simulation results for verification of the improved technical bound. Though the supplementary material provides some figures, the simulation setups are not clearly explained. See more questions in the next box.

2. Novelty. I have mixed feelings about the novelty of this paper. On the one hand, the paper successfully improves the regret bound order to $\sqrt T$. On the other hand, the majority of the algorithm design is based on the classic disturbance-action policy and OCO-M formulation, so I am not sure if the algorithm design is novel enough for a neurips publication.

3. Presentation. The paper seems to be written in a rush. For example, the title of the paper is inconsistent with the title on the webpage. The major algorithm design trick is the ellipsoidal gradient estimator, but the paper didn't explain this estimator or provide intuition. The main theorems are merely stated without any explanation.

**Questions:**

In the main paper,
Q1: the title is inconsistent with that on the openreview webpage. Please make them consistent.

Q2: Why do the authors consider semi-adversarial disturbances? What are the benefits and limitations of this setting? Why do the authors simulate i.i.d. disturbances instead of semi-adversarial disturbances?

In the supplementary material, for the simulation results:

Q3: Is the dynamical system known or unknown? If the system is unknown, how to obtain the optimal $K$ to the LQR problem? If the system is known, how does the algorithm perform under unknown dynamics?

Q4: What's the y axis? Why is it 0.1, 0.01? The authors skipped most parts of the simulation setup and referred the reader to Gradu et al. 2020. But the figures look very different from those in Gradu et al. 2020. It is hard to interpret the simulation results without future explanation.

Q5: The x axis scale goes as large as 100k, which is unreasonable for a control task. Is it possible to plot the figures in the same settings as in Gradu et al. 2020, where the horizon is usually within 1k?

---

> ### Author Rebuttal · Authors · 2023-08-07
>
> Thank you for taking the time to read and review our work. We believe that your concerns are valuable for us to improve our presentation. We hope that the following responses can explain our novelty and contribution better and boost your confidence in our work.
>
> Weakness comments:
>
> **7wFb**: Simulation. The paper lacks simulation results for verification of the improved technical bound. Though the supplementary material provides some figures, the simulation setups are not clearly explained. See more questions in the next box.
> **Response**: We are unsure about the intended meaning of the first comment – the simulation section shows that our algorithm improves on previous methods in practice, validating the theoretical findings. We are happy to substantially clarify the simulation setup (and the reasons behind it) in response to your concerns, and shift a summary to the main paper – thank you for pointing this out.
>
> **7wFb**: Novelty. I have mixed feelings about the novelty of this paper. On the one hand, the paper successfully improves the regret bound order to $\sqrt{T}$. On the other hand, the majority of the algorithm design is based on the classic disturbance-action policy and OCO-M formulation, so I am not sure if the algorithm design is novel enough for a neurips publication.
> **Response**: It is true that all recent nonstochastic control results are based on the paradigm of OCO-M, and DAC policies, so from a high level they look similar. The technical novelty of our paper is in a novel exploration scheme enabled by a new gradient estimator (from Suggala et al. (2021)). We show that this allows the optimal regret bound of $\sqrt{T}$, even in the generality of unknown systems with partially observed states. This improves the best known rate for this fundamental problem, and in fact is tight up to logarithmic factors.
>
> **7wFb**: Presentation. The paper seems to be written in a rush. For example, the title of the paper is inconsistent with the title on the webpage. The major algorithm design trick is the ellipsoidal gradient estimator, but the paper didn't explain this estimator or provide intuition. The main theorems are merely stated without any explanation.
> **Response**: For the title, see Q1 below. For the other points, we thank the valuable feedback and will amend the paper with intuition and nontechnical explanation for the algorithm components and the proof techniques/implications.
>
> Responses to questions:
>
> **Q1**: Thank you for pointing this out. We found the title “Optimal Rates for Bandit Nonstochastic Control” more descriptive than our original proposed title after the abstract submission deadline, and felt unsure whether we can change the title then. The title in the paper is our final decision. We will make sure the title is consistent moving forward.
>
> **Q2**: Semi-adversarial disturbances were chosen as they reflect a variety of real-world settings, in which adversarial agents exercise control only up to system noise (thermal/electrical/kinematic/etc.). Notably, said noise may be of small magnitude, and may in many cases be added manually via control, so it poses no meaningful barrier to application in the vast majority of practical settings. It is, however, a slightly weaker setting than the fully adversarial one – our algorithm’s performance does depend on noise magnitude, and is not shown to succeed in cases with absolutely zero non-adversarial noise. We simulate with various types of non-adversarial noise to comply with the established setting of Gradu et al. Testing with respect to adversarial noise is highly problem-specific, as determining worst-case noise inputs is an extremely challenging problem in general.
>
> **Q3**: Our contribution is for both known and unknown dynamical systems. This is discussed in detail in Section 4: Theorem 4.1 establishes the regret guarantee for LQR with known systems, and Theorem 4.2 establishes the regret guarantee with unknown systems. For the unknown system case, we run Algorithm 3 to estimate the system first and then run our controller algorithm. We showed that the system estimation error propagates through the controller algorithm in a way that the optimal rate is still preserved.
>
> **Q4**: The y-axis represents average error at the given timestep. Could you please clarify what you are referring to in “Why is it 0.1, 0.01”?  Are you asking about logarithmic vs. linear scale? The loss functions are quadratic, and convergence is of polynomial order, so there is no reason to choose a logarithmic scale. Broadly speaking, we intend to provide more methodological exposition for the experiments – see other comments.
>
> **Q5**: It is indeed possible to plot the figures with a smaller horizon; however, we chose not to do so for several reasons. First, part of the goal of the plots is to demonstrate the asymptotic difference between the convergence rate/convergence minimum of Gradu et al.’s work and our own, which is substantially more visible at higher timescales. Second, the low-time experimental setup of Gradu et al. required a large multiplier on learning rate in order to achieve practically-oriented results in settings without large-scale adversarial noise at low timescales (since the loss functions are prone to converge quickly here, artificially inflating the learning rate is advantageous). The advantage that this multiplier gives decreases at large timescales, reducing the gap between the theoretical and altered-practical performance and providing a more accurate representation.
>
>
> Thank you again for your various comments, criticisms, and suggestions for improvement. We hope that our proposed changes address them to your satisfaction – please consider raising your review score if so, and let us know if you have further questions or concerns.

---

> > ### Comment · Reviewer_7wFb · 2023-08-17
> >
> > Thank the authors for their responses! Somehow I find the authors' responses are in a defensive manner instead of trying to clarify my confusion. I hope that we can carry out more discussions to help improve the quality of the paper and resolve the confusions that readers may have. I provide more comments below and am happy to discuss more!
> >
> > **1. On simulation** I also don't fully understand the authors' response here. There isn't a simulation section in the manuscript of the main paper I read, but I may be looking at the wrong manuscript, e.g., an older version due to technical errors in openreview. Please correct me if I am wrong. Regarding the simulations provided in the appendix, it is less organized than a formal simulation section in the main paper and there are many simulation settings missing. I do believe that the numerical verification for this method is important so I would suggest adding a more organized simulation section to the main paper.
> >
> > **2. On novelty** I agree with the authors that the improved rate is important, as I mentioned in my original review. But I am still not fully convinced by the "novel exploration scheme enabled by the gradient estimator in Suggala et al. (2021)". Could the authors provide more discussions on the novelty of the exploration scheme, e.g., why is it different from traditional methods, what's the challenge, how to resolve it?
> >
> > **3. Title** Thanks for the clarification! I empathize with that.
> >
> > **4. semi-adversarial disturbances** The authors provide a very interesting motivation for the semi-adversarial disturbances: the actual control may be corrupted by noises caused by mechanics or system errors. I totally agree with that. But then, with the authors' discussion on the difficulties of the fully adversarial noises, does it mean that mechanics errors actually help with the control? So instead of improving the mechanics device, one should opt for more unreliable ones since it provides some excitation to the system? I am puzzled and fascinated by this counter-intuitive logic here.
> >
> > **5. On unknown system** Thanks for the response. Sorry for not posing my confusion clearer. I was thinking when the system is not open-loop stable, then it needs to be stabilized by a K. With uncertainties, it means a robust stabilizing K, which calls for a small system uncertainty otherwise it does not exist.
> >
> > **6. On the simulation** The authors explained the horizon choices well. Regarding the y axis, I was asking why the range of the y axis looks very different from that in [Gradu et al. 2020], whose y axis order is around 10. The authors omitted most of the simulation setting and referred to [Gradu et al. 2020] but even the scales of the figures are different... Can the authors explain more what the new changes they made for the simulation setting?

---

> > > ### Author Response · Authors · 2023-08-17
> > >
> > > Thank you for providing us with detailed follow-up questions! We are happy to clarify any areas where you found our responses insufficiently clear – please let us know. It's our goal to address your concerns to a satisfactory extent.
> > >
> > > 1. Indeed, the simulation section is contained in the appendix. As mentioned in our responses, we still plan to enhance the simulation section substantially and include more details about the simulation setting. To address your concern, we are happy to answer any questions you may have for our simulation sections here (and will add theses to the revised paper). We still plan to keep the detailed simulation section in the appendix (as we believe our main contribution is the theoretically tiger bound), but we will make sure to add a clear overview of the section to the main paper.
> > >
> > > 2. Online bandit algorithms rely on a stochastic unbiased/low-biased gradient estimator since the algorithm does not have direct access to the loss functions. The most traditional estimator used by [1] is a spherical estimator. That is, given an algorithm update $x_t$, sample $y_t$ from a sphere of appropriate size centered at $x_t$, and create the unbiased gradient estimator of $f_t(x_t)$ using $y_t$. This method does not take into account $f_t$’s structure. Later, [2] proposed an ellipsoid estimator that we make use of here: construct gradient estimator by sampling $y_t$ is instead sampled from an ellipsoid defined by $A_t$’s spectrum centered at $x_t$ (line 5 in algorithm 1). The key challenge here is that we want to extend the result of [2] to OCO with memory. In OCO with memory, the analysis follows from decomposing the regret into $(F_t(x_{t-H+1},\dots, x_t)-f_t(x_t))+(f_t(x_t)-f_t(x^*))$, where $f_t(x_t)=F_t(x_t,\dots,x_t)$. To bound the second term, it is essential that the gradient estimator $g_t$ is a low-bias estimator of $\nabla f_t(x_t)$. Our proposed gradient estimator achieves the low-bias guarantee. We show this by showing that $g_t$ is unbiased for the sum of partial gradients of $F_t(x_{t-H+1},\dots,x_t)$, which is motivated by the observation of unbiased gradient estimators for quadratic functions made by Suggala et al. (2021).
> > >
> > > [1]: Flaxman, Kalai, & McMahan. "Online convex optimization in the bandit setting: gradient descent without a gradient."
> > >
> > > [2]: Hazan & Levy. "Bandit convex optimization: Towards tight bounds."
> > >
> > > 4. This is subtle – it depends on what you mean by “help with the control.” The hard problem in nonstochastic control comes from the “nonstochastic” part. That is, the perturbations and loss functions to the system are selected by an adversary that acts more adversarially than stochastic systems. The intuition is that with the presence of some noise, the adversary’s ability is limited relative to the case in which the adversary also controlled that noise. If we consider a system that has only adversarial noise and add stochastic noise, the added noise does reduce regret, but only by increasing the “baseline” loss – that of the hindsight-best single-response controller. For instance, we add so much noise as to effectively drown out adversary perturbations, we may see our problem as being essentially stochastic, which makes it easier in a certain sense, but our absolute loss will of course be much higher. Your conclusion that “one should opt for more unreliable ones since it provides some excitation to the system” is therefore not quite right – when the problem parameters are still unset (for instance, if stochastic noise can be added or not), absolute error is not the same as regret, and we wish to minimize absolute error. We hope that this resolves your puzzlement.
> > >
> > > 5. Please let us know if we interpret the question wrong. If the question is what to do in the case of unknown stabilizable (but not stable) systems, our results may be generalized by the technique of learning a stabilizing controller. As mentioned in our discussion and conclusion section, Chen & Hazan showed stable controller recovery for unknown systems. We do not have this result for now, but [Chen & Hazan] makes this a future direction we can explore.
> > >
> > > 6. As everything is linear, the y-axis scale is simply a function of the noise magnitudes (which are different in our simulations, since we do not have access to the Gradu et al. simulation code). As a result, visual comparison of the charts is actually meaningful even without y-axis labels, since the results are invariant WRT this scale. Apart from that, the major nonspecified element is the exact nature of the sinusoidal noise (a sinusoidally varying scalar times a constant vector). We do not believe that there are any other meaningful unspecified changes to the setting (and our experimental results seem to conform to those of Gradu et al. in character) – please let us know if you spot any other missing details! We certainly intend to clarify the setting discussion in revision, including fleshing out the detail from the Gradu et al. setup.

---

> > > > ### Comment · Reviewer_7wFb · 2023-08-19
> > > >
> > > > Thanks for the detailed response! It resolved my concerns. But I still think more simulations are necessary even though I understand that the major contribution of this paper is to close a theoretical gap. I will raise my score.

---

> > > > > ### Author Response · Authors · 2023-08-19
> > > > >
> > > > > What additional simulations do you believe would be helpful (and what other properties do you believe they should elucidate)? We based our experimental suite off of Gradu et al. (which was designed to showcase a variety of common practical uses of control), but we would of course be interested to hear any additional suggestions you might have for properties that we could demonstrate better (either through more focused explanation of the current experiments or adding specific new ones).

---

### Official Review · Reviewer_6aiM · 2023-07-28

**Soundness:** 4 excellent
**Presentation:** 4 excellent
**Contribution:** 3 good
**Rating:** 7
**Confidence:** 4

**Summary:**

This paper studies the online control problem with a quadratic cost function, adversarially chosen dynamics, and semi-adversarially chosen losses (they are the sum of a zero-mean term and an obliviously chosen adversarial term). The algorithm proposed lowers the previous $T^{3/4}$ algorithm to a $\sqrt{T}$ algorithm against the class of disturbance response controllers, matching the lower bound. The crux of the analysis is handling the non-independence between plays and noise, which many OCO algorithms assume, and to carefully control the bias of the loss estimators. The authors solve the former with a novel method of using an online-learning with delay framework to bypass and the later by adapting an ellipsoidal gradient estimator.

**Strengths:**

Good organization, really clear presentation and summary of the high-level ideas used in this and previous work. Excellent literature review as well. The overall structure of the paper is well organized, and the algorithm follows naturally from all the components used.

The significance is high, as this paper really makes progress in closing off a field of online control by matching a lower bound. Much steady progress has been made on this problem over the years, so this paper is a great next step for this community. Tangentially, I think some of the ideas for algorithm 1 will be useful for the bandits-with-memory community.

**Weaknesses:**

Assumption 3.3 seems unnatural. Can you provide a bit more context on its necessity? More generally, how are ideas from learning with delay present in your algorithms? I did not find this point very clear.

Might be worth mentioning that there are empirical evaluations in the main body.

Finally, I wish the authors provided more intuition for how the proofs of the theorems worked; for example, what are the implications of using $\hat G$ instead of $G$?

**Questions:**

A few questions are asked above.

**Limitations:**

As a very theoretical work, I don't think any discussion about negative societal impact is necessary.

---

> ### Author Rebuttal · Authors · 2023-08-07
>
> Thank you for your thorough review. We respond to your points here:
>
> 1. **6aiM**: Assumption 3.3 seems unnatural. Can you provide a bit more context on its necessity? More generally, how are ideas from learning with delay present in your algorithms? I did not find this point very clear. **Response**: Assumption 3.3 is common (Gradu et al. 2020, for example) and turns out to be nearly minimal for bandit low-regret learning with memory. The noise in the semi-adversarial model is used by the gradient estimator, and if the adversary can observe and respond to that noise before the agent can, they can effectively remove it. In a case where agent state observation is delayed by one turn (via an off-diagonal transition matrix) and the adversary can condition response on the noise of previous steps, they can cause systematic misestimation of the gradient by effectively changing the noise distribution – or even remove its historical components entirely. Assumption 3.3 is analogous to the usual assumption that the adversary is not adaptive to the most recent algorithm decision (see, for instance, Flaxman et. al. (2008): “Online convex optimization in the bandit setting: gradient descent without a gradient”).
> Broadly, learning with memory is present as we attempt to learn a controller, but our choice of controller on a given step can affect our loss in subsequent steps. We will clarify this pursuant to point 3.
>
> 2. **6aiM**: Might be worth mentioning that there are empirical evaluations in the main body. **Response**: Thank you for pointing this out. We agree, and will add it in the introduction section.
>
> 3. **6aiM**: Finally, I wish the authors provided more intuition for how the proofs of the theorems worked; for example, what are the implications of using $\hat{G}$ instead of $G$? **Response**: First, let us address your specific question: $G$ is the Markov operator introduced in Definition 2.2 (line 151, page 4, main text) associated with the system. When the system is known, we can compute $G$ and therefore directly use $G$ in the algorithm. When the system is unknown, we use Algorithm 3 to estimate the Markov operator and obtain $\hat{G}$. The challenge in the unknown system case is that since $\hat{G}$ is an estimation of the true Markov operator $G$, the estimation error would affect the controls played by the algorithm. We showed that this estimation error would propagate through the algorithm in a way such that the $\tilde{O}(\sqrt{T})$-regret is still preserved for the unknown system (Theorem 4.2, main text). On the general point, we thank you (and other reviewers) for pointing this out, and intend to add substantially more intuition for both the critical parts of the theorems and the structure of the algorithms.
>
> Thank you again for your various comments, criticisms, and suggestions for improvement. We hope that our proposed changes address them to your satisfaction – please let us know if you have further questions or concerns.

---

> > ### Comment · Reviewer_6aiM · 2023-08-16
> >
> > Thanks for the response. Reading the reviews of the other vewiers and the responses, I agree with the crowd that the writing and presentation can be cleaned up, which the authors seemed to have committed too. I think the technical results are still strong, so I will keep my score.

---

### Decision · Program_Chairs · 2023-09-21

**Decision:**

Accept (poster)

**Comment:**

The paper received a uniformly positive evaluation from the reviewers. They generally appreciate the paper's theoretical contribution and agree that the paper addresses a significant problem in the field of control and online learning, by achieving an optimal regret bound in the considered setting. There were also some concerns and suggestions, which should be addressed by the authors in the final version of the manuscript.